# Function Classes for Identifiable Nonlinear Independent Component Analysis

**Simon Buchholz**, **Michel Besserve** & **Bernhard Schölkopf**
Max Planck Institute for Intelligent Systems
Tübingen, Germany
`{sbuchholz,mbesserve,bs}@tue.mpg.de`

## Abstract

Unsupervised learning of latent variable models (LVMs) is widely used to represent data in machine learning. When such models reflect the ground truth factors and the mechanisms mapping them to observations, there is reason to expect that they allow generalization in downstream tasks. It is however well known that such *identifiability* guaranties are typically not achievable without putting constraints on the model class. This is notably the case for nonlinear Independent Component Analysis, in which the LVM maps statistically independent variables to observations via a deterministic nonlinear function. Several families of *spurious solutions* fitting perfectly the data, but that *do not* correspond to the ground truth factors can be constructed in generic settings. However, recent work suggests that constraining the function class of such models may promote identifiability. Specifically, function classes with constraints on their partial derivatives, gathered in the Jacobian matrix, have been proposed, such as orthogonal coordinate transformations (OCT), which impose orthogonality of the Jacobian columns. In the present work, we prove that a subclass of these transformations, conformal maps, is identifiable and provide novel theoretical results suggesting that OCTs have properties that prevent families of spurious solutions to spoil identifiability in a generic setting.

## 1 Introduction

Unsupervised representation learning methods can fit Latent Variables Models (LVM) to complex real world data. While those latent representations allow to create realistic novel samples or represent the data in a compact way [32, 16], they are a priori not related to the ground truth generative factors of the data. It is highly desirable to recover the true underlying source distribution because those are expected to help with various downstream tasks, e.g., out of distribution generalization [45, 2].

One principled framework for representation learning is Independent Component Analysis (ICA) where one tries to recover unobserved sources $s \in \mathbb{R}^d$ from observations $x = f(s)$ and one assumes that the components $s_i$ are independent. An important result is that for linear functions $f$ it is possible to recover $s$ from observations $x$ up to certain symmetries, i.e., the model is *identifiable* [8]. In contrast, for general non-linear models $f$ is highly non-identifiable [25]. This has important consequences for representation learning, in particular the learning of disentangled representations is also unidentifiable without some access to the underlying sources [37]. Notably, this makes theoretical analysis of a large body of methods (see, e.g., [20, 30, 42]) that enforces disentanglement difficult.

Several additional assumptions were suggested to make the ICA problem identifiable. Broadly, there are two directions. First, some works imposed additional or different restrictions on the distribution of the sources. One line of research adds temporal structure by considering time series data [19, 23, 24]. More recently, Hyvärinen et al. [26] proposed to introduce an observed auxiliary variable $u$, e.g., a class label, such that the source distribution has independent components conditional on the auxiliary

variable. They show that under suitable assumptions on the distribution of $u$ and $s$ arbitrary nonlinear mixing functions can be identified. Several recent works extended this approach [29, 48, 53].

Another possibility is to restrict the class of admissible functions by considering more flexible classes than just linear functions, but not allowing arbitrary non-linear functions. The general aim of this approach is to find sufficiently "small" function classes such that ICA is identifiable within this class while making them as large as possible to allow flexible representation of complex data and being applicable to real world problems. So far results in this direction are rather limited. It was shown that the post-nonlinear model is identifiable [49]. Moreover, it has been shown that ICA with conformal maps in dimension 2 is almost identifiable [25]. More recently, identifiability of volume preserving transformations was investigated in the auxiliary variable case in [53] (combining the two possible restrictions) and identifiability based on sparsity of the mixing function was studied in [54].

In this work, we extend the previous works by proving new identifiability results for unconditional ICA. We consider conformal maps (i.e., maps that locally preserve angles) and Orthogonal Coordinate Transforms (OCT) (i.e., maps where $Df^\top Df$ is a diagonal matrix where $Df$ denotes the derivative of $f$). OCTs, that we will also call *orthogonal maps* for simplicity, were recently introduced in the context of representation learning in [17], motivated by the independence of mechanisms assumption from the causality literature. The main focus of this work is to prove new identifiability and partial identifiability results for this class of functions. An overview of our results can be found in Table 1. We summarize the main contributions and the structure of this paper as follows.

- We prove that ICA with conformal maps is identifiable in $d \geq 3$ and extend the earlier results in dimension 2 (Theorems 2-3 in Section 3).
- We define a new notion of local identifiability (Definition 5) and prove that ICA with orthogonal maps is locally identifiable (Theorem 4 in Section 4).
- On the contrary we show that ICA with volume preserving maps is not identifiable not even in the local sense (Theorem 6 in Section 5).
- We introduce new tools to the ICA field: our results are based on connections to rigidity theory (see Section 6), restricting the global structure of functions based on local constraints. Moreover, we exploit the global structure of partial differential equations related to the identifiability problem while most earlier results argue locally using linear algebraic tools.

## 2  Setting

Independent component analysis investigates the problem of identifying underlying sources when observing a mixture of them. We will consider the following general setting: there exists some random hidden vector of sources $s \in \mathbb{R}^d$ and the observed data is generated by

$$x = f(s), \qquad p_s(s) = \prod_{i=1}^d p_i(s_i) = \mathbb{P} \tag{1}$$

where $f : \mathbb{R}^d \to \mathbb{R}^d$ is smooth and invertible. The condition on $s$ means that its coordinates (often referred to as factors of variation) are independent. Formally this means that the distribution $\mathbb{P}$ of $s$ satisfies $\mathbb{P} \in \mathcal{M}_1(\mathbb{R})^{\otimes n}$ where $\mathcal{M}_1(\mathbb{R})$ denotes the probability measures on $\mathbb{R}$. The goal of ICA is to find an unmixing $g : \mathbb{R}^d \to \mathbb{R}^d$ such that $g(x)$ has independent components. Ideally, this should recover the true underlying factors of variation and achieve *Blind Source Separation* (BSS), i.e., $g = f^{-1}$ up to certain symmetries. Identification of the true generative factors of variations of the observations is of interest since these provide a causal and interventional understanding of the data.

An important observation was that, in the generality stated above, identification of $s$ is not possible. In [25] two general constructions of spurious solutions were given: the well known Darmois construction and a construction based on measure preserving transformations. The latter one is closer to our work here and we will discuss those in more detail in Section 4 and Appendix C. In a nutshell it is based on the observation that for measures $\mathbb{P}$ with smooth density one can construct smooth *Measure Preserving Transformations* (MPT), $m : \mathbb{R}^d \to \mathbb{R}^d$ (that mix the different coordinates), i.e., maps that leave $\mathbb{P}$ invariant, such that $m(s) \overset{\mathcal{D}}{=} s$ if $s \sim \mathbb{P}$.[1] This implies that all functions $(f \circ m)^{-1}$ recover independent sources since $(f \circ m)^{-1}(x) \overset{\mathcal{D}}{=} s$ making BSS impossible.

---

[1] We use the notation $X \overset{\mathcal{D}}{=} Y$ to indicate that the two random variables $X$ and $Y$ follow the same distribution

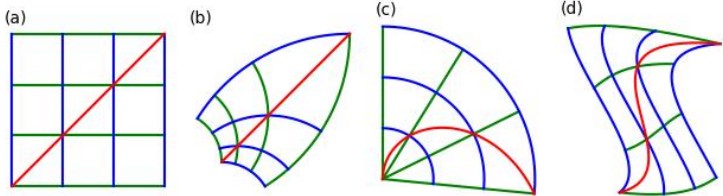

Figure 1: Illustration of the considered function classes. (a) shows a standard coordinate frame, (b) a conformal map applied to this frame which preserves angles, (c) an orthogonal map (polar coordinates) that preserve the orthogonality of lines parallel to the coordinate axes but not all angles (see red line), (d) a volume preserving map.

Thus it is a natural question whether additional assumptions on the mixing function $f$ or distribution of $s$ allow us to identify $f$. Let us define a framework for identifiability. We assume data is generated according to (1) where $f$ belongs to some function class $\mathcal{F}(A, B)$ of invertible functions $A \to B$, which we will always assume to be diffeomorphisms,[2] and $\mathbb{P}$ satisfies $\mathbb{P} \in \mathcal{P}$ for some set of probability distributions $\mathcal{P} \subset \mathcal{M}_1(\mathbb{R})^{\otimes d}$. Finally, let $\mathcal{S}$ be a group of transformations $g : \mathbb{R}^d \to \mathbb{R}^d$ that encodes the allowed symmetries up to which the sources can be identified as follows. The function class will be simply denoted $\mathcal{F}$ when domain/codomain information is irrelevant.

**Definition 1.** (Identifiability) We say that independent component analysis in $(\mathcal{F}, \mathcal{P})$ is identifiable up to $\mathcal{S}$ if for functions $f, f' \in \mathcal{F}$ and distributions $\mathbb{P}, \mathbb{P}' \in \mathcal{P}$ the relation

$$f(s) \stackrel{\mathcal{D}}{=} f'(s') \quad \text{where } s \sim \mathbb{P} \text{ and } s' \sim \mathbb{P}' \tag{2}$$

implies that there is $h \in \mathcal{S}$ such that $h = f'^{-1} \circ f$ on the support of $\mathbb{P}$.

Note that we require the identity $h = f'^{-1} \circ f$ only to hold on the support of $\mathbb{P}$ because, for complex classes $\mathcal{F}$, there is in general no unique extension of $f$ beyond the support of $\mathbb{P}$ and without data the extension cannot be identified. We do not always make this explicit in the following. Put differently, identifiability means that given observations of $x = f(s)$ and knowledge of $(\mathcal{F}, \mathcal{P})$, we can find $g$ such that $g \circ f \in \mathcal{S}$, in particular the reconstructed sources $s' = g(x)$ and the true sources $s$ are related by a symmetry transformation in $\mathcal{S}$. We discuss in Appendix C how to identify the set $\mathcal{S}$ and how spurious solutions to the identification problem can be constructed. In the following, it will be convenient to use the notation $f_*\mathbb{P}$ which denotes the push-forward of the measure $\mathbb{P}$ along the function $f$. We refer to Appendix A for a formal definition, but we note here that the distribution of $f(s)$ equals $f_*\mathbb{P}$ whenever $s \sim \mathbb{P}$. Therefore, (2) can be equivalently written as $f_*\mathbb{P} = f'_*\mathbb{P}'$.

We illustrate Definition 1 through the well known example of linear maps

$$\mathcal{F}_{\mathrm{lin}} = \{f : \mathbb{R}^d \to \mathbb{R}^d : f \text{ is linear and invertible}\}, \tag{3}$$

i.e., $x = As$ for some invertible matrix $A \in \mathbb{R}^{n \times n}$. We further define

$$\mathcal{P}_{\mathrm{lin}} = \{\mathbb{P} \in \mathcal{M}_1(\mathbb{R})^{\otimes d} : \text{at most one component of } \mathbb{P} \text{ is Gaussian}\}, \tag{4}$$

$$\mathcal{S}_{\mathrm{lin}} = \{P\Lambda : P \text{ is a permutation matrix and } \Lambda \text{ is a diagonal matrix}\}. \tag{5}$$

It is easy to check that $\mathcal{S}$ is a group. Then the following identifiability result for $\mathcal{F}_{\mathrm{lin}}$ is well known.

**Theorem 1.** (Theorem 11 in [8]) The pair $(\mathcal{F}_{\mathrm{lin}}, \mathbb{P}_{\mathrm{lin}})$ is identifiable up to $\mathcal{S}_{\mathrm{lin}}$.

This result is optimal as the ordering and scale of the $s_i$ is unidentifiable and the restriction to at most one Gaussian component is required to avoid linear MPTs of multivariate Gaussians. We provide a proof of this result in Appendix D as this serves as a preparation for the more involved Theorem 2 below.

An important observation which was also made in [22] is that with minor differences one can also consider the case where $f : \mathbb{R}^d \to M$ maps to a $d$-dimensional Riemannian manifold $M$. An

---

[2]A diffeomorphism is a differentiable bijective map with differentiable inverse.

Table 1: Overview of new identifiability results. Note that *Identifiable* implies *Locally identifiable* and if *Locally identifiable* does not hold neither of the other two properties can hold.

| Function class | Identifiable (Def. 1) | | Locally identifiable (Def. 5) | | Gaussian only spurious solution | |
|---|---|---|---|---|---|---|
| Linear | ✓ | | ✓ | | ✓ | |
| Conformal | ✓ | (Thm. 2 & 3) | ✓ | | ✓ | |
| Orthogonal | ? | | ✓ | (Thm. 4) | ✗ | (Prop. 1) |
| Volume preserving | ✗ | | ✗ | (Thm. 6) | ✗ | |
| General nonlinear | ✗ | | ✗ | (Lemma 1) | ✗ | |

important example for this setting is the case where $M \subset \mathbb{R}^n$ is a submanifold of a higher dimensional Euclidean space. This covers the standard setting of unsupervised representation learning where high dimensional observations (often images) are created from low dimensional factors of variation mirroring the well known manifold hypothesis [50]. Note that this setting essentially covers the case of undercomplete ICA, where we consider $f : \mathbb{R}^d \to \mathbb{R}^n$ with $n > d$. The only difference is that we assume that we already know the submanifold $M$ that $f$ maps to. This manifold can, however, be identified from the observations $x = f(s)$ under minor regularity assumption on $f$ and the support of the data distribution. To avoid technical difficulties we assume that the manifold is already known. Note that we restrict our attention to the case where the factors of variations are parametrized by a Euclidean space. An extension to product manifolds and a combination with the approach in [21] is an interesting question left to future work.

In the next sections we discuss our results on identifiability of ICA for different function classes. An illustration of the considered classes can be found in Figure 1. They are all characterized by a local condition on their gradient. Previously, in [22] it was shown that the function class of local isometries is identifiable. Local isometries have been used frequently in machine learning, and more specifically in representation learning [44, 50, 11]. Our main results consider two generalizations of these function classes, conformal maps and orthogonal coordinate transformations. Conformal maps preserve angles locally and have been used in computer vision [46, 51, 18]. For $d = 2$, conformal maps essentially consist of all biholomorphic mappings of simply connected open domains of the complex plane, and thus constitute a "large", non-parametric family, as a consequence of the Riemann Mapping Theorem [38]. For $d \geq 3$, Liouville's Theorem implies that this class contains relatively "few" functions in fixed dimension (i.e., mapping $\mathbb{R}^d \to \mathbb{R}^d$), in the sense that it is a parametric family, with parameter dimension quadratic in $d$ (see Theorem 7). However, this is a rich class when the target space is higher dimensional than the domain (i.e., $\mathbb{R}^d \to \mathbb{R}^n$, $d < n$). OCT are an even more general class that were motivated based on the principle of *independent causal mechanisms* in [17]. Notably, it contains all conformal maps precomposed with nonlinear entrywise reparametrizations of the source components (see Corollary 1). It is however much larger, as one can for example concatenate arbitrary functions from the large family of 2d conformal mappings to obtain higher dimensional OCTs. Moreover, many works showed that training VAEs promotes orthogonality of the columns of the input Jacobian [43, 55, 33, 40, 47] and this has been empirically shown to be a good inductive bias for disentanglement. Indeed, these algorithms are widely used in representation learning and often recover semantically meaningful representations [34, 5, 30, 20].

## 3   Results for conformal maps

Our first main result is an extension of Theorem 1 to conformal maps. A conformal map is a map that locally preserves angles, i.e. locally it looks like a scaled rotation. It can be shown that this is equivalent to the following definition.

**Definition 2.** (Conformal map) We define for domains $\Omega \subset \mathbb{R}^d$ the set of conformal maps by $\mathcal{F}_{\text{conf}} = \{f \in C^1(\Omega, \mathbb{R}^d) : Df(x) = \lambda(x)O(x)\}$ where $\lambda : \Omega \to \mathbb{R} \setminus \{0\}$ is a scalar function and $O : \Omega \to O(d)$ is a map to orthogonal matrices (i.e., $O(x)^{-1} = O(x)^\top$).

All our results also hold for the more general class of conformal maps $f : \Omega \to M$ where $M$ is a Riemannian manifold. The complete definition can be found in Appendix B. For convenience we define signed permutation matrices by

$$\text{Perm}_\pm(d) = \{P \in \mathbb{R}^{d \times d} : Q \in \mathbb{R}^{d \times d}, \text{ such that } Q_{ij} = |P_{ij}|, \text{ is a permutation matrix}\}, \quad (6)$$

i.e. the set of matrices whose entry-wise absolute value is a permutation. Later we will also use the notation $\mathrm{Diag}(d)$ and $\mathrm{Perm}(d)$ for $d \times d$ diagonal and permutation matrices, respectively. We define

$$\mathcal{S}_{\mathrm{conf}} = \{x \to \kappa P x + a \text{ where } P \in \mathrm{Perm}_\pm(d), a \in \mathbb{R}^d, \kappa \in \mathbb{R} \setminus \{0\}\} \tag{7}$$

and

$$\mathcal{P}_{\mathrm{conf}} = \mathcal{P}_1^{\otimes n} \cap \mathcal{P}_{\mathrm{lin}}, \qquad \text{where}$$
$$\mathcal{P}_1 = \{\mu \in \mathcal{M}_1(\mathbb{R}), \text{ there is } \emptyset \neq O \subset \mathbb{R} \text{ open, s.t. } \mu \text{ has positive } C^2 \text{ density on } O\}. \tag{8}$$

While this condition might appear a bit technical it actually only rules out pathological cases like the cantor measure or densities which are nowhere differentiable and probably it could be relaxed further. In particular $\mathcal{P}_1$ contains all probability measures with piecewise smooth densities. Then the following identifiability for conformal maps in dimension $d > 2$ holds.

**Theorem 2.** *For $d > 2$, ICA with respect to the pair $(\mathcal{F}_{\mathrm{conf}}, \mathcal{P}_{\mathrm{conf}})$ is identifiable up to $\mathcal{S}_{\mathrm{conf}}$.*

This means that we can identify conformal maps up to three symmetries, namely constant shifts of the distributions, rescaling of all coordinates by the same constant factor, and permutations of the coordinates. The proof is in Appendix E. The main ingredient in the proof is that conformal maps in dimension $d > 2$ are very rigid and can be characterized explicitly, as we will discuss in Section 6.

We remark that it might be more natural to not fix the scale of the sources and allow arbitrary coordinate-wise rescaling. The result can be easily extended to accommodate this. We define

$$\mathcal{S}_{\mathrm{reparam}} = \{g : \mathbb{R}^d \to \mathbb{R}^d | \, g = P \circ h \text{ where } P \in \mathrm{Perm}_\pm(d) \text{ and } h : \mathbb{R}^d \to \mathbb{R}^d \text{ with}$$
$$h(x) = (h_1(x_1), \ldots, h_d(x_d))^\top \text{ for some } h_i \in C^1(\mathbb{R}, \mathbb{R}) \text{ with } h_i' > 0\}. \tag{9}$$

It is easy to see that $\mathcal{S}_{\mathrm{reparam}}$ is a group. We define the class of parameterized conformal maps by $\mathcal{F}_{\mathrm{r-conf}} = \{f \circ h \, | f \in \mathcal{F}_{\mathrm{conf}}, h \in \mathcal{S}_{\mathrm{reparam}}\} \cap C^3(\Omega, M)$ and then get the following Corollary.

**Corollary 1.** *For $d > 2$, ICA with respect to the pair $(\mathcal{F}_{\mathrm{r-conf}}, \mathcal{P}_{\mathrm{conf}})$ is identifiable up to $\mathcal{S}_{\mathrm{reparam}}$ if we assume in addition that the observational distribution cannot be expressed as $f_*\mathbb{P}$ for some $f \in \mathcal{F}_{\mathrm{conf}}$ and $\mathbb{P} \in \mathcal{M}_1(\mathbb{R})^{\otimes d}$ which has at least two Gaussian components.*

The additional restriction on the observational distribution is clearly necessary to exclude the non-identifiability of Gaussian distributions.

For dimension $d = 2$ it was shown [25] that conformal maps can be identified up to a rotation when fixing one point of the conformal map (setting $f(0) = 0$). The authors also claim, without proof, that the remaining ambiguity can be removed for typical probability distributions. We extend their result by removing the condition that one point is fixed and prove full identifiability with a minor assumption on the involved densities. We define the following set of probability measures on $\mathbb{R}^2$

$$\mathcal{P}_{\mathrm{conf2}} = \{\mathbb{P} = \mathbb{P}_1 \otimes \mathbb{P}_2 \in \mathcal{M}(\mathbb{R})^2 | \text{ s.t. } \mathrm{supp}(\mathbb{P}_i) \text{ is a bounded interval } I_i \text{ and}$$
$$\mathbb{P}_i \text{ has density bounded above and below on } I_i\} \tag{10}$$

Then we get the following result.

**Theorem 3.** *For $d = 2$, ICA with respect to the pair $(\mathcal{F}_{\mathrm{conf}}, \mathcal{P}_{\mathrm{conf2}})$ is identifiable up to $\mathcal{S}_{\mathrm{conf}}$.*

This means that we can identify conformal maps on compact domains in dimension 2 up to shifts, permutations of coordinates, and scale. Note that we can also identify conformal maps if $\mathbb{P}$ has full support $\mathbb{R}^2$ using the same proof as for $d > 2$ (see Lemma 2 in the supplement) and an extension as in Corollary 1 is possible. The proof of this result is in Appendix E.

## 4   Results for orthogonal maps

Recently, in [17], the more general class of OCTs was considered in the context of ICA. They referred to orthogonal coordinates as IMA maps, referencing to independent mechanisms. This nomenclature was motivated by the causality literature and we refer to their paper for an extensive motivation and further results. As we focus on theoretical results for this function class we stick to the more common term of OCTs. Orthogonal coordinate transformations are defined as the set of functions whose derivative have orthogonal columns, i.e., the vectors $\partial_i f$ and $\partial_j f$ are orthogonal for $i \neq j$.

**Definition 3.** (OCT maps) We define for domains $\Omega \subset \mathbb{R}^d$ the set of OCT maps (orthogonal coordinates) by $\mathcal{F}_{\mathrm{OCT}} = \{f \in C^1(\Omega, \mathbb{R}^d) : Df(x)^\top Df(x) \in \mathrm{Diag}(d)\}$.

**OCTs constitute a rich class of functions.** The study of OCTs has a long history and already in the 19th century the structure of all OCTs defined in a neighbourhood of a point were characterized [10, 4]. Later, also the set of global orthogonal coordinate systems on $\mathbb{R}^d$ was characterized [28]. As those results are not easily accessible we will provide here a simple argument showing that OCTs constitute a rich class of functions. We first note that $\mathcal{F}_{\mathrm{OCT}}$ contains the above $\mathcal{F}_{\mathrm{r-conf}}$, as functions in the later class have a $Df(x)$ that takes the form of a Jacobian of a conformal map whose columns are rescaled by derivatives of the entry-wise reparametrizations, such that they remain orthogonal. However, $\mathcal{F}_{\mathrm{OCT}}$ is much bigger than $\mathcal{F}_{\mathrm{r-conf}}$. For example, take a $n$-tuple $(f^1, ..., f^n)$ of arbitrary injective 2D conformal maps $f^k : \Omega_k \to \mathbb{R}^2 \in \mathcal{F}_{\mathrm{conf}}$ where $\Omega_k \subset \mathbb{R}^2$ and build the "concatenated" map $f_{\mathrm{conc}} : \Omega_1 \times \cdots \times \Omega_n \to \mathbb{R}^{2n}$ given by

$$f_{\mathrm{conc}}(s) = (f_1^1(s_1, s_2), f_2^1(s_1, s_2), \dots, f_1^n(s_{2n-1}, s_{2n}), f_2^n(s_{2n-1}, s_{2n}))^\top . \tag{11}$$

The Jacobian of $f_{conc}$ is block diagonal, such that columns associated to different diagonal blocks are obviously orthogonal, and columns pertaining to the same k-th diagonal block are orthogonal by conformality of $f_k$. With such a construction, that we can also further post-compose with transformations in $\mathcal{F}_{\mathrm{conf}}$ on $\mathbb{R}^{2n}$, we can thus build a large non-parametric subclass of $\mathcal{F}_{\mathrm{OCT}}$ on $\mathbb{R}^{2n}$. This construction can be easily adapted to the case of odd dimensions.

**Setting for identifiability with OCTs.** OCTs can also be generalized to maps whose target is a $d$-dimensional manifold (see definition in Appendix B), and the following results will also apply to such case. First, we note that we can only hope to identify a mechanism $f \in \mathcal{F}_{\mathrm{OCT}}$ up to coordinate-wise transformations and permutations, i.e., maps in $\mathcal{S}_{\mathrm{reparam}}$. Indeed, if $f \in \mathcal{F}_{\mathrm{OCT}}$ and $g \in \mathcal{S}_{\mathrm{reparam}}$ then $f \circ g \in \mathcal{F}_{\mathrm{OCT}}$. Thus, in particular $f_*\mathbb{P} = (f \circ g)_*(g^{-1})_*\mathbb{P}$. This implies that given observations from $f_*\mathbb{P}$ we can identify $f$ and $\mathbb{P}$ only up to $g \in \mathcal{S}_{\mathrm{reparam}}$. More precisely, for any (sufficiently smooth) $\mathbb{P}'$ there is $f'$ such that $f_*\mathbb{P} = f'_*\mathbb{P}'$ where we pick $g$ such that $\mathbb{P}' = g_*^{-1}\mathbb{P}$. [3]

As the distribution of the $s_i$ is not identifiable, we map it to a fixed reference distribution that we choose to be the uniform distribution on $(0, 1)^d$. We introduce the shorthand $C_d = (0, 1)^d$ for the standard open unit cube (exclusion of the boundary will be important for our result) and denote by $\nu$ the uniform (Lebesgue) measure on $C_d$. For fixed base measure $\nu$ the symmetry group is reduced to permutations and reflections, i.e., maps in $P \in \mathrm{Perm}_\pm(d)$.

We conjecture that for 'typical' pairs $(f, \mathbb{P}) \in \mathcal{F}_{\mathrm{OCT}} \times \mathcal{P}_{\mathrm{OCT}}$, ICA is identifiable with respect to $\mathcal{S}_{\mathrm{reparam}}$ (with a suitable definition of $\mathcal{P}_{\mathrm{OCT}}$, e.g., $\mathcal{P}_{\mathrm{OCT}} = \mathcal{P}_{\mathrm{conf}}$). However, we leave a precise statement for future work. Below we will show a weaker notion of identifiability for OCTs, but, before that, we first exhibit exceptional classes of spurious solutions for ICA with OCTs.

**Spurious solutions for ICA with OCTs.** We first note that just as for linear maps and conformal maps (see Thm. 1) Gaussian distributions can hamper identification. This is because arbitrary measures with a factorized density can be pushed forward into multivariate Gaussians using a suitable coordinate-wise transformation.

*Fact* 1. Let $\mathbb{P}$ be a probability measure on $\mathbb{R}$ with bounded density $p$ and cumulative distribution function $F_\mathbb{P}$. Denote the cumulative distribution function of the standard normal by $F_\mathcal{N}$. Let $h_\mathbb{P} = F_\mathcal{N}^{-1} \circ F_\mathbb{P}$. Then $(h_\mathbb{P})_*\mathbb{P}$ has a standard normal distribution.

This implies that if $\mathbb{P} = \mathbb{P}_1 \otimes \ldots \otimes \mathbb{P}_d$ and $x = f(s) = (h_{\mathbb{P}_1}(s_1), \ldots, h_{\mathbb{P}_d}(s_d))$ then $f_*\mathbb{P}$ follows a standard normal distribution. In particular, for every $A \in \mathrm{O}(d)$ the distribution of $Ax$ is standard normal and its components are independent. Note that, in contrast to conformal and linear maps, it is not sufficient to exclude Gaussian source distributions: due to the flexibility of the function class, we have to exclude that the pair $(f, \mathbb{P})$ has a Gaussian observational distribution $f_*\mathbb{P}$. Next we show that a more general construction using OCTs is possible.

**Proposition 1.** *Let $\mathbb{P}$ be a rotation invariant distribution on $\mathbb{R}^d$ with smooth density. Then there is a smooth and invertible (on its image) function $f : C_d \to \mathbb{R}^d$ with $f \in \mathcal{F}_{\mathrm{OCT}}$ such that $f_*\nu = \mathbb{P}$.*

The proof of this result can be found in Appendix F. The main idea in the proof is that $d$-dimensional polar coordinates do the trick up to coordinate-wise rescaling. This proposition implies that the entire family $\{f_R = R \circ f \mid R \in O(n)\}$ satisfies $(f_R)_*\nu = f_*\nu$ and, by definition of $\mathcal{F}_{\mathrm{OCT}}$ we have

---

[3]This is possible if both distributions have compact connected support where they have a smooth positive density. We ignore difficulties associated with unbounded support or non-regular measures here

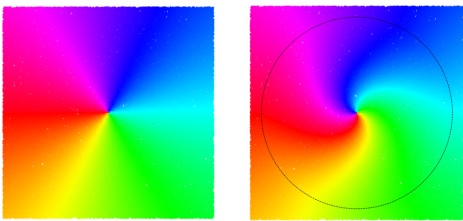
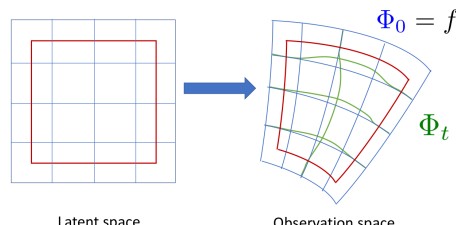

Figure 2: Illustration of radius dependent rotations as defined in Lemma 1. The left figure shows the initial sources. In the right figure a radius dependent volume preserving transformation was applied (see Appendix C).

Figure 3: Smooth invariant deformations. The blue grid indicates the transformation $f$, while the green grid shows the deformed map $\Phi_t$. Outside the red box $\Phi_t = f_t$ holds (see Definition 5).

$f_R \in \mathcal{F}_{\mathrm{OCT}}$ because $f \in \mathcal{F}_{\mathrm{OCT}}$ implies $R \circ f \in \mathcal{F}_{\mathrm{OCT}}$. In particular all inverses $g_R = f_R^{-1}$ recover independent sources because $(g_R)_* f_* \nu = \nu$ and BSS is not possible in a meaningful way in this (special) case. The construction in Proposition 1 gives spurious solutions for substantially more observational distributions than just the Gaussian (this is indicated in Table 1). Nevertheless we do not view this as a general obstacle to identifiability results for OCTs for two reasons. Firstly the spurious solutions only apply to carefully chosen pairs of function and base measure such that we obtain a still very non-generic (radial) observational distribution. Moreover, the function constructed in Proposition 1 cannot be extended to $\overline{C_d} = [0,1]^d$ such that it remains invertible (in the language of differential geometry: $f$ is only an immersion not an embedding of a submanifold). In this sense the main message of Proposition 1 is that an identifiability result for $\mathcal{F}_{\mathrm{OCT}}$ needs to contain assumptions ruling out those spurious solutions (just like Gaussians are excluded for linear ICA).

**Local Stability of OCTs.** We now give partial results towards identifiability of OCTs. While we do not prove general identifiability for this class, we demonstrate their *local rigidity*: OCTs cannot be continuously deformed to obtain spurious solutions. This is in stark contrast to the general nonlinear case, which we will discuss for comparison below. Actually, we will show the following slightly stronger statement. Suppose we know some initial data generating mechanism $f_0$ because we have, e.g., access to samples $(s, f_0(s))$. Now we assume that the mixing $f_t$ depends smoothly on some parameter $t$ which could be, e.g., time or an environment. Then we can identify $f_t$ if $f_t \in \mathcal{F}_{\mathrm{OCT}}$ for all $t$ and given access to samples $f_t(s)$. This would not be true if $f_t$'s would be unconstrained nonlinear mixing functions. We now make these statements precise. We first define smooth deformations of a mixing function.

**Definition 4.** (Smooth invariant deformations) Let $\mathcal{F}$ be some function class. Consider a family of differentiable transformations $\Phi \in C^1((-T, T) \times \mathbb{R}^d, M)$ for some $T > 0$ and a smooth, $d$-dimensional manifold $M$ such that $\Phi_t = \Phi(t, \cdot)$ is a diffeomorphism onto its image. We call $\Phi_t$ a smooth invariant deformation if $\Phi_t(\cdot) \in \mathcal{F}$ for all $t$.

An illustration of this definition can be found in Figure 3. Based on invariant deformations we can now define a local identifiability property of ICA in a given function class.

**Definition 5.** (Local identifiability of ICA). Consider a function class $\mathcal{F}$. Let $f_t$ be a smooth invariant deformation in $\mathcal{F}$ that is analytic in $t$. Let $\Phi_t$ be another smooth invariant deformation in $\mathcal{F}$ analytic in $t$ such that $\Phi_0 = f_0$ and $(\Phi_t)_* \nu = (f_t)_* \nu$ for all $t$. Assume that there is $\varepsilon > 0$ such that $\Phi_t(s) = f_t(s)$ if $\mathrm{dist}(s, \partial C_d) < \varepsilon$. Then we say that ICA in $\mathcal{F}$ is *locally identifiable at* $(f_t, \nu)$ if these assumptions imply $\Phi_t = f_t$ for all $t$. We call $\mathcal{F}$ *locally identifiable* if it is locally identifiable at $(f_t, \nu)$ for all analytic local deformations $f_t$.

We remark that our notion of local identifiability is indeed similar to the concept of local identifiability in the context of parameter identification of neural networks [3]. Local identifiability of a function class means that we can identify smooth deformations $f_t \in \mathcal{F}$ from some initial mixing $f_0 \in \mathcal{F}$ if $f_0$ is known on the whole latent domain, and the behaviour of $f_t$ close to the domain's boundary is known for all $t$. This can be interpreted as plain identifiability in the concept drift setting [52, 15]. Formulated differently, it means that an adversary cannot smoothly deform the function $f_0$ in a subset whose boundary is away from the boundary of the domain, such that the outcome is ambiguous given

the resulting observational distributions. An experimental illustration of this setting and Theorem 4 below can be found in Appendix H. The notion of locality in Definition 5 should be understood as "non-global" and notably does not imply restrictions to a small neighbourhood, as local properties often do. The non-globality manifests itself in two ways: we consider smooth transformations of the ground truth, i.e., small changes of the data generating function $f_0$ and in addition we assume that the changes are not everywhere in $s$, i.e., sources $s$ close to the boundary are kept invariant. An extension to other source distributions beyond fixed $\nu$ (and possibly changing with $t$) is possible but not necessary in the context of OCTs as explained above. We can show that local identifiability holds true in $\mathcal{F}_{\mathrm{OCT}}$.

**Theorem 4.** *The function class $\mathcal{F}_{\mathrm{OCT}}$ is locally identifiable.*

The proof, in Appendix F, introduces new tools to the field of ICA. The main idea is to consider the vector field $X$ that generates the deformation $\Phi_t$ and then rewrite the assumption as systems of partial differential equations for $X$. The proof is then completed by showing that the only solution of this system vanishes. Let us state one simple consequence of this theorem.

**Corollary 2.** *Let $\Phi_t$ be a smooth analytic invariant deformation in $\mathcal{F}_{\mathrm{OCT}}$ such that $(\Phi_t)_*\nu = f_*\nu$ for all $t$ and there is $\varepsilon > 0$ such that $\Phi_t(s) = f(s)$ if $\mathrm{dist}(s, \partial C_d) < \varepsilon$. Then $\Phi_t = f$ for all $t$.*

Let us reiterate what this corollary shows: we cannot smoothly and locally transform the function $f$ such that (1) the observational distribution remains invariant, i.e., equal to $f_*\nu$, and (2) the deformed functions remain OCTs.

At a high level this result suggests that OCTs can be identified if we know $f$ close to the boundary of the support of $s$, e.g., by having, in addition to unlabelled data $f(s)$, labelled data $(s, f(s))$ for those $s$ where one coordinate $s_i$ is extremal. Note that we actually do not show this result as there might be further solutions which are not connected by smooth transformations. We expect that those results can be generalized substantially. In particular, we conjecture that for "most" functions $f$ the boundary condition can be removed thus giving a stronger local identifiability result up to the boundary of the support of $s$. As a partial result in this direction we prove the following theorem.

**Theorem 5.** *Let $f : C_d \to \mathbb{R}^d$ be given by $f(x) = RDx$, with $R \in \mathrm{O}(d)$ and $D = \mathrm{Diag}(\mu_1, \ldots, \mu_d)$ where $\mu_i$ are i.i.d. samples from a distribution supported on the positive reals $\mathbb{R}_+$ which has a density. Suppose that $\Phi_t$ is a smooth invariant deformation in $\mathcal{F}_{\mathrm{OCT}}$ such that $\Phi_0 = f$, $(\Phi_t)_*\nu = f_*\nu$, and $\Phi_t$ is analytic in $t$. Then for almost all $\mu_i$ (i.e., with probability one) this implies $\Phi_t = f$ on $C_d$, i.e., $\Phi_t$ is constant in time.*

In Appendix F we show that this theorem follows from a slightly stronger result stated as Theorem 10 which has a similar proof as Theorem 4. We do expect that the conclusion of the Theorem actually holds for all $\mu_i$ not just almost all, but we are unable to show this.

**Comparison with ICA for general nonlinear functions.** Let us emphasize that those results are non-trivial as they establish a large difference between ICA with generic nonlinear maps and ICA with OCTs. To clarify this, we state that no result similar to Theorem 4 holds without the assumption that $\Phi_t \in \mathcal{F}_{\mathrm{OCT}}$. Put differently, the function class $\mathcal{F}_{\mathrm{nonlinear}}$ is not locally identifiable.

*Fact* 2. Suppose $f : C_d \to \mathbb{R}^d$ is a diffeomorphism on its image. Then there are uncountably many smooth deformations $\Phi_t$ of $(f, \nu)$ such that (i) $(\Phi_t)_*\nu = f_*\nu$ and (ii) there is an $\varepsilon > 0$ such that $\Phi_t(s) = f(s)$ whenever $\mathrm{dist}(s, \partial C_d) < \varepsilon$.

For completeness, we provide a general construction that is close to our proof of Theorem 4 in Appendix C in the supplement. A very clear construction for this result was given in [25].

**Lemma 1** (Smoothly varying radius dependent rotations (see [25])). *Let $R : \mathbb{R} \times \mathbb{R}_+ \to \mathrm{O}(d)$ be a smooth function mapping to orthogonal matrices and let $a \in C_d$. Assume that $R(t, r) = \mathrm{id}$ for $r \geq \mathrm{dist}(a, \partial C_d)$. Then the map $s \to h_{R,a}(t, s) = R(|s - a|, t)(s - a) + a$ preserves the uniform measure $\nu$ on $C_d$ for all $t$ so that $f \circ h_{R,a}(t, s) \overset{\mathcal{D}}{=} f(s)$ for all $t$ if $s$ is distributed according to $\nu$.*

An illustration of this construction is shown in Figure 2 (see App. C for details). Clearly, by concatenation this allows us to create a vast family of spurious solutions. Note that those solutions are excluded when restricting to OCTs which is a corollary of Theorem 4.

**Corollary 3.** *Suppose $f \in \mathcal{F}_{\mathrm{OCT}}$. Let $\Phi_t$ be the smooth invariant deformation defined by $\Phi_t = f \circ h_{R,a}(t, \cdot)$ where $h_{R,a}$ is as in Lemma 1. If $\Phi_t \in \mathcal{F}_{\mathrm{OCT}}$ for all $t$ this implies that $h_{R,a}(t, s) = s$ and $\Phi_t = f$ for all $t$.*

We now summarize our view on the results of this section informally (we do not claim that the statements below regarding (infinite dimensional) manifolds can be made rigorous). For a given data generating mechanism $(f_0, \nu)$ we expect that typically the set of all solutions $M_{\mathrm{OCT}} = \{f \in \mathcal{F}_{\mathrm{OCT}} | \ f_* \nu = (f_0)_* \nu\} \subset \mathcal{F}_{\mathrm{OCT}}$ is a zero dimensional submanifold, i.e., consists of isolated spurious solutions and we prove this when fixing the boundary (see Theorem 4) while the corresponding submanifold of general nonlinear spurious solutions $M_{\mathrm{nonlinear}} = \{f : C_d \to \mathbb{R}^d | \ f_* \nu = (f_0)_* \nu\}$ is infinite dimensional even when requiring $f(s) = f_0(s)$ close to the boundary of $C_d$.

# 5 Results for volume preserving maps

Let us finally consider volume preserving transformations. For $\Omega \subset \mathbb{R}^d$, those are defined as the set of functions $\mathcal{F}_{\mathrm{vol}} = \{f : \Omega \to \mathbb{R}^d | \ \det Df(x) = 1 \text{ for all } x \in \Omega\}$. Invertible volume preserving deformations have the property that they preserve the standard (Lebesgue)-measure $\lambda$ in the sense that $f_* \lambda_\Omega = \lambda_{f(\Omega)}$. Recently it was proposed that volume preserving functions are a suitable function class for ICA. Here we show that those functions are not sufficiently rigid to allow identifiability of ICA in the unconditional case. Note that Lemma 1 and Fact 2 already show how to construct spurious solutions for the case that the base distribution is the uniform measure $\nu$. However, for an arbitrary distribution $\mathbb{P}$ this is slightly more difficult because we need to find maps $g$ that preserve $\mathbb{P}$, i.e., $g_* \mathbb{P} = \mathbb{P}$, and are volume preserving, i.e., preserve the standard measure.

**Theorem 6.** *Let $p$ be a twice differentiable probability density with bounded gradient. Suppose that $x = f(s)$ where the distribution $\mathbb{P}$ of $s$ has density $p$ and $f$ is a diffeomorphism with $\det Df(x) = 1$ for $x \in \mathbb{R}^d$. Then there is a family of functions $f_t : \mathbb{R} \times \mathbb{R}^d \to \mathbb{R}^d$ with $f_0 = f$ and $f_t \neq f_0$ for $t \neq 0$ such that $\det Df_t(x) = 1$ and $(f_t)_* \mathbb{P} = f_* \mathbb{P}$.*

The proof and an illustration are in Appendix G. It is based on the flows generated by suitable explicit vector fields. As those flows can be concatenated we obtain a large family of spurious solutions. We think that the approach used here is a powerful technique to construct counter-examples to identifiability in ICA. As local identifiability is weaker than identifiability we conclude that ICA in $\mathcal{F}_{\mathrm{vol}}$ is not identifiable. Note that this is even true when we know the distribution of $s$. A rigorous version of this statement is that if $(\mathcal{F}_{\mathrm{vol}}, \mathcal{P}_{\mathrm{vol}})$ is identifiable with respect to $\mathcal{S}_{\mathrm{vol}}$ then $\mathcal{S}_{\mathrm{vol}}$ contains functions mixing coordinates $s_i, s_j$ for $i \neq j$ (i.e., there is $h \in \mathcal{S}_{\mathrm{vol}}$ such that $\partial_i \partial_j h \neq 0$).

# 6 Relation to rigidity theory

Our results rely on rigidity properties of certain function classes. Rigidity refers to the property that a local constraint on the derivative of a function implies global restrictions on the shape of the function. These type of results are of interest in the field of continuum mechanics [6] where, e.g., the condition $Df \in \mathrm{SO}(d)$ is used to describe deformation of rigid solids and the condition $\mathrm{Det}\, Df = 1$ to describe incompressible fluids.

We now state a well known rigidity result for conformal maps that is the main input in the proof of Theorem 2. This result shows that there are very few conformal maps in dimension $d > 2$.

**Theorem 7** (Liouville). *Let $d > 2$, $\Omega \subset \mathbb{R}^d$ open and connected, $f : \Omega \to \mathbb{R}^d$ conformal. Then*

$$f(x) = b + \alpha A(x - a)/|x - a|^\varepsilon \tag{12}$$

*where $b, a \in \mathbb{R}^d$, $\alpha \in \mathbb{R}$, $A \in \mathrm{O}(d)$, and $\varepsilon \in \{0, 2\}$.*

Originally this result was shown by Liouville [36], a modern treatment is [27]. In particular, this shows that conformal maps are (up to translations) rotations or rotations followed by an inversion.

To illustrate the strength of Theorem 7 we compare it to the setting of volume preserving maps which satisfy no similar rigidity property. Intuitively the different rigidity properties are already apparent from the connection to solids, which can merely be rotated and shifted, and fluids which can also be stirred leading to chaotic deformations. Rigorously the different behaviours can be clarified by the observation that conformal maps have a finite number of parameters and thus a finite number of constraints (e.g., of the form $f(x) = y$) allows to identify them. In contrast volume preserving maps cannot be identified from finitely many constraints as the following proposition shows.

**Proposition 2.** *For $d > 2$ and $\{x_1, \ldots, x_n, y_1, \ldots, y_n\} \subset \mathbb{R}^d$, all pairwise different, there is a volume preserving diffeomorphism $f : \mathbb{R}^d \to \mathbb{R}^d$, $f \in \mathcal{F}_{\mathrm{vol}}$ such that $f(x_i) = y_i$.*

Let us emphasize that the different rigidity properties are not at all surprising when arguing based on degrees of freedom or numbers of constraints. While volume preserving maps enforce only a single scalar constraint on the Jacobian $Df$ the condition for conformal maps gives $n(n+1)/2 - 1$ constraints on the Jacobian.

Let us finally comment on OCTs where the picture is not as well understood. As discussed in Section 4 it is known [10, 4] that OCTs constitute a rich, non-parametric class of functions and therefore OCTs are much more flexible than conformal maps. We illustrated this with example OCT constructions in Section 4 leveraging 2D conformal maps. Nevertheless, it is not known if and what rigidity properties can be derived for OCTs. However, our results suggest that the additional measure preservation condition in the context of ICA gives enough rigidity to (almost) give identifiability of ICA. In this sense OCTs might be a good function class for ICA as it is rich enough to allow complex representations of data while at the same time being sufficiently rigid to still provide a notion of identifiability whose strength remains to be determined.

## 7 Discussion

ICA is long known to be identifiable for linear maps, baring pathological cases, and highly non-identifiable for general nonlinear ones. Surprisingly, similar results for function classes of intermediate complexity remain scarce. In this work we address this question with several identifiability results for different function classes. Our first main result is that ICA is identifiable in the class of conformal maps (up to classical ambiguities). This considerably extends previous claims, limited to a specific 2D setting [25], and ruling out several families of spurious solutions [17]. On the negative side we show that the ICA problem for volume preserving maps admits a large class of spurious solutions. Finally, we show that OCTs satisfy certain weaker notions of local identifiability.

In our proofs, we draw connections to methods and techniques that, to the best of our knowledge, have not been used in the context of ICA before. We relate the identifiability problem in ICA to the rigidity of the considered function class $\mathcal{F}$ and use tools from the theory of partial differential equations. These techniques have been applied very successfully to the analysis of elastic solids [7, 6] and we believe that there are many applications of these methods in the field of ICA.

While the main focus of current research after the seminal work of Hyvärinen et al. [26] is on the auxiliary variable case, there are three reasons to consider unconditional ICA. Firstly, it is a fundamental research question that is, as illustrated by our results, deeply rooted in functional analysis. Secondly there is high application potential for completely unsupervised learning without any auxiliary variables, as the corresponding datasets do not require labelling or specific experimental settings. Thirdly, it is very likely that the techniques can be generalized to the auxiliary variable case.

Another important open problem is assessing the type of constraints on ground truth mechanisms, encoded by function classes, that are relevant for real world data. It is plausible that those mechanisms are typically much more regular than generic nonlinear functions. Recently, Gresele et al. [17] suggested, based on arguments from the causality literature that $\mathcal{F}_{\mathrm{OCT}}$ is a natural class for representation learning (and our results show it also has favourable theoretical properties), but this will require experimental confirmation on real world data.

Finally, a central question from a machine learning perspective is the ability to design learning algorithms that can train LVMs with identifiable function class constraints. Interestingly, Gresele et al. [17] showed that OCT maps can be learnt using a closed from regularized likelihood loss, thereby providing, supported by our result, a full-fledged identifiable nonlinear ICA framework.

## Acknowledgments and Disclosure of Funding

We thank Luigi Gresele and Julius von Kügelgen for helpful discussions and feedback on an earlier version of this paper. We thank the anonymous reviewers whose comments helped us to make the paper more accessible. This work was in part supported by the German Federal Ministry of Education and Research (BMBF) through the Tübingen AI Center (FKZ: 01IS18039B) and the Machine Learning Cluster of Excellence, EXC number 2064/1 – Project number 390727645.

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
