## Supplementary Material

In the appendix we provide the proofs of our results and we discuss the relevant background. It is structured as follows. We first introduce some mathematical background in Appendix A and extend the function class definitions to Riemannian manifolds in Appendix B. We discuss a general construction of spurious solutions in Appendix C. Then we provide the proofs of our main results in Sections E to G.

## A    Mathematical background

For the convenience of the reader we collect some mathematical definitions and notations. All definitions can be found in standard textbooks.

**Pushforward of measures.**    For a measure $\mu$ on a (measurable) space $X$ and a measurable map, $f : X \to Y$ the pushforward measure $f_*\mu$ is defined by $(f_*\mu)(A) = \mu(f^{-1}(A))$ for any measurable set $A \subset Y$. Here $f^{-1}(A) = \{x \in X \mid f(x) \in A\}$ denotes the preimage of $A$ under $f$. Sometimes the pushforward measure is denoted by $f^{\#}\mu$. Note that no further restrictions on $f$ are necessary, in particular $f$ does not need to be invertible. One important property that we will use frequently is the relation $(g \circ f)_*\mu = f_*(g_*\mu)$.

Note that if $s \sim \mathbb{P}$, i.e., $s$ has distribution $\mathbb{P}$ then $f(s) \sim f_*\mathbb{P}$. Indeed, this is obvious as $P(f(s) \in A) = P(s \in f^{-1}(A)) = \mathbb{P}(f^{-1}(A))$. In the context of ICA it is convenient to mostly talk about distributions and push-forwards as we typically never observe pairs $(s, f(s))$. For later usage we also recall the transformation formula for random variables. If $f \in C^1(\mathbb{R}^n, \mathbb{R}^n)$ is an invertible diffeomorphism and $\mathbb{Q} = f_*\mathbb{P}$ where $\mathbb{P}$ and $\mathbb{Q}$ have density $p$ and $q$ respectively then

$$q(y) = p(f^{-1}(y))|\operatorname{Det} Df^{-1}(y)|. \tag{13}$$

Note that the potentially more familiar version for random variables reads as follows. Let $X$ and $Y$ be random variables satisfying $Y = f(X)$, then their densities are related by

$$p_Y(y) = p_X(f^{-1}(y))|\operatorname{Det} Df^{-1}(y)|. \tag{14}$$

**Diffeomorphisms.**    Let $U, V \subset \mathbb{R}^d$. A diffeomorphism from $U$ to $V$ is a bijective map $f : U \to V$ such that $f \in C^1(U, V)$ and $f^{-1} \in C^1(V, U)$. Note that it is not sufficient to assume that $f$ is bijective and in $C^1(U, V)$, the classic counterexample being $f(x) = x^3$. A sufficient condition is that $Df(x)$ is an invertible matrix for all $x$. Sometimes we loosely speak about diffeomorphisms $f : U \to \mathbb{R}^d$ which should be understood as $f$ being a diffeomorphism on its image $f(U)$.

**Vector fields and flows**    Vector fields can be introduced nicely in the language of differential geometry. However, we think that for the purpose of this paper it is more appropriate to give a more down to earth discussion focused on $\mathbb{R}^d$. We refer to [35] for a general introduction. A vector field is a map $X : \mathbb{R}^d \to \mathbb{R}^d$. Our reason to consider vector fields is that they can be used to describe smooth transformations of $\mathbb{R}^d$. We will assume that $X$ is Lipschitz continuous. We define the flow of a vector field as a map $\Phi : \mathbb{R} \times \mathbb{R}^d \to \mathbb{R}^d$ such that

$$\Phi_0(x) = x, \qquad \partial_t \Phi_t(x) = X(\Phi_t(x)). \tag{15}$$

Let us remark concerning the notation that it is convenient to put the $t$ argument below, i.e., we write $\Phi_t(x) = \Phi(t, x)$. Moreover, when applying differential operators $D$ they will by default only act on the spatial variable $x$, i.e., $D\Phi_t(x)$ denotes the derivative of the function $x \to \Phi_t(x)$ for a fixed $t$. Note that the solutions of differential equations can exhibit blow-up so $\Phi_t(x)$ might not be defined for all times $t$. However, when we assume that $X$ is bounded the flow exists globally.

It can be shown that if $X$ is $k$ times differentiable then so is $\Phi_t$ and one can conclude that then $\Phi_t$ is a diffeomorphism ($\Phi$ is bijective by the uniqueness of ordinary differential equations). We will

also consider time-dependent vector fields $X : (-T, T) \times \mathbb{R}^d \to \mathbb{R}^d$ where the flows can be defined similarly, replacing $X$ by $X_t$.

We are particularly interested in the action of flows on probability measures, i.e., we consider the measures $(\Phi_t)_* \mathbb{P}$ for some initial measure $\mathbb{P}$. It can be shown that if the density of $\mathbb{P}$ is $p$ then the density $p_t$ of $(\Phi_t)_* \mathbb{P}$ satisfies the continuity equation

$$\partial_t p_t + \mathrm{Div}(p_t X_t) = 0 \quad \text{and} \quad p_0 = p. \tag{16}$$

Here $\mathrm{Div}$ denotes the divergence which is defined by $\mathrm{Div}\, f = \sum_i \partial_i f_i = \mathrm{Tr}\, Df$. One important consequence is that the flow of the vector field preserves the measure, i.e., $(\Phi_t)_* \mathbb{P}$ iff $\mathrm{Div}(pX_t) = 0$ for all $t$. Moreover, the standard Lebesgue measure in $\mathbb{R}^d$ is preserved if $\mathrm{Div}(X_t) = 0$, i.e., divergence free vector fields generate volume preserving flows and vice versa.

**Additional notation.** We at some places use the notation $[n] = \{1, \ldots, n\}$. We also use the $\mathcal{O}$ notation. Recall that $f(x) = \mathcal{O}(g(x))$ as $x \to \infty$ means that there are constants $x_0$ and $C > 0$ such that $f(x) \leq Cg(x)$ for $x \geq x_0$. Recall that we introduced the notation $C_d = (0, 1)^c$ in the main part and we denoted by $\nu$ the uniform measure on $C_d$. We write 'iff' as a shorthand for 'if and only if'.

# B    Function class definitions for general manifolds

Here we extend the definition of the function classes to Riemannian manifolds. Riemannian manifolds are manifolds $M$ equipped with a metric $g$. For complete definitions we again refer to [35]. We denote the differential of a map $f : M \to N$ at $x \in M$ by $(\mathrm{d}f)_s : T_s M \to T_{f(s)} N$. As before, we use the notation $Df(s) \in \mathbb{R}^{d \times d'}$ for the usual derivative of a map $f : \mathbb{R}^d \to \mathbb{R}^{d'}$. The matrix $Df(s)$ is the representation of $(\mathrm{d}f)_s$ in the standard basis. A smooth map $f : M \to N$ between Riemannian manifolds $(M, g)$ and $(N, h)$ is conformal if there is a function $\lambda : M \to \mathbb{R}_+$ such that

$$(f^*h) = \lambda g \tag{17}$$

where $f^*h$ denotes the pullback metric. This means that for $X, Y \in T_x M$

$$h((\mathrm{d}f)_s X, (\mathrm{d}.f)_s Y) = \lambda(s) g(X, Y). \tag{18}$$

Moreover, we observe that the adjoint $(df)_s^*$ satisfies by definition

$$g((\mathrm{d}f)_s^* (\mathrm{d}f)_s X, Y) = h((\mathrm{d}f)_s X, (\mathrm{d}f)_s Y) = \lambda(s) g(X, Y). \tag{19}$$

We conclude that

$$(\mathrm{d}f)_s^* (\mathrm{d}f)_s = \lambda(s) \cdot \mathrm{Id}_{T_s M}. \tag{20}$$

In the case that $M = N = \mathbb{R}^d$ both equipped with the standard metric the definition is seen to be equivalent to Definition 2 given in the main part. When $N$ is a $d$-dimensional submanifold of $\mathbb{R}^{d'}$ with $d' > d$ then we obtain the pointwise condition

$$Df(s)^\top Df(s) = \lambda(s) \mathrm{Id}_{d \times d}, \tag{21}$$

i.e., $Df(s)$ has orthogonal columns with equal norm. Note that the concatenation of conformal maps is conformal and the inverse of conformal diffeomorphisms is again conformal.

Next, we extend the definition of orthogonal coordinate transformations. We remark that orthogonal coordinates are chart dependent so no coordinate free definition can be given. Thus we focus on the case where the domain of the function is $\mathbb{R}^d$ which we equip with the standard metric and we consider the standard orthogonal coordinate vector fields which we denote by $e_i$. Then we call a smooth map $f : \mathbb{R}^d \to M$ for a Riemannian manifold $(M, g)$ an orthogonal coordinate transformation if for $i \neq j$ and all $s \in \mathbb{R}^d$

$$g((\mathrm{d}f)_s e_i, (\mathrm{d}f)_s e_j) = 0. \tag{22}$$

Again, this condition can be equivalently written as

$$\langle e_i (\mathrm{d}f)_s^* (\mathrm{d}f)_s e_j \rangle_{\mathbb{R}^d} = 0 \tag{23}$$

for $i \neq j$. If $M$ is a submanifold of $\mathbb{R}^{d'}$ with the metric induced from the standard Euclidean metric we obtain the natural generalization of Definition 3, i.e.,

$$Df(s)^\top Df(s) \in \mathrm{Diag}(d). \tag{24}$$

An important remark is that orthogonal coordinates do not exist for all manifolds as there are obstructions. Manifolds with this property are called locally diagonalizable. This is closely related to the representation capability of the function class.

# C   Measure preserving transformations and spurious solutions

In this section we review the construction of spurious solutions for the ICA problem. We assume that we consider ICA in the class $(\mathcal{F}, \mathcal{P})$ where $\mathcal{F}$ is a function class and $\mathcal{P}$ a class of probability measures. We are interested in understanding the set of spurious solutions for a pair $(f, \mathbb{P})$, i.e., the set of pairs $(f', \mathbb{P}') \in \mathcal{F} \times \mathcal{P}$ such that $f_* \mathbb{P} = f'_* \mathbb{P}'$. We now note that if we can construct $h$ such that $h_* \mathbb{P}' = \mathbb{P}$ and set $f' = f \circ h$ we have $f_* \mathbb{P} = f'_* \mathbb{P}'$. Next we define the subset of right composable functions

$$\mathcal{F}^R = \{f \in \mathcal{F} \mid g \circ f \in \mathcal{F} \text{ for all } g \in \mathcal{F}\} \tag{25}$$

and similarly the subset of left-composable functions

$$\mathcal{F}^L = \{f \in \mathcal{F} \mid f \circ g \in \mathcal{F} \text{ for all } g \in \mathcal{F}\}. \tag{26}$$

We remark that if $\mathcal{F}$ is a group then obviously $\mathcal{F}^L = \mathcal{F}^R = \mathcal{F}$ and the problem reduces to finding measure preserving transformations in $\mathcal{F}$. The classes $\mathcal{F}_{\mathrm{lin}}$, $\mathcal{F}_{\mathrm{nonlinear}}$, $\mathcal{F}_{\mathrm{conf}}$, and $\mathcal{F}_{\mathrm{volume}}$ are all groups. We will comment on $\mathcal{F}_{\mathrm{OCT}}$ below.

We note that if $h \in \mathcal{F}^R$ satisfies $h_* \mathbb{P}' = \mathbb{P}$ then $f' = f \circ h \in \mathcal{F}$ and $f'_* \mathbb{P}' = f_* \mathbb{P}$ so this gives us a spurious solution. A specific case is given by $\mathbb{P} = \mathbb{P}'$ in which case we are looking for measure preserving transformations (MPTs) $h \in \mathcal{F}^R$. Note that such $h$ for a certain $\mathbb{P}$ allows us to construct spurious solutions for all pairs $(f, \mathbb{P})$ with arbitrary $f$.

This implies that if $(\mathcal{F}, \mathcal{P})$ is identifiable with respect to $\mathcal{S}$ then any $h$ as above satisfies $h \in \mathcal{S}$ (if $\mathbb{P}'$ has full support, and otherwise there is a $h' \in \mathcal{S}$ such that $h$ and $h'$ agree in the support of $\mathbb{P}'$).

We consider an example. Let $\mathbb{P}$ be the distribution of the standard Gaussian and $h = R$ for some $R \in \mathrm{O}(d)$. Then $h \in \mathcal{F}_{\mathrm{lin}}$ and $h_* \mathbb{P} = \mathbb{P}$ as the standard Gaussian is invariant under rotations. However, such a linear measure preserving transformation does not exist for other $\mathbb{P} \in \mathcal{M}_1(\mathbb{R})^{\otimes d}$. This is the reason that Gaussian distributions are excluded for linear ICA.

Next we observe (as we discussed in the main part) that if the function class $\mathcal{F}$ is stable by right composition with arbitrary component-wise transformation we can turn any admissible latent distribution into another one. This means we can fix a reference measure (we will usually use $\nu$) and then we can find (at least under suitable regularity assumptions) for $\mathbb{P} \in \mathcal{M}_1(\mathbb{R})^{\otimes d}$ maps $h^{\mathbb{P} \to \nu}, h^{\nu \to \mathbb{P}} \in \mathcal{F}^R$ such that $h^{\mathbb{P} \to \nu}_* \mathbb{P} = \nu$, $h^{\nu \to \mathbb{P}}_* \nu = \mathbb{P}$. If we can find an MPT $h \in \mathcal{F}^R$ such that $h_* \nu = \nu$ mixing the coordinates we can then find spurious solutions for any pair $(f, \mathbb{P})$ by considering $f' = f \circ h^{\nu \to \mathbb{P}} \circ h \circ h^{\mathbb{P} \to \nu}$ because then $f'_* \mathbb{P} = f_* \mathbb{P}$ and $f' \in \mathcal{F}$ by definition of $\mathcal{F}^R$.

For the class $\mathcal{F}_{\mathrm{nonlinear}}$ such MPTs exist, we gave one construction in Lemma 1. In Appendix F we will sketch the proof of this result and discuss another construction based on divergence free vector fields. For the class $\mathcal{F}_{\mathrm{volume}}$ we cannot arbitrarily transform the input distribution. However, we can still find coordinate mixing MPTs for every $\mathbb{P}$ (with smooth density). This will be proved in Appendix G. These constructions rule out any (meaningful) identifiability result for $\mathcal{F}_{\mathrm{nonlinear}}$ and $\mathcal{F}_{\mathrm{volume}}$.

There is also a slightly different approach to construct spurious solutions for a pair $(f, \mathbb{P})$. For any MPT $h \in \mathcal{F}^L$ such that $h_* f_* \mathbb{P} = f_* \mathbb{P}$, i.e., transformations that preserve the observational distribution $\mathbb{Q} = f_* \mathbb{P}$, the function $h \circ f$ defines a spurious solution. Note that this gives spurious solutions for all pairs $(f, \mathbb{P})$ such that $f_* \mathbb{P}$ follows the fixed distribution $\mathbb{Q}$. This approach will allow us to construct spurious solutions in $\mathcal{F}_{\mathrm{OCT}}$ for some fine-tuned pairs $(f, \mathbb{P})$ with $f \in \mathcal{F}_{\mathrm{OCT}}$ (see the proof of Proposition 1 in Appendix F). Note that because of their fine-tuning, such spurious solutions can arguably be considered pathological cases instead of key non-identifiability issues. This is inline with how such solutions a considered in the case of linear ICA.

Let us finally have a closer look at $\mathcal{F}_{\mathrm{OCT}}$. It is quite straightforward to see that

$$\mathcal{F}_{\mathrm{OCT}}^L = \mathcal{F}_{\mathrm{conf}}. \tag{27}$$

In particular, all rotations are contained in $\mathcal{F}_{\mathrm{conf}}^L$. More interestingly, we have

$$\begin{aligned} \mathcal{F}_{\mathrm{OCT}}^R &= \{f \in \mathcal{F}_{\mathrm{OCT}} \mid Df(x) = P(x)\Lambda(x) \text{ for some } P(x) \in \mathrm{Perm}(d), \Lambda(x) \in \mathrm{Diag}(d)\} \\ &= \{f \in \mathcal{F}_{\mathrm{OCT}} \mid f = P \circ h \text{ where } P \in \mathrm{Perm}_\pm(d) \text{ and } h : \mathbb{R}^d \to \mathbb{R}^d \text{ with} \\ &\qquad h(x) = (h_1(x_1), \ldots, h_d(x_d))^\top \text{ for some } h_i \in C^1(\mathbb{R}, \mathbb{R}) \text{ with } h_i' > 0\}. \end{aligned} \tag{28}$$

The first step can be seen using the chain rule and the definition of $\mathcal{F}_{\text{OCT}}$. The second step follows from the fact that the permutation $P$ is necessarily constant for such a function. This again recovers the fact that OCTs can only be identified up to permutations and coordinate-wise reparametrisations. However, we also conclude that all $h \in \mathcal{F}_{\text{OCT}}^R$ act coordinate-wise (up to a permutation), i.e., do not prevent BSS. This shows that for $\mathcal{F}_{\text{OCT}}$ no completely generic spurious solution based on a single MPT mixing the coordinates exists. This is different from $\mathcal{F}_{\text{nonlinear}}$. We emphasize that this does not rule out the existence of a fine-tuned spurious solution for every (or most) pairs $(f, \mathbb{P})$ because then we only need to find $h$ such that $h_* \mathbb{P} = \mathbb{P}$ and $f \circ h \in \mathcal{F}_{\text{OCT}}$. The relation $f \circ h \in \mathcal{F}_{\text{OCT}}$ of course holds for $h \in \mathcal{F}_{\text{OCT}}^R$ but there will, in general, be many more $h$ for a fixed $f$.

## D Proof of Theorem 1

We here, for completeness, give a proof of Theorem 1. While this result is well known we think that it makes sense to include a condensed proof because it contains many of the key steps of the more involved proof for conformal maps in the next section and it is not as well known as the proof based on Darmois-Skitovich Theorem which does not generalize to nonlinear functions.

*Proof of Theorem 1.* We assume that $x \stackrel{\mathcal{D}}{=} As \stackrel{\mathcal{D}}{=} A's'$. We first assume that the densities $p : \mathbb{R}^n \to \mathbb{R}$ and $q : \mathbb{R}^n \to \mathbb{R}$ of $s$ and $s'$ are $C^2$ functions and $p(x) > 0$ for all $x \in \mathbb{R}^d$. We denote their distributions by $\mathbb{P}$ and $\mathbb{Q}$. By independence of the components we can write $p(x) = \prod_i p_i(x_i)$, $q(x) = \prod_i q_i(x_i)$ for some $C^2$ functions $p_i$ and $q_i$. By assumption we conclude that $((A')^{-1}A)_* \mathbb{P} = \mathbb{Q}$. We denote $B = ((A')^{-1}A)^{-1} = A^{-1}A'$ so that $B_*^{-1}\mathbb{P} = \mathbb{Q}$ and the transformation formula for densities implies that

$$q(y) = p(By)\,|\operatorname{Det} B| \quad \Rightarrow \quad \sum_k \ln(q_k(y_k)) = \sum_k \ln(p_k((By)_k) + \ln|\operatorname{Det} B|. \tag{29}$$

The main idea of the proof is to use the observation that for $i \neq j$ and all $y$ such that $q(y) \neq 0$

$$\partial_i \partial_j \ln(q(y)) = \partial_i \partial_j \sum_k \ln(q_k(y_k)) = 0 \tag{30}$$

i.e., mixed second derivatives of the log density vanish. We plug (29) into this equation and get

$$\partial_i \ln(q(y)) = \sum_k B_{ki} \ln(p_k)'((By)_k), \tag{31}$$

$$\partial_i \partial_j \ln q(y) = \sum_k B_{kj} B_{ki} \ln(p_k)''((By)_k). \tag{32}$$

We now denote $x = By$. Then we get (using that $B$ is invertible) that for all $x$ such that $p(x) \neq 0$

$$0 = \sum_k B_{kj} B_{ki} \ln(p_k)''(x_k). \tag{33}$$

Varying one $x_k$ individually we conclude that each summand is constant which implies that either $B_{kj}B_{ki} = 0$ for all $i \neq j$ or $\ln(p_k)''(x_k)$ is constant. It is straightforward to see that the only probability distribution with $\ln(p)'' = \kappa$ for some constant $\kappa$ are Gaussian distributions. Indeed, note that $\ln(p)'' = \kappa$ for some constant implies that $p(x) = \exp(\kappa x^2/2 + c_1 x + c_2)$. If $p$ is the density of a probability distribution and in particular integrable we see that this implies $\kappa < 0$ and $p$ is a Gaussian density with $\kappa = -\sigma^{-2}$. Note that by assumption at most one component of $\mathbb{P}$ is Gaussian, w.l.o.g., $k = 1$.

As $p_k$ is not a Gaussian for $k > 1$ and thus $\ln(p_k)''$ not constant we conclude

$$B_{kj}B_{ki} = 0 \tag{34}$$

for $i \neq j$ and all $k > 1$. Plugging this into (33) we obtain

$$0 = B_{1j}B_{1i} \ln(p_1)''(x_1). \tag{35}$$

Note that if $p_1$ is a Gaussian density then $\ln(p_1)'' = -\sigma^{-2} \neq 0$ so we conclude that in any case $B_{1i}B_{1j} = 0$ for $i \neq j$, i.e., (34) holds as well for $k = 1$.

In other words, at most one entry of every row of $B$ is non-zero. This implies that $B = P\Lambda$ for some $P \in \mathrm{Perm}(d)$ and $\Lambda \in \mathrm{Diag}(d)$. This was to be shown.

Let us clarify what happens when there is more than one Gaussian component. In this case there might be multiple constant non-zero terms in (33) whose contributions can cancel and we cannot conclude that (34) holds for all $k$. This recovers the well known non-uniqueness for Gaussian variables.

It remains to extend the result to distributions whose density is not twice differentiable. By standardizing $s$ and $s'$ we can assume that $B \in \mathrm{O}(n)$. Indeed, when the covariances of $s$ and $s'$ satisfy $\Sigma_s = \Sigma_{s'} = \mathrm{Id}_d$ then $s = Bs'$ implies $B^\top B = B^T \Sigma_{s'} B = \Sigma_s = \mathrm{Id}$. Then $s' \overset{\mathcal{D}}{=} Bs$ implies that for an independent standard normal

$$B(s + N) \overset{\mathcal{D}}{=} s' + BN \overset{\mathcal{D}}{=} s' + N \tag{36}$$

where we used that standard normal variables are invariant under orthogonal maps. Note that $s + N \in \mathcal{P}_{\mathrm{lin}}$. Indeed, $(s_i, N_i) \perp\!\!\!\perp (s_j, N_j)$ implies $(s_i + N_i) \perp\!\!\!\perp (s_j + N_j)$ for $i \neq j$. If $s_i + N_i$ is Gaussian then $s_i$ is Gaussian (consider, e.g., the Fourier transform) so $s + N$ also has at most one Gaussian component. The density of $s + N$ is smooth and pointwise positive, so we can apply the reasoning above to $s + N$ and $s' + N$ and conclude $B \in \mathcal{S}_{\mathrm{lin}}$. $\qquad\square$

# E  Proofs for the results on conformal maps

In this section we give the proofs for Section 3. First, we consider $d > 2$ and then the special case $d = 2$.

## E.1  Proof of Theorem 2

The proof of Theorem 2 uses similar ideas as the proof for Theorem 1, however, the calculations are more involved. The key ingredient is the classification of all conformal maps in Theorem 7. From there we see that we already dealt with the linear case in Theorem 1, so it is sufficient to focus on the case of nonlinear transformations. Recall that the Moebius transformations introduced in (12) are given by

$$f(x) = b + \alpha \frac{A(x - a)}{|x - a|^\varepsilon} \tag{37}$$

where $b, a \in \mathbb{R}^d$, $\alpha \in \mathbb{R}$, $A \in \mathrm{O}(d)$, and $\varepsilon \in \{0, 2\}$. In particular, the Theorem will be an easy consequence of the following lemma.

**Lemma 2.** *Suppose $g : \mathbb{R}^d \to \mathbb{R}^d$ is a nonlinear Moebius transformation, i.e., a map as in (37) with $\varepsilon = 2$. Let $s$, $s'$ be random variables whose distributions are in $\mathcal{P}_{\mathrm{conf}}$. Then $s \overset{\mathcal{D}}{\neq} g(s')$.*

Let us quickly show how it implies Theorem 2 before we prove this lemma.

*Proof of Theorem 2.* We use the same notation as in the proof of Theorem 1. We denote the distribution of $s$ and $s'$ by $\mathbb{P}$ and $\mathbb{Q}$ and we assume $x \overset{\mathcal{D}}{=} f(s) \overset{\mathcal{D}}{=} f'(s')$ with $f, f' : \mathbb{R}^d \to M$ conformal (see Appendix B for the definition). This implies that $(f'^{-1}f)_* \mathbb{P} = \mathbb{Q}$. We denote $g = ((f')^{-1}f)^{-1} = f^{-1}f'$ so that $\mathbb{P} = g_*\mathbb{Q}$. Now we make the simple but important observation that $g$ as a concatenation of a conformal map and the inverse of a conformal map is a conformal map from $\mathbb{R}^d$ to $\mathbb{R}^d$. Thus, we can apply Liouville's Theorem to $g$ (recalled in Theorem 7) which implies that

$$g(y) = b + \frac{\alpha A(y - a)}{|y - a|^\varepsilon} \tag{38}$$

where $b, a \in \mathbb{R}^d$, $\alpha \in \mathbb{R}$, $A \in \mathrm{O}(d)$, $\varepsilon \in \{0, 2\}$. Using Lemma 2 we conclude that $\varepsilon = 0$ and $g$ is linear. Then we can apply Theorem 1 (using that $\mathcal{P}_{\mathrm{conf}} \subset \mathcal{P}_{\mathrm{lin}}$) and conclude that $\alpha A = P\Lambda$ for a permutation matrix and a diagonal matrix $\Lambda$. Since $g$ is conformal we have that $A$ is orthogonal and all eigenvalues have absolute value 1 which implies that $\Lambda_{ii} = \pm\alpha$ for $1 \leq i \leq d$ and thus $A \in M_{\mathrm{perm},\pm}(\mathbb{R}^{d \times d})$. $\qquad\square$

To prove Lemma 2 we need one technical result that shows that local properties of the density $p_i$ (i.e., properties that hold for $x_i \in O_i$ for some non-empty open sets $O_i$) in fact hold for all $x_i \neq 0$. This will be based on the nonlinearity of the map $g$ combined with the factorization $p(x) = \prod_i p_i(x_i)$ of the densities. An illustration can be found below in Figure 4.

**Lemma 3.** *Let $O = O_1 \times \ldots \times O_d \subset \mathbb{R}^d$ and $U = U_1 \times \ldots \times U_d \subset \mathbb{R}^d$ where $O_i$ and $U_i$ are non-empty open sets. Let $g(x) = Ax/|x|^2$ for an orthogonal matrix $A$. Assume that $g(O) = U$. Then $U_i$ is either $(0, \infty)$, $(-\infty, 0)$ $(-\infty, 0) \cup (0, \infty)$, or $(-\infty, \infty)$. Moreover, if the $i$-th row of $A$ is not equal to a (signed) standard basis vector then $U_i = \mathbb{R}$.*

Informally the result follows from the fact that coordinate planes $\{x_i = c\}$ are mapped to spheres by $g$ except for $c = 0$. We assume that the boundary of $O$ and $U$ is the union of subsets of hyperplanes. However, $g(O) = U$ then implies that the boundaries of $O$ and $U$ are mapped to each other. Since the boundaries of both sets are the union of subsets of hyperplanes that are mapped to hyperplanes we conclude their boundaries must be a subset of the coordinate axes hyperplanes $H_i = \{x : x_i = 0\}$ (because they would be mapped to spherical caps by $g$). For completeness, we give a careful proof below. Let us highlight that this lemma essentially rules out counterexamples with finite support because we can apply the lemma to the set $\{x \in \mathbb{R}^d : p(x) > 0\}$. This is in contrast to the 2-dimensional case where conformal maps between any two rectangles exist making the proof of the corresponding statement below more difficult.

We now prove Lemma 2.

*Proof of Lemma 2.* The proof is a bit lengthy, so we first give an informal overview of the main steps. As before, we denote the distribution of $s$ and $s'$ by $\mathbb{P}$ and $\mathbb{Q}$. We argue by contradiction, so we assume that $\mathbb{P} = g_*\mathbb{Q}$. By assumption $g(x) = b + \alpha A(x - a)/|x - a|^2$. We now proceed in several steps that constrain the structure of $g$. Let us briefly describe the steps of the proof.

In Step 1 and 2 we eliminate the trivial symmetries of the measure and show that the mild local regularity assumption on the measures imply global regularity.

Then, in Steps 3 and 4 we derive in (48) a condition similar to (33) but more involved.

To exploit this condition we look in Steps 5 and 6 at certain limiting regimes where the terms become much simpler and almost reduce to the condition of the linear case. This allows us to conclude that $A$ is a permutation matrix in Step 7.

Using that $A$ is a permutation matrix in (48) we get in Step 8 a much simpler relation that almost factorizes. This allows us to derive a simple differential equation in Step 9 which restricts the potential densities to a simple parametric form. This allows us to conclude.

**Step 1: Elimination of trivial symmetries.** First we show that we can assume $a = b = 0$ and $\alpha = 1$. We denote the shifts on $\mathbb{R}^d$ by $T_a(x) = x + a$ and the dilations $D_\alpha(x) = \alpha x$. Then we can rewrite $g = T_b \circ D_\alpha \circ g_0 \circ T_{-a}$ with $g_0(x) = Ax/|x|^2$ and therefore $(g_0)_*(T_{-a})_*\mathbb{Q} = (D_{\alpha^{-1}} \circ T_{-b})_*\mathbb{P}$. Since shifts and dilations preserve the class $\mathcal{P}_{\mathrm{conf}}$ it is sufficient to show the result for $g = g_0$. To simplify the notation we drop the 0 in the following and just assume $g(x) = Ax/|x|^2$.

**Step 2: Support and smoothness of distributions.** The goal of this step is to show that under the assumption of Lemma 2 the density of $\mathbb{P}$ and similarly of $\mathbb{Q}$ is positive and $C^2$ away from the coordinate hyperplanes $\{x_i = 0\}$ for a union of quadrants, while it vanishes on the remaining quadrants. This will be a consequence of Lemma 3. Let $U_i = \mathrm{Int}(\mathrm{supp}\,\mathbb{P}_i)$. By assumption, $\mathbb{P} \in \mathcal{P}_{\mathrm{conf}}$ which entails $U_i \neq \emptyset$. Then $U = U_1 \times \ldots \times U_n = \mathrm{Int}(\mathrm{supp}\,\mathbb{P})$. Define $O_i$ similarly for $\mathbb{Q}$. The relation $g_*\mathbb{Q} = \mathbb{P}$ implies $g(O) = U$. Applying Lemma 3 we conclude that $U$ is the union of quadrants.

The same argument will imply that $p$ is actually $C^2$ away from the coordinate planes. We consider the interior of the set of points where $\mathbb{P}$ has a twice differentiable and positive density and call it $U'$. For $x \in U'$ there is a density $p$ in a neighbourhood of $x$ and it factorizes by the independence assumption. The relation

$$\partial_i^2 p(x)/p(x) = \partial_i^2 p_i(x_i)/p_i(x_i) \tag{39}$$

implies that then $p_i$ is twice differentiable at $x_i$. Vice-versa, if all $p_i$ are twice differentiable at $x_i$ then $p$ is twice differentiable at $x = (x_1, \ldots, x_d)$. This implies that $U' = U'_1 \times \ldots \times U'_d$ for some

open sets $U_i'$. By definition of $\mathcal{P}_{\text{conf}}$ the sets $U_i'$ are non-empty. Similarly, we define $O'$. The relation (14) for the densities $p$ and $q$ implies, together with the smoothness of $g$, that $g(O') = U'$. Then we apply again Lemma 3 to conclude that $p$ and $q$ are $C^2$ functions away from the hyperplanes $\{x_i = 0\}$ and if the density is non-zero at a point in a quadrant then it is positive in the complete interior of the quadrant.

Finally, $U_i' = \mathbb{R}$ if the $i$-th row of $A$ has more than one non-zero entry (again by Lemma 3). By definition this means that $p_i \in C^2(\mathbb{R})$ and $p_i(x) > 0$ for such $i$ and all $x$.

Let us emphasize here, that we already finished the proof of Lemma 2 for probability distributions with bounded support. This is more difficult in dimension 2 because there are two-dimensional conformal functions mapping rectangles to rectangles. We will consider this in the proof of Theorem 3 below.

A major step in the proof is to show that $A$ is a permutation matrix under the assumptions of the lemma. We define the index set $I \subset [d]$ as the set of all indices $i$ such that the $i$-th row of the matrix $A$ has only one non-zero entry. Our goal is to show that $I = [d]$. We have shown so far that $p_k$ is positive and twice differentiable if $k \notin I$.

The proof in the linear case relied on the relation (33). We now derive a similar relation for non-linear Moebius transformations.

**Step 3: Derivative formulas.** For future reference we note (using $A \in \mathrm{O}(n)$) that (for $i \neq j$)

$$(Dg(y))_{kj} = \partial_j g_k(y) = \left( \frac{A}{|y|^2} - 2\frac{Ay \otimes y}{|y|^4} \right)_{kj} = \frac{A_{kj}}{|y|^2} - 2\frac{(Ay)_k y_j}{|y|^4}, \tag{40}$$

$$(\partial_i \partial_j g_k)(y) = -2\frac{A_{kj} y_i + A_{ki} y_j}{|y|^4} + \frac{8(Ay)_k y_i y_j}{|y|^6} \tag{41}$$

$$\mathrm{Det}(Dg(y)) = \mathrm{Det}\, \frac{A}{|y|^2}\, \mathrm{Det}\left( \mathrm{Id} - 2\frac{y \otimes y}{|y|^2} \right) = -|y|^{-2d}. \tag{42}$$

**Step 4: Derivation of a condition for the densities** We apply the same reasoning as in the proof of Theorem 1 to derive partial differential equations for the density $p$. The condition $s = g(s')$, or equivalently $g^{-1}(s) = s'$ and the density relation (14) imply

$$q(y) = p(g(y))|\mathrm{Det}\,\nabla g(y)|. \tag{43}$$

This implies

$$q(y) = p\left(g(y)\right) | \mathrm{Det}\,\nabla(Ay|y|^{-2})| = p\left(g(y)\right)|y|^{-2d} \tag{44}$$

$$\Rightarrow \sum_k \ln(q_k(y_k)) = \sum_k \ln(p_k(g_k(y))) - 2d\ln(|y|). \tag{45}$$

We calculate for $i \neq j$

$$\partial_i \ln(q(y)) = \sum_k (\partial_i g_k)(y)(\ln p_k)'(g(y)) - 2d\partial_i \ln(|y|),$$

$$0 = \partial_j \partial_i \ln(q(y)) = -2d\partial_j\partial_i \ln(|y|) + \sum_k (\partial_i\partial_j g_k)(y)(\ln p_k)'(g_k(y)) \tag{46}$$

$$+ \sum_k (\partial_i g_k)(y)(\partial_j g_k)(y)(\ln p_k)''(g_k(y)).$$

Evaluating the derivatives using (40) and (41) we get

$$0 = 4d\frac{y_i y_j}{|y|^4}$$

$$+ \sum_k \left( \frac{8(Ay)_k y_i y_j}{|y|^6} - 2\frac{A_{kj} y_i + A_{ki} y_j}{|y|^4} \right) \ln(p_k)'(g_k(y)) \tag{47}$$

$$+ \sum_k \left( \frac{A_{kj}}{|y|^2} - 2\frac{(Ay)_k y_j}{|y|^4} \right)\left( \frac{A_{ki}}{|y|^2} - 2\frac{(Ay)_k y_i}{|y|^4} \right) \ln(p_k)''(g_k(y)).$$

Finally we express the variable $y$ through $x = g(y) = Ay|y|^{-2}$. Note that then $|y| = |x|^{-1}$ and $y = A^{-1}x|x|^{-2}$. Plugging this in the last equation we get

$$
\begin{aligned}
0 &= 4d(A^{-1}x)_i(A^{-1}x)_j \\
&\quad + \sum_k \left(8x_k(A^{-1}x)_i(A^{-1}x)_j - 2|x|^2 \left(A_{kj}(A^{-1}x)_i + A_{ki}(A^{-1}x)_j\right)\right)\ln(p_k)' \\
&\quad + \sum_k \left(A_{kj}|x|^2 - 2x_k(A^{-1}x)_j\right)\left(A_{ki}|x|^2 - 2x_k(A^{-1}x)_i\right)\ln(p_k)'' \\
&= 4dA_{mi}x_m A_{lj}x_l \\
&\quad + \sum_k \left(8x_k A_{mi}x_m A_{lj}x_l - 2|x|^2 \left(A_{kj}A_{mi}x_m + A_{ki}A_{lj}x_l\right)\right)\ln(p_k)' \\
&\quad + \sum_k \left(A_{kj}|x|^2 - 2x_k A_{lj}x_l\right)\left(A_{ki}|x|^2 - 2x_k A_{mi}x_m\right)\ln(p_k)''
\end{aligned}
\tag{48}
$$

where we used $A^{-1} = A^\top$ as $A$ is orthogonal and we used Einstein summation convention to sum over indices that appear twice (we kept the sum over $k$ for better readability). Note that this expression is not homogeneous in $x$.

**Step 5: Simplifications as $x_r \to \infty$.** We fix an index $1 \le r \le d$. The strategy is now to send $x_r \to \infty$ while keeping the other coordinates bounded. We can assume by reflecting coordinates that the quadrant $\{x_i > 0, \forall i\}$ is contained in the support of $\mathbb{P}$ and $p$ has a positive $C^2$ density there. We can then rewrite (48)

$$
\begin{aligned}
0 &= \mathcal{O}(x_r^2) + (8x_r^3 A_{ri}A_{rj} - 4x_r^3 A_{rj}A_{ri})\ln(p_r)' + \mathcal{O}(x_r^2\ln(p_r)') + \mathcal{O}(x_r^3) \\
&\quad + (A_{rj}x_r^2 - 2x_r A_{rj}x_r)(A_{ri}x_r^2 - 2x_r A_{ri}x_r)\ln(p_r)'' + \mathcal{O}(x_r^3\ln(p_r)'') \\
&\quad + \sum_{k \ne r} A_{kj}A_{ki}x_r^4\ln(p_k)'' + \mathcal{O}(x_r^3) \\
&= x_r^4 \sum_k A_{kj}A_{ki}\ln(p_k)'' + 4x_r^3\ln(p_r)' + \mathcal{O}\left(x_r^3(1 + \ln(p_r)'') + x_r^2\ln(p_r)'\right).
\end{aligned}
\tag{49}
$$

We conclude that

$$
0 = \sum_k A_{kj}A_{ki}\ln(p_k)'' + \frac{4A_{ri}A_{rj}\ln(p_r)'}{x_r} + \mathcal{O}\left(\frac{1 + \ln(p_r)''}{x_r} + \frac{\ln(p_r)'}{x_r^2}\right).
\tag{50}
$$

By varying $x_k \ne x_r$ this almost implies that $A_{kj}A_{ki}\ln(p_k)'' = c_k$ for some constant whenever $p_k > 0$ and twice differentiable. However, for this we need to show that the terms hidden in $\mathcal{O}(\cdot)$ are really negligible, i.e. $\ln(p_r)''$ and $\ln(p_r)'/x_r$ are bounded as $x_r \to \infty$ so that the remainder becomes $o(1)$ which we will establish.

Note that if such a relation could be derived we could conclude, similarly to the linear case, that $A$ is a permutation matrix.

**Step 6: Boundedness of $\ln(p_r)''$ and $\ln(p_r)'/x_r$.** Recall that $I$ is the set of indices such that the $i$-th row of $A$ has only one non-zero entry. Let $r \notin I$ and pick $j, i$ such that $A_{rj}A_{ri} \ne 0$. Fix all coordinates $x_k$ except $x_r$ so that $p_k$ is positive and twice differentiable at $x_k$. Then we can express (50) as

$$
\ln(p_r)'' + \frac{4\ln(p_r)'}{x_r} = R(x_r)
\tag{51}
$$

where the remainder term $R$ contains the remaining terms. The expression $R$ of course depends on the other coordinates $x_k$ for $k \ne r$ but since they are considered fixed here we can view $R$ as a function of $x_r$ alone.

Equation (50) then implies that there is $M > 0$ sufficiently large such that for $x_r > M$ the remainder term $R(x)$ satisfies for some constant $c > 0$.

$$
|R(x)| \le \frac{1}{2}|\ln(p_r)''| + \frac{|\ln(p_r)'|}{x_r} + c.
\tag{52}
$$

Here the last constant term bounds the $x_r^{-1}$ contribution. Suppose $\ln(p_r)' \leq 0$. Then we can bound

$$
\begin{aligned}
0 &= \ln(p_r)''(x_r) + \frac{4\ln(p_r)'(x_r)}{x_r} - R(x_r) \\
&\leq \ln(p_r)''(x_r) + \frac{1}{2}|\ln(p_r)''(x_r)| + \frac{4\ln(p_r)'(x_r)}{x_r} + \frac{|\ln(p_r)'(x_r)|}{x_r} + c \\
&\leq \ln(p_r)''(x_r) + \frac{1}{2}|\ln(p_r)''(x_r)| + c
\end{aligned}
\tag{53}
$$

We find that $\ln(p_r)'' \geq -2c$ (for $\ln(p_r)'' > 0$ this is clear, and otherwise we can absorb the absolute value part). We conclude by integration that for all $x_r > M$ (note that the bound is trivially true if $\ln(p_r)' > 0$)

$$
\ln(p_r)'(x_r) \geq \min(0, \ln(p_r)'(M)) - 2c(x_r - M).
\tag{54}
$$

Similarly we can bound for $\ln(p_r)'(x_r) \geq 0$

$$
\begin{aligned}
0 &= \ln(p_r)''(x_r) + \frac{4\ln(p_r)'(x_r)}{x_r} - R(x_r) \\
&\geq \ln(p_r)''(x_r) - \frac{1}{2}|\ln(p_r)''(x_r)| + \frac{4\ln(p_r)'(x_r)}{x_r} - \frac{|\ln(p_r)'(x_r)|}{x_r} - c \\
&\geq \ln(p_r)''(x_r) - \frac{1}{2}|\ln(p_r)''(x_r)| - c
\end{aligned}
\tag{55}
$$

implying $\ln(p_r)''(x_r) \leq 2c$ for $x_r \geq M$ such that $\ln(p_r)'(x_r) > 0$. We obtain

$$
\ln(p_r)'(x_r) \leq \max(0, \ln(p_r)'(M)) + 2c(x_r - M).
\tag{56}
$$

Together the last two steps imply that $|\ln(p_r)'(x_r)| \leq C + Cx_r$ for some $C > 0$ and $x_r > M$. Going back to (51) we conclude that there is $C > 0$ such that $|\ln(p_r)''(x_r)| \leq C$ for $x_r > M$. We conclude that for $r \notin I$ and all $i \neq j$

$$
0 = \sum_k A_{kj}A_{ki}\ln(p_k)'' + \frac{4A_{ri}A_{rj}\ln(p_r)'}{x_r} + O\left(x_r^{-1}\right).
\tag{57}
$$

**Step 7: $A$ is a permutation matrix.** If $I = [n]$ we are done. So there is $r \notin I$ and using (57) we conclude by varying $x_k \neq x_r$ that

$$
A_{kj}A_{ki}\ln(p_k)'' = c
\tag{58}
$$

for some constant (depending on $k$, $i$, and $j$). Note that if we assumed that at most one $p_k$ is a Gaussian density we could conclude as in the linear case. However, this assumption is not necessary, as we will now show.

By definition, $A_{kj}A_{ki}\ln(p_k)'' = 0$ for $k \in I$ because there is only one non-zero entry in row $k$ of $A$. We have seen in Step 2 that for $k \notin I$ the density $p_k \in C^2(\mathbb{R})$ and is positive. By assumption, we can find $i \neq j$ such that $A_{kj}A_{ki} \neq 0$ for $k \notin I$. The relation (58) then implies for $k \notin I$ that $\ln(p_k)''(x_k) = \beta_k$ for some constant $\beta_k < 0$ ($p_k$ is a probability density) and all $x_k \in \mathbb{R}$. Then $\ln(p_k)'(x_k) = \beta_k x_k + \gamma_k$ for some constant $\gamma_k$. With $x_r \to \infty$ we conclude from (57) that for $r \notin I$

$$
0 = \sum_{k \notin I} A_{kj}A_{ki}\beta_k + 4A_{ri}A_{rj}\beta_r.
\tag{59}
$$

Summing this over $r \notin I$ we get

$$
0 = \sum_{r \notin I}\left(\sum_{k \notin I} A_{kj}A_{ki}\beta_k + 4A_{ri}A_{rj}\beta_r\right) = (d - |I| + 4)\sum_{k \notin I} A_{kj}A_{ki}\beta_k.
\tag{60}
$$

Dividing (60) by $d - |I| + 4$ and subtracting it from (59) we conclude that

$$
A_{ri}A_{rj}\beta_r = 0
\tag{61}
$$

for all $r \notin I$ and all $i \neq j$. Since $\beta_r$ is non-zero this implies that $A_{ri}A_{rj} = 0$ for $i \neq j$ and thus $r \in I$, a contradiction. This establishes $I = [d]$ and thus $A$ is a signed permutation matrix. By permuting and reflecting the coordinates of $\mathbb{P}$ we can assume $A = \mathrm{Id}$ in the following.

**Step 8: Simplifications in** (48) **for** $A = \mathrm{Id}$. First we remark that for $A = \mathrm{Id}$ the function $g$ leaves the quadrants invariant. It is thus sufficient to consider the case where $p$ and $q$ vanish outside $\{x_i > 0 \ \forall i\}$ and show that no solutions of $s = g(s')$ exist under this condition. Using step 2 we can assume that $p_i(x_i) > 0$ for all $x_i > 0$. In the following all domain are assumed to be the positive half-line. For $A = \mathrm{Id}$ the condition (48) becomes for $x = (x_1, \ldots, x_d)$ such that $x_i > 0$

$$
\begin{aligned}
0 = {} & 4d x_i x_j - 2|x|^2 x_i \ln(p_j)' - 2|x|^2 x_j \ln(p_i)' - 2|x|^2 x_j x_i (\ln(p_i)'' + \ln(p_j)'') \\
& + \sum_k 8 x_k x_i x_j \ln(p_k)' + \sum_k 4 x_k^2 x_i x_j \ln(p_k)''.
\end{aligned}
\tag{62}
$$

Dividing this by $2 x_i x_j$ we obtain

$$
0 = 2d - |x|^2 \left( \frac{\ln(p_j)'}{x_j} + \ln(p_j)'' + \frac{\ln(p_i)'}{x_i} + \ln(p_i)'' \right) + \sum_k 4 x_k \ln(p_k)' + 2 x_k^2 \ln(p_k)''.
\tag{63}
$$

We now assume that $d > 2$. Let $i$, $j$, and $r$ be pairwise different. Using the last display with $i$, $j$ and $j$, $r$ and subtracting the resulting equations we obtain

$$
0 = |x|^2 \left( \frac{\ln(p_i)'}{x_i} + \ln(p_i)'' - \frac{\ln(p_r)'}{x_r} - \ln(p_r)'' \right).
\tag{64}
$$

Varying $x_r$ and $x_i$ independently and since $i \in [n]$ is arbitrary we conclude that there is a constant $\kappa$ such that

$$
\frac{\ln(p_i)'}{x_i} + \ln(p_i)'' = \kappa
\tag{65}
$$

for all $i$ and $x_i > 0$.

**Step 9: Conclusion for** $n > 2$. The solutions of the ODE $y(x)/x + y'(x) = \kappa$ are given by

$$
\frac{\alpha}{x} + \frac{\kappa x}{2}
\tag{66}
$$

where $\alpha$ is any constant. We conclude that there are constants $\alpha_j$ such that

$$
\ln(p_j)(x) = \alpha_j \ln(x) - \kappa \frac{x^2}{4} + c \Rightarrow p_j(x) \propto x^{\alpha_j} e^{-\frac{\kappa x^2}{4}}.
\tag{67}
$$

This implies that

$$
q(y) = p(y/|y|^2)|y|^{-2d} \propto \prod_j \frac{y_j^{\alpha_j}}{|y|^{2\alpha_j}} e^{-\frac{\kappa y_j^2}{4|y|^4}} |y|^{-2d}.
\tag{68}
$$

By applying the main argument to $q$ we infer that $q_j$ has to have again the same structure as in (67) so we conclude that $\kappa = 0$ and $\sum_j \alpha_j = -d$. Alternatively, one directly sees that $q$ only factorizes as $q(y) = \prod q_i(y_i)$ if those conditions hold. It is easy to see that those densities satisfy the assumptions. However, $x^\alpha$ is never integrable so there are no probability distributions satisfying the relations $s = g(s')$ This ends the proof for $d > 2$.

**Step 10: Conclusion for** $d = 2$. For $n = 2$ we cannot simplify (63) by considering indices $i \neq j \neq k$. Instead, we directly exploit (63) to obtain a similar conclusion. Similarly to the argument in Step 6 it can be shown that $\ln(p_r)''$ and $\ln(p_r)'/x_r$ are bounded for $x_r$ away from 0. Then we consider $x_1 \to \infty$ in (63) and divide by $x_1^2$. We get (using $\{i, j\} = \{1, 2\}$)

$$
0 = \left( \frac{\ln(p_1)'}{x_1} + \ln(p_1)'' + \frac{\ln(p_2)'}{x_2} + \ln(p_2)'' \right) + 4 \frac{\ln(p_1)'}{x_1} + 2 \ln(p_1)'' + O(x_1^{-1}).
\tag{69}
$$

By varying $x_2$ we conclude just as for $d > 2$ that $\frac{\ln(p_2)'}{x_2} + \ln(p_2)''$ is constant. We conclude as before. $\qquad\square$

Let us now prove the geometric result from Lemma 3 above.

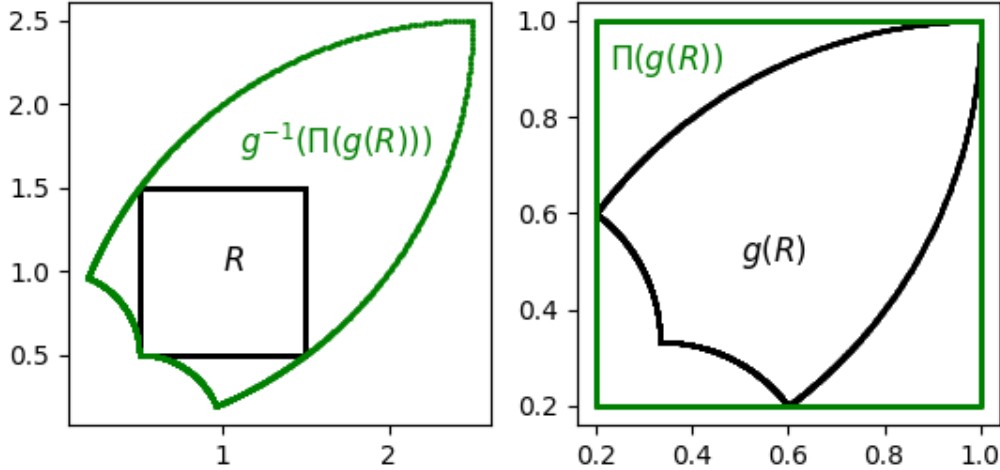

Figure 4: The black rectangle $R$ on the left is mapped by a conformal map to the black shape $g(R)$ on the right. When mapping the smallest rectangle $\Pi(g(R))$ containing $g(R)$ (green rectangle on the right) back to $g^{-1}(\Pi(g(R)))$ (green shape on the left) we obtain a larger set.

*Proof.* The main idea of the proof is that a box contained in $U$ after inversion is distorted so that its convex hull (contained in $O$) is strictly bigger than the box image so that inverting backwards gives us a bigger box in $U$ except for some special cases. An illustration of this argument is shown in Figure 4. The formal argument below is slightly technical. An illustration of the actual argument can be found in Figure 5.

To simplify the notation we write $\iota$ for the inversions $x \to x/|x|^2$. Then we get $g = A \circ \iota = \iota \circ A$. We consider the projections $\pi_i : \mathbb{R}^d \to \mathbb{R}$ projecting on the $i$-th coordinate. We consider a map $\Pi$ on subsets of $\mathbb{R}^d$ defined by $\Pi(M) = \pi_1(M) \times \ldots \times \pi_d(M)$. Let $\mathcal{C}$ denote the convex hull of a set. Let $R \subset O$ be a (connected) box. Then $g(R) \subset U$ implies $\Pi(g(R)) \subset \Pi(U) = U$. Since $g(R)$ is connected $\Pi(g(R))$ is convex and thus $\mathcal{C}(g(R)) \subset \Pi(g(R))$. We conclude that $g^{-1}(\mathcal{C}(g(R)) \subset g^{-1}(\Pi(g(R)) \subset g^{-1}(U) \subset O$. As $A$ is linear we have $A^{-1}\mathcal{C}A(M) = \mathcal{C}(M)$ for any set $M \subset \mathbb{R}^d$. Thus, we get $g^{-1}(\mathcal{C}(g(R)) = \iota \mathcal{C} \iota(R)$.

W.l.o.g. we now suppose that there is a box $R = (x_1, y_1) \times \ldots (x_d, y_d) \subset O$ with $0 < x_i < y_i$. We write $x' = (x_2, \ldots, x_d)$, $y' = (y_2, \ldots, y_d)$. We consider the point

$$z = \frac{1}{2}\iota((y_1, x')^\top) + \frac{1}{2}\iota((y_1, y')^\top). \tag{70}$$

Clearly $z \in \mathcal{C} \iota R$. Then

$$\iota(z) = \iota\left(\frac{1}{2}\iota((y_1, x')^\top) + \frac{1}{2}\iota((y_1, y')^\top)\right) \in \iota \mathcal{C} \iota(R) \tag{71}$$

We calculate

$$\iota\left(\frac{1}{2}\iota((y_1, x')^\top) + \frac{1}{2}\iota((y_1, y')^\top))\right) = \iota\left(\frac{1}{2}\frac{(y_1, x')^\top}{y_1^2 + x'^2} + \frac{1}{2}\frac{(y_1, y')^\top}{y_1^2 + y'^2}\right)$$

$$= \frac{\frac{1}{2}\frac{(y_1, x')^\top}{y_1^2 + x'^2} + \frac{1}{2}\frac{(y_1, y')^\top}{y_1^2 + y'^2}}{\left\|\frac{1}{2}\frac{(y_1, x')^\top}{y_1^2 + x'^2} + \frac{1}{2}\frac{(y_1, y')^\top}{y_1^2 + y'^2}\right\|^2} \tag{72}$$

We bound (using $x'^2 < y'^2$)

$$\left\|\frac{1}{2}\frac{(y_1, x')^\top}{y_1^2 + x'^2} + \frac{1}{2}\frac{(y_1, y')^\top}{y_1^2 + y'^2}\right\|^2 < \frac{1}{2}\left\|\frac{(y_1, x')^\top}{y_1^2 + x'^2}\right\|^2 + \frac{1}{2}\left\|\frac{(y_1, y')^\top}{y_1^2 + y'^2}\right\|^2$$

$$= \frac{1}{2}\frac{1}{y_1^2 + x'^2} + \frac{1}{2}\frac{1}{y_1^2 + y'^2}. \tag{73}$$

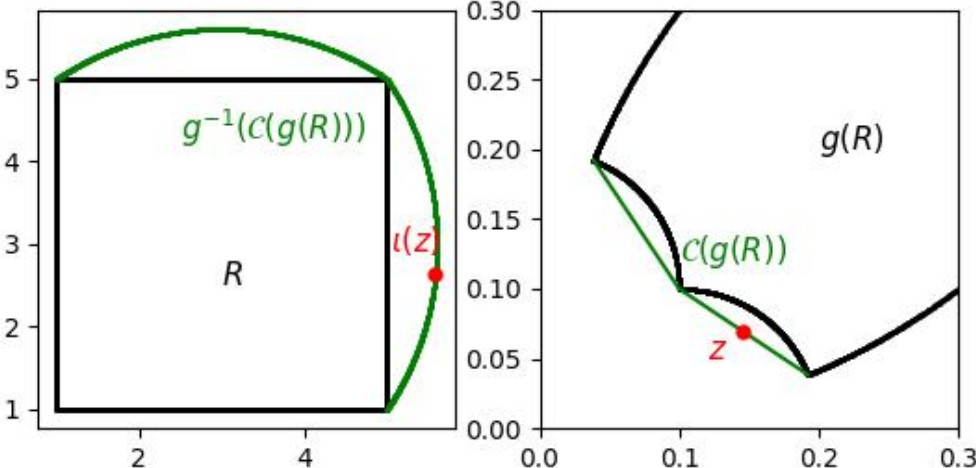

Figure 5: The black rectangle $R$ on the left is mapped by $\iota$ to the black shape $g(R)$ on the right. The green shape on the right shows the convex hull $\mathcal{C}(g(R))$. The point $z$ is defined as in the text and its image $\iota(z)$ lies outside $R$.

Together the last displays imply that

$$\iota\left(\frac{1}{2}\iota((y_1, x')^\top) + \frac{1}{2}\iota((y_1, y')^\top))\right)_1 > y_1. \tag{74}$$

Let $z_1$ be maximal such that $(x_1, z_1) \subset O_1$. Then the reasoning above shows that $z_1 = \infty$. The same reasoning for the other coordinates implies that $R' = (x_1, \infty) \times \ldots \times (x_d, \infty) \subset O$. By applying the same reasoning to sequences of boxes in $g(R)$ approaching the origin we conclude that $U$ is the union of quadrants (and the same holds for $O$).

It remains to prove the last remark. As quadrants are invariant under $\iota$ we have $\iota(O) = O$ and conclude $AO = U$, or equivalently $A^\top U = O$. It is sufficient to show that $0 \in U_i$. For simplicity we assume $R = (0, \infty)^d \subset U$, the generalization to other quadrants is immediate. Since we assume that the $i$-th row $v_i = A^\top e_i$ of $A$ is not equal to a signed standard basis vector it has at least two non-zero entries. Thus we can find $w$ such that $w \cdot v_i = 0$ and all entries of $w$ are non-zero. Since $w$ is orthogonal to the span of $Ae_i$ there is a vector $\alpha$ such that $A^\top \alpha = w$ and $\alpha_i = 0$. By adding a suitable vector $\beta$ we can ensure that $(\beta + \alpha)_i = 0$, all entries of $A^\top(\alpha + \beta)$ are non-zero and $(\beta + \alpha)_j > 0$ for $j \neq i$. The second condition can be satisfied by picking the entries of $\beta$ one after another. The conditions $(\beta + \alpha)_j > 0$ for $j \neq i$ and $(\beta + \alpha)_i = 0$ imply that $\beta + \alpha \in \overline{U}$ since we assumed $(0, \infty)^d \subset U$. But then $A^\top(\alpha + \beta) \in \overline{O}$ and since $A^\top(\alpha + \beta)$ is strictly contained in a quadrant (all entries are non-zero) we conclude that $A^\top(\alpha + \beta) \in O$ and thus $\alpha + \beta \in U$. This implies $(\alpha + \beta)_i = 0 \in U_i$.

$\square$

### E.2 Proof of Corollary 1

Here we prove the simple extension of Theorem 2 to rescaled conformal maps.

*Proof of Corollary 1.* Suppose that there are $f, f' \in \mathcal{F}_{r-\text{conf}}$ and $\mathbb{P}, \mathbb{P}' \in \mathcal{P}_{\text{conf}}$ such that

$$f_*\mathbb{P} \overset{\mathcal{D}}{=} f'_*\mathbb{P}'. \tag{75}$$

Then we need to show that $f^{-1}f' \in \mathcal{S}_{\text{reparam}}$. By definition of $\mathcal{F}_{r-\text{conf}}$ there are $g, g' \in \mathcal{F}_{\text{conf}}$ and $h, h' \in \mathcal{S}_{\text{reparam}}$ such that $f = g \circ h$, $f' = g' \circ h'$. The smoothness condition for $\mathcal{F}_{r-\text{conf}}$ implies $h, h'$ are three times differentiable and thus $\mathbb{Q} = h_*\mathbb{P}$ and $\mathbb{Q}' = h'_*\mathbb{P}'$ satisfy the smoothness

condition of measures in $\mathcal{P}_{\text{conf}}$. From (75) we infer $g_*\mathbb{Q} \stackrel{\mathcal{D}}{=} g'_*\mathbb{Q}'$ and this is the observational distribution. By the assumption on the observational distribution, we conclude that $\mathbb{Q} \sim g^{-1}(x)$ and $\mathbb{Q}' \sim g'^{-1}(x)$ have at most one Gaussian component and thus $\mathbb{Q}, \mathbb{Q}' \in \mathcal{P}_{\text{conf}}$. Applying Theorem 2 we infer that $g^{-1}g' \in \mathcal{S}_{\text{conf}} \subset \mathcal{S}_{\text{reparam}}$. Since $\mathcal{S}_{\text{reparam}}$ is a group, we conclude that $f^{-1}f' = h^{-1} \circ (g^{-1}g') \circ h' \in \mathcal{S}_{\text{reparam}}$. This ends the proof. □

### E.3  Proof of Theorem 3

Now we address the case $d = 2$. Let us first note that conformal maps $f : \Omega \to \mathbb{R}^2$ for $\Omega \subset \mathbb{R}^2$ can be identified with holomorphic maps $g : \Omega \to \mathbb{C}$ with non-vanishing derivatives by considering $g(x + iy) = f_1(x + iy) + if_2(x + iy)$. Moreover,

$$|\det Df| = |g'|^2 \tag{76}$$

where $g'$ denotes the complex derivative. The proof of Theorem 3 is based on the fact that conformal maps between rectangles can be characterized rather explicitly. In particular, we use the following result.

**Theorem 8** (Schwarz-Christoffel mapping). *Conformal maps $f$ that map the unit disk $\mathbb{D} = \{z \in \mathbb{C} | |z| < 1\}$ to a polygon with angles $\alpha_k\pi$ for $1 \leq k \leq n$ can be written as*

$$f(z) = C \int_0^z \prod_{k=1}^n (w - w_k)^{-(1-\alpha_k)} \, \mathrm{d}w + K \tag{77}$$

*for two constants $C, K \in \mathbb{C}$ and $w_k \in \mathbb{C}$ with $|w_k| = 1$.*

A proof of this result can be found in any textbook on complex analysis, e.g., [1]. With this result we can prove Theorem 3.

*Proof of Theorem 3.* Assume there are two probability distribution $\mathbb{P}_1 \in \mathcal{P}_{\text{conf2}}$ and $\mathbb{P}_2 \in \mathcal{P}_{\text{conf2}}$ with densities $p_1$ and $p_2$ supported on finite rectangles $R_1$ and $R_2$ and a conformal map $f : R_1 \to R_2$ such that $f_*\mathbb{P}_1 = \mathbb{P}_2$. Let $h : \mathbb{D} \to R_1$ be a conformal map which by the Riemann mapping theorem exists and by Theorem 8 can be expressed as in (77), i.e.,

$$h(z) = C \int_0^z \prod_{k=1}^4 (z - w_k)^{-1/2} \, \mathrm{d}z + K \tag{78}$$

where $|w_k| = 1$ and we used that all angles in a rectangle are $\pi/2$. The points $w_k$ are the preimages of the corners of $R_1$. Then the derivative of $h$ can be written as

$$h'(z) = C \prod_{k=1}^4 (z - w_k)^{-1/2}. \tag{79}$$

Note that $h'$ is bounded above and below away from the points $w_k$ and $|h'(x)| \to \infty$ as $x \to w_k$. We can write $f = (f \circ h) \circ h^{-1}$ where $g = f \circ h$ is a conformal map from $\mathbb{D}$ to $R_2$. This can be written again as in (77) with points $w'_1, \ldots, w'_4$ which are the preimages of the corners of $R_2$, i.e.,

$$g(z) = C' \int_0^z \prod_{k=1}^4 (z - w'_k)^{-1/2} \, \mathrm{d}z + K' \tag{80}$$

The condition $f_*\mathbb{P}_1 = \mathbb{P}_2$ implies that

$$p_1(x) = p_2(f(x))|f'(x)|^2 \tag{81}$$

(using the relation (76) between complex derivative and maps from $\mathbb{R}^2 \to \mathbb{R}^2$). Now we infer

$$|f'(x)|^2 = |(f \circ h \circ h^{-1}(x)|^2 = |(f \circ h)'(h^{-1}(x))|^2 |(h^{-1})'(x)|^2 = \frac{|g'(y)|^2}{|h'(y)|^2} \tag{82}$$

where $y = h^{-1}(x)$. Applying this with $y \to w_k$ we get $h'(y) \to \infty$. Using the assumption that the densities of $\mathbb{P}_1$ and $\mathbb{P}_2$ are bounded above and below we infer from (81) and (82) that then

$|g'(y)| \to \infty$ from which we conclude $y \to w'_{k'}$ for some $k'$. This implies $w_k = w'_{k'}$. After renaming the corners, we get $w_k = w'_k$ for all $k$. From (78) and (80) we see that

$$C'^{-1}(g(z) - K') = C^{-1}(h(z) - K). \tag{83}$$

Thus $g(z) = C'h(z)/C - KC'/C + K'$ so we conclude that

$$(g \circ h^{-1})(h(z)) = g(z) = C'h(z)/C - KC'/C + K'. \tag{84}$$

In particular there are constants $A, B \in \mathbb{C}$ such that $f(x) = (g \circ h^{-1})(x) = Ax + B$. From $f(R_1) = R_2$ we infer that $A \in \mathbb{R} \cup i\mathbb{R}$. This ends the proof. $\qquad\square$

Let us finally remark that the identifiability of conformal maps for distributions with full support in $d = 2$ follows just as in $d \geq 3$ because every bijective conformal map of the Riemann sphere to itself is a Moebius transformation so we can apply Lemma 3. We expect that the result can be extended to more general densities using the same strategy and a more careful analysis of the density close to the boundary.

# F   Proofs for the results on OCTs

In this section we collect the missing proofs for Section 4.

## F.1   Proof of Proposition 1

First we prove Proposition 1. We refer to Appendix C for a general review and characterization of spurious solutions.

*Proof of Proposition 1.*   The proof essentially shows that polar coordinates are an example of such a function. In dimension $d$ they are defined by $\Phi : (a, b) \times [0, \pi] \times [0, \pi/2]^{d-2} \to \mathbb{R}^d$ with

$$\begin{aligned}
x_1 &= r \sin(\varphi) \sin(\theta_1) \ldots \sin(\theta_{d-2}) \\
x_2 &= r \cos(\varphi) \sin(\theta_1) \ldots \sin(\theta_{d-2}) \\
x_3 &= r \cos(\theta_1) \ldots \sin(\theta_{d-2}) \\
&\vdots \; \vdots \qquad \vdots \\
x_d &= r \cos(\theta_{d-2}).
\end{aligned} \tag{85}$$

The following lemma is well known.

**Lemma 4.** *The polar coordinates $\Phi$ defined above satisfy:*

1. *The transformation satisfies $\Phi \in \mathcal{F}_{\text{OCT}}$.*

2. *The determinant of the Jacobian is given by*

$$\det D\Phi_d = r^{d-1} \sin\theta_1 \sin(\theta_2)^2 \ldots \sin(\theta_{d-2})^{d-2}. \tag{86}$$

3. *The image of $\Phi_d$ is, up to a set of measure zero, the annulus $\{x \in \mathbb{R}^n : a < |x| < b\}$.*

*Proof.*   For the first part we only need to show that $D\Phi$ has orthogonal columns which can formally be done by induction noting that

$$\Phi_d(r, \varphi, \theta_1, \ldots, \theta_{d-2}) = (\Phi_{d-1}(r, \varphi, \theta_1, \ldots, \theta_{d-3}) \sin(\theta_{d-2}), r \cos(\theta_{d-2})). \tag{87}$$

The determinant and the image can also be derived from this recursion. $\qquad\square$

We now define for $k \in \mathbb{N}$ the functions $g_k : (0, \pi/2) \to \mathbb{R}$

$$g_k(\theta) = \int_0^\theta \sin(t)^k \, dt. \tag{88}$$

Clearly $g_k'(\theta) = \sin(\theta)^k$. They are strictly increasing functions with positive derivative, i.e., diffeomorphisms, so we can define their inverses on an open interval $h_k : I_k \to (0, \pi/2)$ which are also differentiable functions. Note that

$$h_k'(\alpha) = (g_k'(h_k(\alpha)))^{-1} = \sin(h_k(\alpha))^{-k}. \tag{89}$$

Denote the density of $\mathbb{P}$ by $p$. By assumption $\mathbb{P}$ is invariant under rotations so we can write with a slight abuse of notation $p(s) = p(|s|)$. We assume that $p(|s|)$ is positive on a, possibly unbounded, interval $(a, b)$. We consider $q : \mathbb{R}_+ \to \mathbb{R}$ given by

$$q(r) = d\omega_d r^{d-1} p(r) \tag{90}$$

where $\omega_d$ is the volume of the unit ball in dimension $d$ and these constants ensure that $q$ is a probability density. Define $F_q : (0, \infty) \to [0, 1]$ as the cdf for the probability density $q$, i.e.,

$$F_q(r) = \int_0^r q(z)\, \mathrm{d}z. \tag{91}$$

Since $q(z) > 0$ iff $t \in (a, b)$ we conclude that $F_q$ restricted to $(a, b)$ is a continuous and strictly increasing from 0 to 1 and has a positive derivative. Hence, we can define $\psi : (0, 1) \to (a, b)$ by $\psi(t) = F_q^{-1}(t)$ and $\psi$ is differentiable with

$$\psi'(t) = (F_q'(\psi(t)))^{-1} = \frac{1}{q(\psi(t))}. \tag{92}$$

We define the domains $D_d = (0, 1) \times (0, 2\pi) \times I_1 \times \ldots I_{d-2}$ Now we consider the map $h : D_d \to (a, b) \times (0, 2\pi) \times (0, \pi)^{d-2}$ given by

$$h(t, \varphi, \alpha_1, \ldots, \alpha_{d-2}) = (\psi(t), \varphi, h_1(\alpha_1), \ldots, h_{d-2}(\alpha_{d-2})). \tag{93}$$

Note that $h$ is a coordinate-wise transformation and the determinant of its Jacobian is given by

$$\det Dh = \frac{1}{q(\psi(t))} \prod_{k=1}^{d-2} \sin(h_k(\alpha))^{-k}. \tag{94}$$

Moreover Lemma 4 gives

$$\det(D\Phi)(h(t, \varphi, \alpha_1, \ldots, \alpha_{d-2})) = \psi(t)^{d-1} \sin(h_1(\alpha_1)) \sin(h_2(\alpha_2))^2 \ldots \sin(h_{d-2}(\alpha_{d-2}))^{d-2} \tag{95}$$

and thus

$$\det D(\Phi \circ h)(t, \varphi, \alpha_1, \ldots, \alpha_{d-2}) = \frac{\psi(t)^{d-1}}{q(\psi(t))}. \tag{96}$$

We now consider the probability measure $\mathbb{Q} = \mathcal{U}(0, 1) \otimes \mathcal{U}((0, 2\pi)) \otimes \mathcal{U}(I_1) \otimes \ldots \otimes \mathcal{U}(I_{n-2})$ where $\mathcal{U}$ denotes the uniform distribution on an interval. Denote by $\lambda : \mathbb{R}^d \to \mathbb{R}^d$ the scaling function $\lambda(s) = (s_1, 2\pi s_2, |I_1|s_3, \ldots, |I_{d-2}|s_d)$. Clearly $\lambda$ maps $C_d$ to $(0, 1) \times (0, 2\pi) \times I_1 \times \ldots \times I_{d-1}$ and $\lambda_* \nu = \mathbb{Q}$. We now consider the measure

$$(\Phi \circ h \circ \lambda)_* \nu = (\Phi \circ h)_* \mathbb{Q}. \tag{97}$$

We denote its density by $\tilde{p}$ and obtain using (13), (96), and that the density of $\mathbb{Q}$ is constant

$$\begin{aligned}
\tilde{p}(x) &\propto |\operatorname{Det} D(\Phi \circ h)^{-1})(x))| \\
&= \frac{1}{|\operatorname{Det} D(\Phi \circ h)((\Phi \circ h)^{-1}(x))|} \\
&= \frac{q\big(\psi((\Phi \circ h)_1^{-1}(x))\big)}{\psi((\Phi \circ h)_1^{-1}(x))^{d-1}}.
\end{aligned} \tag{98}$$

Note that by definition of $\Phi$ we have $\Phi^{-1}(x)_1 = |x|$ and $h$ acts coordinatewise where the action on the first coordinate is $\psi$ so that

$$\psi((\Phi \circ h)^{-1}(x)_1) = \psi((h^{-1} \circ \Phi^{-1})(x)_1) = \psi(\psi^{-1}(|x|)) = |x|. \tag{99}$$

We conclude, using (90) and the last display, that

$$\tilde{p}(x) \propto \frac{q(|x|)}{|x|^{d-1}} \propto p(|x|). \tag{100}$$

This shows

$$\mathbb{P} = (\Phi \circ h \circ \lambda)_* \nu. \tag{101}$$

It remains to be shown that $\Phi \circ h \circ \lambda \in \mathcal{F}_{\mathrm{OCT}}$. But we have seen in Lemma 4 that $\Phi \in \mathcal{F}_{\mathrm{OCT}}$ and $h \circ \lambda$ is an invertible coordinate-wise transformation so that $h \circ \lambda \in \mathcal{F}_{\mathrm{OCT}}^R$ (see (28)) implying $\Phi \circ h \circ \lambda \in \mathcal{F}_{\mathrm{OCT}}$. Note that we here essentially use the fact that precomposition with a function that acts on each coordinate separately preserves orthogonality of the columns of the Jacobian. □

### F.2   Proof of Theorems 4

We now consider smooth deformations of a data generating mechanism $x = f(s)$. For this it is helpful to phrase these as flows generated by vector fields. For a brief review of these notions we refer to Appendix A and for an extensive introduction we refer to any textbook on differential geometry. We now give a complete proof of Theorem 4.

*Proof of Theorem 4.* To simplify the notation we first focus on the case where $M = \mathbb{R}^d$. The necessary modifications for the manifold case are indicated afterwards. We define

$$\Psi_t = (\Phi_t)^{-1} \circ f_t \tag{102}$$

so that $\Psi_t : \mathbb{R}^d \to \mathbb{R}^d$ is a diffeomorphism for every $t$ and $\Psi_0(s) = s$ by assumption. We denote the vector field that generates $\Psi_t$ by $X : (0,1)^d \to \mathbb{R}^d$, i.e., $X$ satisfies

$$\partial_t \Psi_t(x) = X_t(\Psi_t(x)). \tag{103}$$

The assumption $(\Phi_t)_* \nu = f_* \nu$ implies $\nu = (\Psi_t)_* f_* \nu$. Then the continuity equation (16) implies that $\mathrm{Div}(X_t) = 0$ on $(0,1)^d$. By assumption, $\Phi_t \in \mathcal{F}_{\mathrm{OCT}}$, which means that

$$(D\Phi_t)^\top D\Phi_t = \Lambda_t \tag{104}$$

where $\Lambda_t : (0,1)^d \to \mathrm{Diag}(d)$ maps to diagonal matrices. Similarly $f_t \in \mathcal{F}_{\mathrm{OCT}}$ implies that there is a function $\Omega_t : (0,1)^d \to \mathrm{Diag}(d)$ such that

$$(Df_t)^\top Df_t = \Omega_t. \tag{105}$$

We now evaluate $\partial_t \Omega_t$ in terms of $\Psi_t$ and $\Phi_t$. To evaluate the time derivative it is convenient to write $\Psi(t,s)$ instead of $\Psi_t(s)$. We get, using $f_t(s) = \Phi(t, \Psi(t,s))$,

$$
\begin{aligned}
\partial_t Df_t(s) &= \partial_t \left( (D\Phi)(t, \Psi(t,s))(D\Psi)(t,s) \right) \\
&= (\partial_t D\Phi)(t, \Psi(t,s))(D\Psi)(t,s) + \sum_{k=1}^d (\partial_t \Psi)_k(t,s)(\partial_k D\Phi)(t, \Psi(t,s))(D\Psi)(t,s) \\
&\quad + (D\Phi)(t, \Psi(t,s))(D\partial_t \Psi)(t,s) \\
&= (\partial_t D\Phi_t)(\Psi_t(s))(D\Psi_t)(s) + \sum_{k=1}^d (X_t)_k(\Psi_t(s))(\partial_k D\Phi_t)(\Psi_t(s))(D\Psi_t)(s) \\
&\quad + (D\Phi_t)(\Psi_t(s))(D(X_t(\Psi_t(s))) \\
&= \left( \left( (\partial_t D\Phi_t) + \sum_{k=1}^d (X_t)_k(\partial_k D\Phi_t) + (D\Phi_t)(DX_t) \right) \circ \Psi_t(s) \right)(D\Psi_t)(s).
\end{aligned}
\tag{106}
$$

Note that also

$$Df_t(s) = (D\Phi_t)(\Psi_t(s))(D\Psi_t)(s). \tag{107}$$

Combining this with the last display we get (dropping the positional argument $s$ for conciseness)

$$
\begin{aligned}
(D\Psi_t)^{-\top}(\partial_t\Omega_t)(D\Psi_t)^{-1} &= (D\Psi_t)^{-\top}\left(Df_t^\top(\partial_t Df_t) + (\partial_t Df_t)^\top Df_t\right)(D\Psi_t)^{-1} \\
&= \Bigg[D\Phi_t^\top\left((\partial_t D\Phi_t) + \sum_{k=1}^d (X_t)_k(\partial_k D\Phi_t) + (D\Phi_t)(DX_t)\right) \\
&\qquad + \left((\partial_t D\Phi_t) + \sum_{k=1}^d (X_t)_k(\partial_k D\Phi_t) + (D\Phi_t)(DX_t)\right)^\top D\Phi_t\Bigg]\circ\Psi_t \\
&= \Bigg[D\Phi_t^\top(\partial_t D\Phi_t) + (\partial_t D\Phi_t)^\top D\Phi_t + D\Phi_t^\top D\Phi_t DX_t + DX_t^\top D\Phi_t^\top D\Phi_t \\
&\qquad + \sum_{k=1}^d (X_t)_k\left((\partial_k D\Phi_t)^\top D\Phi_t + D\Phi_t^\top(\partial_k D\Phi_t)\right)\Bigg]\circ\Psi_t \\
&= \left(\partial_t\Lambda_t + \Lambda_t DX_t + DX_t^\top\Lambda_t + \sum_k (X_t)_k\partial_k\Lambda_t\right)\circ\Psi_t.
\end{aligned}
\tag{108}
$$

We now consider $t=0$ and drop the $t$ argument from the notation. By assumption $\Psi_0 = \mathrm{Id}$ so the equation simplifies to

$$
\partial_t\Omega = \partial_t\Lambda + \Lambda DX + DX\Lambda + \sum_k X_k\partial_k\Lambda.
\tag{109}
$$

Now we use that $\Lambda_t$ and $\Omega_t$ map to diagonal matrices for all $t$, in particular $\partial_t\Lambda$ and $\partial_t\Omega$ are diagonal matrices. We conclude that for $i\neq j$ the equation

$$
\begin{aligned}
0 &= (\Lambda DX)_{ij} + ((DX)^\top\Lambda)_{ij} = \Lambda_{ii}(DX)_{ij} + (DX)_{ji}\Lambda_{jj} \\
&= \Lambda_{ii}\partial_j X_i + \Lambda_{jj}\partial_i X_j
\end{aligned}
\tag{110}
$$

holds. Thus, we obtain a system of first order Partial Differential Equations (PDE) for $X_{t=0}$. We now write $\Lambda_j = \Lambda_{jj}$. We also fix an $i\in\{1,\dots,d\}$ in the following. Then we can rewrite (110) concisely as

$$
\Lambda_i\partial_j X_i + \Lambda_j\partial_i X_j = 0 \quad \text{for } i\neq j.
\tag{111}
$$

We divide equation (111) by $\Lambda_j$ apply $\partial_j$ and sum over $j\neq i$ to obtain

$$
\sum_{j\neq i}\partial_j\left(\frac{\Lambda_i}{\Lambda_j}\partial_j X_i\right) = -\sum_{j\neq i}\partial_j\partial_i X_j = -\partial_i\operatorname{Div}X + \partial_i^2 X_i = \partial_i^2 X_i.
\tag{112}
$$

This implies that $X_i$ satisfies the wave equation

$$
\partial_i^2 X_i - \sum_{j\neq i}\partial_j(a_j\partial_j X_i) = 0 \text{ on } (0,1)^d
\tag{113}
$$

where $a_j = \Lambda_i/\Lambda_j$. Note that by assumption $a_j\in C^1((0,1)^d)$ and $a_j$ is positive because we assumed that $\Phi_t$ are diffeomorphisms implying $\Lambda_j > 0$ (because $D\Phi$ is invertible). Now we use the assumption that $\Phi_t(s) = f_t(s)$ if $\mathrm{dist}(s,\partial C_d) < \varepsilon$. This implies for such $s$ that $s = (\Phi_t^{-1}\circ f_t)(s) = \Psi_t(s)$, i.e., $\Psi_t(s)$ is constant. We conclude that $0 = \partial_t\Psi_t(s) = X_t(\Psi_t(s)) = X_t(s)$ for all $s$ satisfying $\mathrm{dist}(s,\partial C_d) < \varepsilon$.

Now we claim that this together with the PDE (113) implies that $X_i$ vanishes everywhere. We set $\varepsilon' = \varepsilon/d$. Then we have $\mathrm{dist}(s,\partial C_d) < \varepsilon$ for all $s\notin(\varepsilon',1-\varepsilon')^d$, Thus $X_i$ solves the PDE (113) on $(\varepsilon',1-\varepsilon')^d$ with vanishing boundary data and vanishing derivatives at the boundary. Then the uniqueness of solutions for the Cauchy problem for hyperbolic PDE of second order which we stated in Theorem 9 below implies that there is at most one solution. Note that the ellipticity condition in (131) follows by noting that the functions $a_j$ are continuous and positive, and thus $\min_{(\varepsilon,1-\varepsilon)^d} a_j > 0$.

Since $X_i = 0$ clearly solves the PDE, we conclude that $X_i = 0$. This argument applies to all $i$ so we conclude that $X_{t=0} = 0$. Note that if $f_t = f_0$ is constant (this case is state in Corollary 2) the left

hand side of (108) vanishes for all $t$. Then we can infer with the same argument that $X_t = 0$ for all $t$ and thus $\Psi_t = \mathrm{Id}$ and $\Phi_t = f$.

We now proceed with the general case. Since by assumption $\Psi_t$ is analytic in $t$ it is sufficient to show that $\partial_t^{l+1} \Psi_{t=0}(s) = 0$ for all $l \in \mathbb{N}$ which implies that $\Psi_t(s) = \Psi_0(s) = s$. We denote $X_t^{(l)} = \partial_t^l X_t$. By definition of $X_t$ and the chain rule it is easy to see that $X_{t=0}^{(l')} = 0$ for $0 \le l' \le l$ implies $\partial_t^{l+1} \Psi_{t=0}(s) = 0$. We now show this by induction. We apply $\partial_t^l$ to (108) at $t = 0$. We obtain (using $\Psi_0 = \mathrm{Id}$)

$$\partial_t^{l+1} \Omega_t = \partial_t^{l+1} \Lambda_t + \Lambda D X_t^{(l)} + (D X_t^{(l)})^\top \Lambda_t + \sum_k (X_0^{(l)})_k \partial_k \Lambda_t + R(X_t, \ldots, X_t^{(l-1)}) \quad (114)$$

where $R$ denotes a remainder term where each summand contains a factor of the form $X_{t=0}^{(l')}$ for some $0 \le l' < l$. Indeed, every time we differentiate a $\Psi_t$ term we get a $X_t$ term. By the induction hypothesis $X_{t=0}^{(l')} = 0$ for $l' < l$ and therefore $R = 0$. We conclude that $X^{(l)}$ satisfies (111) just as $X = X^{(0)}$. Differentiating $\mathrm{Div}\, X = 0$ with respect to $t$ we also conclude $\mathrm{Div}\, X^{(l)} = 0$ and as before we conclude that $X^{(l)}(s) = 0$ for $s \notin (\varepsilon', 1 - \varepsilon')^d$. As above this implies $X_{t=0}^{(l)}(s) = 0$. This ends the proof if $M = \mathbb{R}^d$.

We now discuss the necessary extensions for the general case. So we now assume that $f_t, \Phi_t : \mathbb{R}^d \to M$ for some Riemannian manifold $(M, g)$. The main strategy is to establish that (109) holds where the definitions of the involved quantities are adapted suitably. Then we can conclude the proof as above. Note that if we could find orthonormal coordinates locally the same proof applies. However, this is general not possible so we need to argue more carefully. We define $\Psi_t : \mathbb{R}^d \to \mathbb{R}^d$ and $X_t$ as above, i.e., $\Psi_t = (\Phi_t)^{-1} \circ f_t$ and $\partial_t \Psi_t = X_t(\Psi_t)$. Then we define the matrix functions $\Lambda_t, \Omega_t \in \mathbb{R}^{d \times d}$ by

$$(\Lambda_t)_{ij}(s) = \langle (d\Phi_t)(s) e_i, (d\Phi_t)(s) e_j \rangle_g, \quad (115)$$

$$(\Omega_t)_{ij}(s) = \langle (df_t)(s) e_i, (df_t)(s) e_j \rangle_g. \quad (116)$$

By assumption $\Lambda_t$ and $\Omega_t$ are diagonal. We consider a (inverse) chart $\eta : U \to M$ where $U \subset \mathbb{R}^d$. We now argue locally on $\eta(U) \subset M$ but we do not denote domain restriction of the function to improve readability. We define

$$\tilde{\Phi}_t = \eta^{-1} \circ \Phi_t, \quad (117)$$

$$\tilde{f}_t = \eta^{-1} \circ f_t. \quad (118)$$

Maps $\tilde{\Phi}_t$ and $\tilde{f}_t$ map $\mathbb{R}^d$ to itself and we can consider their usual derivatives. Moreover, we have $\Psi_t = (\tilde{\Phi}_t)^{-1} \circ \tilde{f}_t$. Observe that (using $\eta \tilde{\Phi} = \Phi$) we can rewrite $\Lambda_t$ using the induced metric through $\eta$ on $\mathbb{R}^d$

$$\begin{aligned}
(\Lambda_t)_{ij}(s) &= \langle (d\eta)(\tilde{\Phi}_t(s))(d\tilde{\Phi}_t(s)) e_i, (d\eta)(\tilde{\Phi}_t(s))(d\tilde{\Phi}_t(s)) e_j \rangle_g \\
&= \langle (d\tilde{\Phi}_t(s)) e_i, (d\eta)^*(\tilde{\Phi}_t(s))(d\eta)(\tilde{\Phi}_t(s))(d\tilde{\Phi}_t(s)) e_j \rangle_{\mathbb{R}^d} \\
&= \langle D\tilde{\Phi}_t(s) e_i, G(\tilde{\Phi}_t(s)) D\tilde{\Phi}_t(s) e_j \rangle_{\mathbb{R}^d}
\end{aligned} \quad (119)$$

where we defined $G : U \to \mathbb{R}^{d \times d}$ by $G(z) = (d\eta)^*(z)(d\eta)(z)$ which captures the pullback metric on $\mathbb{R}^d$. We can express this concisely and get similarly for $\Omega_t$

$$\Lambda_t(s) = D\tilde{\Phi}_t(s)^\top G(\tilde{\Phi}_t(s)) D\tilde{\Phi}_t(s), \quad (120)$$

$$\Omega_t(s) = D\tilde{f}_t(s)^\top G(\tilde{f}_t(s)) D\tilde{f}_t(s). \quad (121)$$

We now consider as above $\partial_t \Omega_t$ and get using the last display

$$\partial_t \Omega_t = (\partial_t D\tilde{f}_t)^\top G(\tilde{f}_t) D\tilde{f}_t + D\tilde{f}_t^\top (\partial_t G(\tilde{f}_t)) D\tilde{f}_t + D\tilde{f}_t^\top G(\tilde{f}_t)(\partial_t D\tilde{f}_t). \quad (122)$$

Next we calculate

$$\partial_t(G \circ \tilde{f}_t) = \partial_t(G \circ \tilde{\Phi}_t \circ \Psi_t) = \partial_t(G \circ \tilde{\Phi}_t) \circ \Psi_t + \sum_k ((X_t)_k \circ \Psi_t) \cdot (\partial_k(G \circ \tilde{\Phi}_t)) \circ \Psi_t \quad (123)$$

Using the relations (106), (107) (for $\tilde{f}$) and (123) in (122) we obtain

$$(D\Psi_t)^{-\top}\partial_t\Omega_t(D\Psi_t)^{-1} = A_1 + A_2 + A_3, \tag{124}$$

where

$$A_1 = \left[(\partial_t D\tilde{\Phi}_t)^\top(G\circ\tilde{\Phi}_t)D\tilde{\Phi}_t + (D\tilde{\Phi}_t)^\top(G\circ\tilde{\Phi}_t)(\partial_t D\tilde{\Phi}_t) + (D\tilde{\Phi}_t)^\top(\partial_t(G\circ\tilde{\Phi}_t))D\tilde{\Phi}_t\right]\circ\Psi_t$$

$$= (\partial_t\Lambda_t)\circ\Psi_t,$$

$$A_2 = \left[DX_t^\top(D\tilde{\Phi}_t)^\top(G\circ\tilde{\Phi}_t)D\tilde{\Phi}_t + (D\tilde{\Phi}_t)^\top(G\circ\tilde{\Phi}_t)D\tilde{\Phi}_t DX_t\right]\circ\Psi_t$$

$$= (DX_t^\top\Lambda_t + \Lambda_t DX_t)\circ\Psi_t,$$

$$A_3 = \left[\sum_{k=1}^d (X_t)_k\left((\partial_k D\tilde{\Phi}_t)^\top(G\circ\tilde{\Phi}_t)(D\tilde{\Phi}_t) + (D\tilde{\Phi}_t)^\top(G\circ\tilde{\Phi}_t)(\partial_k D\tilde{\Phi}_t)\right)\right.$$

$$\left. + (D\tilde{\Phi}_t)^\top\sum_k(X_t)_k\cdot(\partial_k(G\circ\tilde{\Phi}_t))D\tilde{\Phi}_t\right]\circ\Psi_t$$

$$= \left(\sum_k(X_t)_k\partial_k\Lambda_t\right)\circ\Psi_t. \tag{125}$$

Plugging the relations in (124) we obtain

$$(D\Psi_t)^{-\top}\partial_t\Omega_t(D\Psi_t)^{-1} = \left(\partial_t\Lambda_t + \Lambda_t DX_t + DX_t^\top\Lambda_t + \sum_k(X_t)_k\partial_k\Lambda_t\right)\circ\Psi_t. \tag{126}$$

Thus we established that the relation (108) also holds in the manifold case. The rest of the proof is the same. □

The proof of Corollary 2 is trivial.

*Proof of Corollary 2.* Apply Theorem 4 with $f_t = f_0$ constant. The assumption that $\Phi_t$ is analytic in $t$ can be dropped as explained in the proof of Theorem 4. □

For reference, we now state the uniqueness result for second order hyperbolic partial differential equations. Let $U \subset \mathbb{R}^n$ open, bounded and let $U_T = U \times (0, T)$. Consider the boundary problem

$$\partial_t^2 u + Lu = f \text{ in } U_T \tag{127}$$

$$u = 0 \text{ in } \partial U \times [0, T] \tag{128}$$

$$u = g, \partial_t u = h \text{ on } U \times \{0\} \tag{129}$$

where $f : U_T \to \mathbb{R}$ and $g, h : U \to \mathbb{R}$ are given functions which we assume to be $C^1$ and $g = 0$ on $\partial U$. The function $u : U_T \to \mathbb{R}^d$ is the unknown. The operator is assumed to be an elliptic operator given by

$$Lu = -\sum_{i,j=1}^n \partial_i(a^{ij}(x,t)\partial_j u) \tag{130}$$

where we assume $a^{ij} \in C^1(\bar{U}_T)$, $a^{ij} = a^{ji}$, and that there is $\theta > 0$ such that

$$\sum_{i,j=1}^n \xi_i\xi_j a^{ij}(x,t) \geq \theta|\xi|^2 \tag{131}$$

for all $(x,t) \in U_T$ and $\xi \in \mathbb{R}^n$. Then the following result holds.

**Theorem 9** (Theorem 4 in Section 7.2 in [14])**.** *Under the assumptions above there is a unique weak solution $u$ of the system* (127) *with boundary values as in* (128) *and* (129).

For our purposes it is not necessary to define weak solution let us just emphasize that any classical solution is a weak solution so this implies uniqueness of classical solutions.

The key obstacle to improve upon this result and to remove the compact support condition on $X$ is that the resulting PDE in equation (113) is well posed for the Cauchy initial value problem but it is not well posed for the Dirichlet problem or for mixed Dirichlet and Neumann boundary data. In particular, solutions are, in general, not unique. Furthermore, there are no general uniqueness results for first order systems as in (110). Note that the existence of a non-trivial divergence free solution $X_0$ of (110) does not imply that a non-constant flow $\Phi_t$ exists because this is not sufficient to define the flow for positive times.

We illustrate the influence of the boundary condition further below, when we prove Theorem 5.

### F.3 Proof of Theorem 5

In this section we show that a family of simple mixing functions is locally identifiable for most parameter values even when the mixing is not known close to the boundary. Note that actually we can construct a set of parameter values for which this holds giving a slightly stronger result that we state now. Theorem 5 will be simple consequence of this result.

**Theorem 10.** *Let* $f : C_d \to \mathbb{R}^d$ *be given by* $f(x) = RDx$, *where* $R \in \mathrm{O}(d)$ *and* $D = \mathrm{Diag}(\mu_1, \ldots, \mu_d)$ *with* $\mu_i > 0$ *and* $\mu_i^{-2}$ *are linearly independent over the rational numbers* $\mathbb{Q}$. *Suppose that* $\Phi_t$ *is a smooth invariant deformation in* $\mathcal{F}_{\mathrm{OCT}}$ *such that* $\Phi_0 = f$, $(\Phi_t)_*\nu = f_*\nu$, *and* $\Phi_t$ *is analytic in t. Then* $\Phi_t = f$ *on* $C_d$, *i.e.,* $\Phi_t$ *is constant in time.*

*Proof.* The initial part of the proof proceeds as in the proof of Theorem 4 and we keep using the same notation. In particular, $\Psi_t = (\Phi_t)^{-1} \circ f$ and $X_t$ is given by $\partial_t \Psi_t = X_t \circ \Psi_t$.

Note that now $f_t$ is constant in $t$ and therefore $\Omega_t = Df_t^\top Df_t$ is constant in $t$. We now investigate the boundary conditions for equation 113. Note that

$$\nu = (\Phi_t)_*^{-1}(\Phi_t)_*\nu = (\Phi_t)_*^{-1}f_*\nu = (\Psi_t)_*\nu. \tag{132}$$

So $\Psi_t$ preserves $\nu$ and we conclude that $\Psi_t((0,1)^d) = (0,1)^d$. Let us denote by

$$D_i = \{x \in [0,1]^d \,|\, x_i \in \{0,1\}\} \tag{133}$$

the boundary hyperplanes and write $D = \partial C_d = \partial(0,1)^d = \bigcup_i D_i$. As $\Psi_t$ maps $(0,1)^d$ bijectively to itself we conclude that

$$(X_t)_i = 0 \quad \text{on } D_i. \tag{134}$$

We now focus on $t = 0$ and use the shorthand $X = X_0$. Then the differential equation (111) implies that

$$\partial_i X_j = \Lambda_i/\Lambda_j \partial_j X_i = 0 \quad \text{on } D_i. \tag{135}$$

We conclude that the function $X_i$ solves the following mixed Dirichlet and Neumann type boundary problem

$$\partial_i^2 X_i - \sum_{j \neq i} \partial_j(a_j \partial_j X_i) = 0 \text{ on } (0,1)^d \tag{136}$$

$$\partial_j X_i = 0 \text{ on } D_j \text{ for } j \neq i \tag{137}$$

$$X_i = 0 \text{ on } D_i. \tag{138}$$

Recall here that $a_j = \Lambda_i/\Lambda_j$. So far, we have not used any specific assumption except that $\Phi_t$ is a continuous deformation and $\Phi_0 \in \mathcal{F}_{\mathrm{OCT}}$. So the existence of non-trivial continuous deformations implies that a certain hyperbolic PDE has a non-trivial solution. Unfortunately, this type of boundary value problem for hyperbolic equations is not well posed and has not always a unique solution. We now show that in the specific setting of the theorem uniqueness holds. In this case $f$ is linear and

$$Df = R \, \mathrm{Diag}(\mu_1, \ldots, \mu_d) \tag{139}$$

so

$$\Lambda = (Df)^\top Df = \mathrm{Diag}(\mu_1^2, \ldots, \mu_d^2). \tag{140}$$

This implies

$$a_j = \Lambda_i/\Lambda_j = \mu_i^2/\mu_j^2, \tag{141}$$

in particular $a_j$ is constant. So the equation (136) becomes a constant coefficient hyperbolic equation which can be solved explicitly.

We can now use Theorem 1 from [12] (and a simple scaling argument) we conclude that the system (136) has a unique solution which is $X_i = 0$ (actually this result is for $X_i = 0$ on $\partial D$ but the proof is still valid). To give an intuition, we note that separation of variable is possible in this setting and all solutions to the boundary value problem (136) and (137) (i.e., without the boundary condition (138) for $D_i$) can be expressed as a linear combination of the form

$$X_i(s) = f(s_i) \prod_{j \neq i} \varphi_j(s_j) \tag{142}$$

where $\varphi_j$ are eigenfunctions of the problem $\varphi_j'' = \lambda_j \varphi_j$ on $(0,1)$ and $\varphi_j'(0) = \varphi_j'(1) = 0$. It is easy to see that those are given by $\cos(\pi m_j t)$ where $m_j \in \mathbb{N}_0$ and then $\varphi_j''(s_j) = \pi^2 m_j^2 \varphi_j$. Solving for $f$ we find from (136) that $f$ satisfies the ode

$$f''(s_i) = \left( \sum_{j \neq i} \pi^2 \frac{\mu_i^2}{\mu_j^2} m_j^2 \right) f(s_i). \tag{143}$$

Using now that $X_i(0) = 0$ (by (138)) we conclude $f(0) = 0$ and therefore

$$f(s_i) = C \sin(\pi \alpha s_i) \tag{144}$$

where $\alpha = \sqrt{\sum_{j \neq i} m_j^2 \mu_i^2/\mu_j^2}$, or equivalently

$$0 = \alpha^2 \mu_i^{-2} - \sum_{j \neq i} m_j^2 \mu_j^{-2}. \tag{145}$$

Now the condition $X_i(s) = 0$ for all $s \in [0,1]^d$ with $s_i = 1$ is satisfied if and only if $f(1) = 0$ which holds iff $\alpha \in \mathbb{N}_0$. Note that this argument also implies to solutions that are sums of functions as in (142) by linear independence. Then the assumption that $\mu_i^{-2}$ are linearly independent over $\mathbb{Q}$ implies that $\alpha = 0$ (and $m_j = 0$) which implies $X_i = 0$. Note that this argument only applies at $t = 0$ because it heavily relies on the explicit form of $\Phi_0 = f$. However, we can apply the same reasoning to $\partial_t^k X_t$ inductively (just as in the proof of Theorem 4) and then conclude using the assumption that $\Phi_t$ is analytic in $t$.

The complete argument goes as follows. We take the time derivative of equation (109) (recall that $\partial_t \Omega_t = 0$ as $f_t = f_0$ and get, denoting $\dot X_t = \partial_t X_t$ and $\dot \Lambda_t = \partial_t \Lambda_t$,

$$(DX_t)^\top \dot\Lambda_t + \Lambda_t DX_t + (D\dot X_t)^\top \Lambda_t + \Lambda_t D\dot X_t \in \text{Diag}(d). \tag{146}$$

We have seen that $DX_0 = 0$ so we infer

$$(D\dot X_0)^\top \Lambda_0 + \Lambda_0 D\dot X_0 \in \text{Diag}(d) \tag{147}$$

and $\text{Div}\, \dot X_t = \partial_t \text{Div}\, X_t = 0$. The same arguments as before imply $\dot X_0 = 0$ on $(0,1)^d$. By induction all time derivatives of $X_0$ vanish. This implies that $\partial_t^l \Psi_{t=0}(s) = 0$ for all $s$ and $l$, i.e., its Taylor expansion at $t = 0$ disappears and since we assumed $\Phi_t$ to be analytic in $t$ so is $\Psi_t$ and we conclude that $\Psi_t(s) = \Psi_0(s) = s$ and therefore $\Phi_t(s) = f(s)$. $\qquad\square$

The proof of Theorem 5 is now simple.

*Proof of Theorem 5.* Using Theorem 10 we only need to show that with probability 1 the real numbers $\mu_i^{-2}$ are independent over $\mathbb{Q}$. Note that by assumption $\mu_i$ has a density. Thus, also the distribution of $\alpha_i = \mu_i^{-2}$ has a density, i.e., is absolutely continuous with respect to the Lebesgue measure. For a vector of rational numbers $(q_1, \dots, q_d)$ the set $H_q = \{\alpha \in \mathbb{R}^d \mid \sum q_i \alpha_i = 0\}$ is a codimension 1 hyperplane and thus has Lebesgue measure 0. As $\mathbb{Q}$ is countable this implies that $N = \bigcup_{q \in \mathbb{Q}^d} H_q$ is a null set. Note that $\alpha \in N$ iff $\alpha_i$ are linearly dependent over $\mathbb{Q}$. Since the distribution of $\alpha_i = \mu_i^{-2}$ is absolutely continuous with respect to the Lebesgue measure, we conclude that

$$\mathbb{P}(\alpha \in N) = 0 \tag{148}$$

which implies the result. $\qquad\square$

### F.4 Proofs for the construction of spurious solutions

Finally, we show how flows can be used to construct families of solutions to the ICA problem. This section contains the technical results missing in the overview given in Appendix C.

The first construction was described in Lemma 1. Let us for completeness give a proof (we emphasize again that this result is essentially taken from [25]).

*Sketch of proof of Lemma 1.* Note that it is sufficient to show that the maps $h_{R,a}$ are volume preserving for fixed $t$ so we ignore the time argument. It is easy to see that $h_{R,a}$ is bijective (the inverse is given $h_{Q,a}$ where $Q(t,r) = R(t,r)^{-1}$). Then we only need to show that $\mathrm{Det}\, Dh_{R,a}(s) = 1$ for all $s$. We calculate (denoting $r = |s - a|$)

$$
\begin{aligned}
(Dh_{R,a}(s))_{ij} = \partial_j(h_{R,a})_i &= R(r)_{ij} + \sum_k (\partial_j R)_{ik}(|s-a|)(s-a)_k \\
&= \partial_j(h_{R,a})_i = R(r)_{ij} + \sum_k (\partial_r R)_{ik}(r)(s-a)_k \partial_j |s-a|.
\end{aligned}
\tag{149}
$$

We conclude (writing $R' = \partial_r R$)

$$
Dh_{R,a}(s) = R(r) + R'(r)(s-a) \otimes \nabla |s-a| = R(r) + \frac{1}{|s-a|} R'(r)(s-a) \otimes (s-a). \tag{150}
$$

Then we obtain, using the matrix determinant lemma for rank 1 updates ($\mathrm{Det}(A + u \otimes v) = (1 + u \cdot A^{-1}v)\,\mathrm{Det}\,A$

$$
\mathrm{Det}\, Dh_{R,a}(s) = \left(1 + \frac{1}{|s-a|}(s-a)R(r)^\top R'(r)(s-a)\right)\mathrm{Det}(R(r)). \tag{151}
$$

Now we use that $R(r) \in \mathrm{O}(d)$ so $\mathrm{Det}(R(r)) = 1$ and $R(r)^{-1} = R(r)^\top$. Differentiating $R(r)^\top R(r) = \mathrm{Id}_d$ with respect to $r$ we conclude that $R(r)^\top R'(r)$ is skew which implies

$$
(s-a)R(r)^\top R'(r)(s-a) = 0. \tag{152}
$$

We have therefore shown $\mathrm{Det}\, Dh_{R,a}(s) = 1$, completing the proof. $\qquad\square$

Now we give another construction that also establishes Fact 2 based on suitable divergence free vector fields. All we need to construct is divergence free vector fields with compact support. Consider any smooth function $\varphi : \mathbb{R}^d \to \mathbb{R}$ such that its support is contained in $\Omega$. Then we consider the vector fields $X^{ij} : \mathbb{R}^d \to \mathbb{R}^d$ for $1 \leq i < j \leq d$ given by

$$
X_i^{ij} = \partial_j \varphi, \quad X_j^{ij} = -\partial_j \varphi, \quad X_k^{ij} = 0 \quad \text{for } k \notin \{i, j\}. \tag{153}
$$

Then we get $\mathrm{Div}\, X^{ij} = \partial_i \partial_j \varphi - \partial_j \partial_i \varphi = 0$. So those vector fields are divergence free and we conclude that the space

$$
\mathcal{X} = \{X : \mathbb{R}^d \to \mathbb{R}^d | \mathrm{supp}(X) \subset \Omega, \ \mathrm{Div}\, X = 0\} \tag{154}
$$

is infinite dimensional. Every $X \in \mathcal{X}$ generates a flow $\Phi_t$ defined by

$$
\partial_t \Phi_t = X(\Phi_t), \quad \Phi_0(s) = s. \tag{155}
$$

Using equation (16) we conclude that $(\Phi_t)_* \nu = \nu$ because $\nu$ has a constant density and the support condition of $X$ ensures that $\Phi_t((0,1)^d) = (0,1)^d$. Then the family $f_t = f \circ \Phi_t$ has the property that $(f_t)_* \nu = f_* \nu$. Note that this construction can be easily generalized to source distributions $\mathbb{P}$ with differentiable density $p$. In this case the condition $\Phi_t \mathbb{P} = \mathbb{P}$ is satisfied when $\mathrm{Div}(pX) = 0$. Clearly it is sufficient to consider $X = Y/p$ where $Y \in \mathcal{X}$ (assuming that $p > c$ for some $c > 0$ on $\Omega$).

## G   Proofs for the result on volume preserving maps

Next we show that this construction can be generalized to volume preserving transformations and we prove Theorem 6. Note that in the special case that the distribution of $s$ is $\nu$ the construction above already works. This is a special case because the condition $(f_t)_* \nu = f_* \nu$ already implies that $f_t$ is volume preserving as soon as $f$ is volume preserving as the density of $\nu$ is constant. So in this case the condition that $f_t$ is volume preserving and $(f_t)_* \nu = f_* \nu$ essentially agree which is not the case for general base measures.

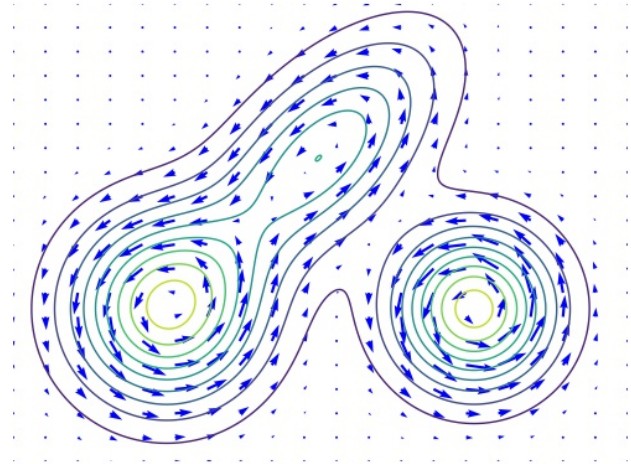

Figure 6: A sketch of the vector fields $X^{ij}$ for $d = 2$ constructed in the proof of Theorem 6. The closed lines are level lines of the probability density (which is a Gaussian mixture here). Note that the vector field is parallel to the level lines and its magnitude proportional to the norm of the gradient of the density.

*Proof of Theorem 6.* We define a suitable vector field explicitly. Consider $X^{ij} : \mathbb{R}^d \to \mathbb{R}^d$ for $1 \leq i < j \leq d$ defined by

$$X_k^{ij} = \begin{cases} \partial_j p & k = i \\ -\partial_i p & k = j \\ 0 & k \notin \{i, j\}. \end{cases} \tag{156}$$

An illustration of this vector field is given in Figure 6. We consider the family of functions $f_t = f \circ \Phi_t^{ij}$ where the flow $\Phi_t^{ij}$ is defined by $\Phi_0^{ij}(s) = s$ and $\partial_t \Phi_t^{ij}(s) = X(\Phi_t^{ij}(s))$. Note that boundedness of $\nabla p$ and $p \in C^2$ imply that $\Phi_t$ exists globally and defines a diffeomorphism. We claim that $\Phi_t^{ij}$ satisfies $\operatorname{Det} \Phi_t^{ij}(s) = 1$ for all $s$ and $(\Phi_t^{ij})_* \mathbb{P} = \mathbb{P}$. The former condition means that $\Phi^{ij}$ preserves the standard volume (Lebesgue-measure) which is the case if $\operatorname{Div}(X^{ij}) = 0$ while the second relation is satisfied if $\operatorname{Div}(pX^{ij}) = 0$ by equation (16). We calculate

$$\operatorname{Div} X^{ij} = \partial_i \partial_j p - \partial_j \partial_i p = 0. \tag{157}$$

We also find

$$\operatorname{Div}(X^{ij} p) = p \operatorname{Div}(X^{ij}) + X^{ij} \cdot \nabla p = \partial_j p \partial_i p - \partial_i p \partial_j p = 0. \tag{158}$$

This ends the proof. $\qquad \square$

To give an example, we consider $d = 2$ and $\mathbb{P}$ with rotation invariant density $p(s) = p(|s|)$. Then $X(s) = f(|s|)s^\perp$ where $s^\perp = (s_2, -s_1)^\top$ and the flow lines are circles around the origin where the speed depends on the radius through the derivative of $p(|s|)$. Let us add some remarks concerning this result.

*Remark* 1.     1.  The constructed flows are non-trivial, i.e., not constant because the probability density cannot be constant (as we assumed it to be $C^2$) and thus $X^{ij}$ is not identically vanishing.

    2.  It is easy to see (e.g., through the example above) that the flows $\Phi^{ij}$ will, in general, mix the coordinates $i$ and $j$ thus this really shows that ICA is not identifiable for volume preserving maps.

3. While we construct a finite family of solutions they can be combined, e.g.,

$$f' = f \circ \Phi_{t_1}^{i_1 j_1} \circ \ldots \cdot \Phi_{t_k}^{i_k j_k} \tag{159}$$

   to yield a large space of solutions.

4. By choosing coordinates cleverly, it is possible to construct a vector field $X$ satisfying $\mathrm{Div}(X) = \mathrm{Div}(pX) = 0$ with compact support. So even knowing $f$ close to the boundary of the support of $\mathbb{P}$ is not sufficient to uniquely identify $f$.

5. While it is not possible to identify ICA using volume preserving transformations, it can be possible to identify $f(s)$ for certain values of $f$ if $\mathbb{P}$ is known. If $p$ has a unique maximum at $s_0$ then $x_0 = f(s_0)$ will be the point with the largest density of $x$ because volume preserving transformations transform the density trivially (see (14)).

Let us finally sketch a proof of Proposition 2.

*Proof of Proposition 2.* We assume in addition that the line segment $t_i = \{x_i + \lambda(y_i - x_i) \mid \lambda \in [0, 1]\}$ does not contain any $x_j$ or $y_j$ for $j \neq i$. The generalization to the general case is straightforward, e.g., by composing two diffeomorphisms as constructed here. It is clearly sufficient to construct volume preserving diffeomorphisms $h_i$ such that $h_i(x_i) = y_i$ and $h_i(x_j) = x_j$, $h_i(y_j) = y_j$ for $j \neq i$ which can then be composed. We consider the vector field $X_i = (y_i - x_i)\varphi$ where $\varphi$ is a smooth cut-off function with $\varphi(x) = 1$ for $x \in t_i$ and $\varphi(x_j) = \varphi(y_j) = 0$ for $j \neq i$. Using Theorem 2 in [9] we conclude that there is $Y_i$ such that $\mathrm{supp}(Y_i) \subset \mathrm{supp}(\nabla \varphi)$ and $\mathrm{Div}(X_i + Y_i) = 0$. Considering the flow of $Z_i = X_i + Y_i$ up to time 1 we obtain a function $h_i$ as desired. Indeed, $Z_i(x_j) = Z_i(y_j) = 0$ for $j \neq i$ because $x_j$ and $y_j$ are by construction of $\varphi$ outside the support of $X_i$ so $X_i(x_j) = X_i(y_j) = Y_i(x_j) = Y_i(y_j) = 0$ for $i \neq j$. Moreover, for $x \in t_i$ we get $Z_i(x) = X_i(x) + Y_i(x) = y_i - x_i$. This implies that the flow $\Phi_t^i$ of $Z$ satisfies $\Phi_t^i(x_i) = x_i + t(y_i - x_i)$ for $t \in [0, 1]$. In particular, $h_i = \Phi_t^i$ is as desired. $\square$

## H  Experimental illustration of local identifiability

We provide a toy experiment to illustrate the meaning and significance of Theorem 4[4]. Note that this shall just underpin this specific setting, for general experiments concerning the usefulness of orthogonal coordinate transforms we refer to [17]. We consider a function $f_0 \in \mathcal{F}_{\mathrm{OCT}}$ and then assume that there is a smooth time-dependent transformation $f_t$ such that $f_t \in \mathcal{F}_{\mathrm{OCT}}$ ($f_t$ is a smooth invariant deformation in the language of the paper). We only observe the changing output distribution but the latent sources are unobserved.

Suppose, however, that we know the initial mixing $f_0$, i.e., we trained an initial model $\Phi_0$, s.t., $\Phi_0 = f_0$. Then we train $\Phi_t$ starting from $\Phi_0$ such that $\Phi_t(s) \overset{\mathcal{D}}{=} f_t(s)$ where $s$ is distributed according to some base distribution $\mu$ of the latent variables. If we in addition enforce $\Phi_t \in \mathcal{F}_{\mathrm{OCT}}$ Theorem 4 essentially tells us that $\Phi_t = f_t$ while no such guarantee exists without the functional restriction. Here, we verify experimentally that this is indeed the case in a simple setting.

**Detailed Experimental Setting**  We work in dimension 2 and consider a standard normal base distribution $\mu$. We consider polar coordinates for $f$

$$f_{\mathrm{polar}}(r, \varphi) = (r \sin(\varphi), r \cos(\varphi)). \tag{160}$$

We then define $f_t^{\mathrm{pol}} = f_{\mathrm{polar}} \circ h_t$ where $h_t$ is a coordinate-wise transformation defined for $0 \leq t \leq 1$ by

$$h_t(s_1, s_2) = \left( s_1 + \frac{t}{2}\sin(s_1 + t) + 3, \frac{s_2 + t}{2} \right). \tag{161}$$

Note that we shift the first coordinate and scale the angular coordinates to ensure that the map is injective (except for the light tail of the Gaussian). The additional sin term makes the rescaling non-trivial, i.e., not just a time dependent shift. We also consider the setting where

$$f_t^{\mathrm{rot}}(s_1, s_2) = e^{tW}(4s_1, s_2) \tag{162}$$

---

[4]Code is available at `https://github.com/simonant/ident-ica`

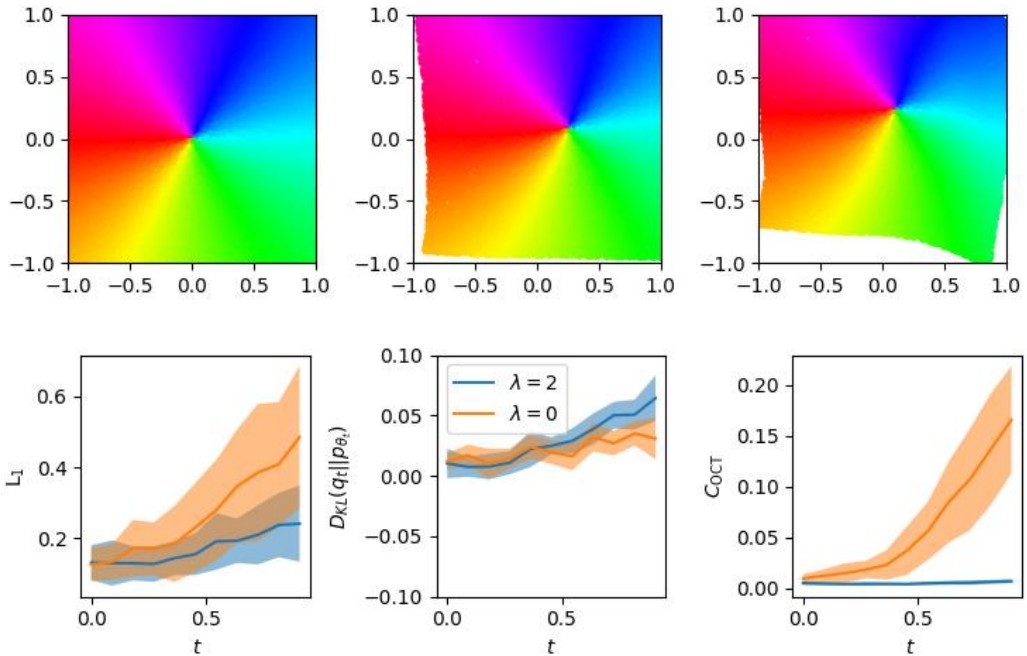

Figure 7: Experiment with $f_t^{\mathrm{pol}}$. **Top row: (left)** ground truth of latent variables **(middle)** reconstructed sources $g_\theta^{-1} \circ f_{t=1}^{\mathrm{pol}}$ for $C_{\mathrm{OCT}}$ regularized training **(right)** reconstructed sources for unregularized training **Bottom row:** Orange curves: Unregularized training. Blue curves: $C_{\mathrm{OCT}}$-regularized training. **(left)** $L_1$ distance ground truth - reconstruction over time (see (167)) **(middle)** Forward KL-divergence over time **(right)** $C_{\mathrm{OCT}}$ (see (165) over time.

and

$$W = \begin{pmatrix} 0 & 1 \\ -1 & 0 \end{pmatrix}. \tag{163}$$

To model $\Phi_t$ we use a normalizing flow model [41, 39]. while this is also convenient for the experiments, it is important to not use models that implicitly promote orthogonal columns of the Jacobian, e.g., VAEs (see end of Section 2) to extract the effect of enforcing $\Phi_t \in \mathcal{F}_{\mathrm{OCT}}$. We write $\Phi_t(s) = g(\theta_t, s) = g_{\theta_t}(s)$ where $\theta$ denote the trainable parameters of the flow which will vary with time. For our implementation we use nflows [13] and we use 5 masked affine autoregressive transformation layers with 15 hidden features followed by random permutations. Then the following procedure is used. We train the normalizing flow such that $f_0 = g(\theta_0, \cdot)$. We assume the base distribution of the flow is $\mu$ and denote the induced distribution $(g_\theta)_* \mu$ by $p_\theta$. We discretize the time interval in 10 intervals with endpoints $t_1$ to $t_{10}$. Iteratively we train $g(\theta_{t_i}, \cdot) = g(\theta_i, \cdot)$ starting from $g(\theta_{i-1}, \cdot)$ to maximize the likelihood of observations from $x_i \sim (f_{t_i})_* \mu$, i.e., we consider the loss

$$L_{\mathrm{ML}}(\theta) = \mathbb{E}_{x_i}(-\log p_\theta(x_i)). \tag{164}$$

We do this without regularization and with a regularization that promotes $g_{\theta_t} \in \mathcal{F}_{\mathrm{OCT}}$. Here we use the IMA contrast introduced in [17] which we will call $C_{\mathrm{OCT}}$ for conciseness. It is defined by

$$C_{\mathrm{OCT}}(f, \mu) = \int \mu(s) \left( \sum_k \log |\partial_k f| - \log \mathrm{Det}\, Df \right) \tag{165}$$

and we consider the total loss

$$L_{\mathrm{Reg}} = L_{\mathrm{ML}} + \lambda \cdot C_{\mathrm{OCT}}(g_\theta(\cdot), \mu). \tag{166}$$

Note that $C_{\mathrm{OCT}}$ is non-negative and vanishes exactly on OCTs (see Prop. 4.4 in [17]). We will use $\lambda = 0$ to train an unregularized model and $\lambda = 2$ for $f^{\mathrm{pol}}$ and $\lambda = 50$ for $f^{\mathrm{rot}}$ to train regularized models. The values of $\lambda$ were found through a grid search.

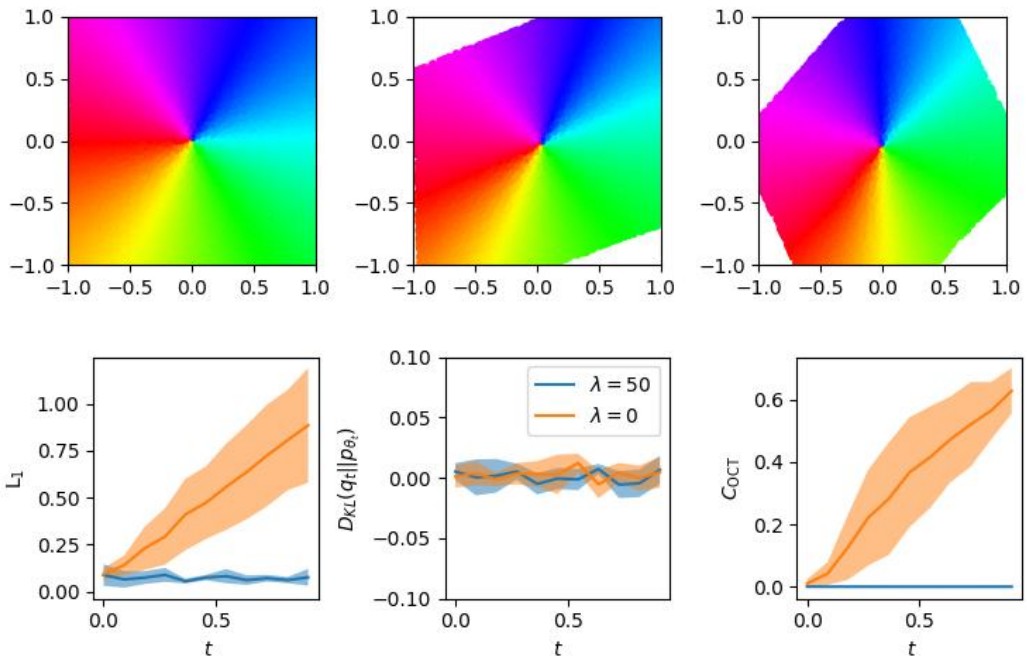

Figure 8: Same as Figure 7 for $f_t^{\mathrm{rot}}$. **Top row: (left)** ground truth of latent variables **(middle)** reconstructed sources $g_\theta^{-1} \circ f_{t=1}^{\mathrm{pol}}$ for $C_{\mathrm{OCT}}$ regularized training **(right)** reconstructed sources for unregularized training **Bottom row:** Orange curves: Unregularized training. Blue curves: $C_{\mathrm{OCT}}$-regularized training **(left)** $L_1$ distance ground truth - reconstruction over time (see (167)) **(middle)** Forward KL-divergence over time **(right)** $C_{\mathrm{OCT}}$ (see (165) over time.

**Results**  To measure how well the model recovers the ground truth sources we consider

$$L_1(\theta, t) = \int \mu(s) |g_\theta^{-1}(f_t(s)) - s|. \tag{167}$$

Note that more complicated measures like the Amari distance are not necessary because our initialization removed the permutation symmetries. We also consider $C_{\mathrm{OCT}}(g_\theta)$ to approximate the distance to $\mathcal{F}_{\mathrm{OCT}}$ and the forward KL-divergence $D_{KL}(q_i || p(\theta_i, \cdot)) = \mathbb{E}_{q_i(x)}(\log(q_i(x)) - \log(p(\theta_i, x))$ where $q_i = (f_i)_* \mu$ denotes the observational distribution at time $t_i$. Figures 7 and 8 indicate the results for the maps $f_t^{\mathrm{pol}}$ and $f_t^{\mathrm{rot}}$, respectively. Note that in both cases, the regularized and the unregularized model have small forward KL-divergence to the observational distribution, i.e., they both track the changing observational distribution. However, the regularized model recovers the ground truth latent variables more faithfully, while the unregularized model evolves towards a spurious solution. Finally, we note that the regularizer indeed ensures orthogonality of the columns of the Jacobian, i.e., $g_\theta \in \mathcal{F}_{\mathrm{OCT}}$ (small $C_{\mathrm{OCT}}$).

**Training details**  For training we use stochastic gradient descend with the ADAM-optimizer [31] with default learning rate and train for 100 steps with a batch size of 256 where we generate i.i.d. samples from the observational distribution in each step. We averaged our results over 10 runs. Total compute time was less than 48h on a workstation.