# OpenReview forum: "Function Classes for Identifiable Nonlinear Independent Component Analysis"
_NeurIPS.cc/2022/Conference — NeurIPS 2022 Accept_

### Official Review · Reviewer_uZNP · 2022-07-10

**Rating:** 6
**Confidence:** 3
**Soundness:** 3 good
**Presentation:** 3 good
**Contribution:** 3 good

**Summary:**

This paper gives some deep insights for the identifiability of the limited function classes between linear non-Gaussian ICA and unrestricted nonlinear ICA, including conformal maps, orthogonal maps and volume preserving maps.

**Questions:**

See above

**Strengths And Weaknesses:**

Overall, the paper is well written and easy to follow. I am no expert in this area, but as far as I can tell, the theoretical contributions appear sound, the quality of writing is generally strong. I believe it work will be a welcome contribution to the identifiability of nonlinear ICA. I am just confused by the real application scenarios of these function classes mentioned in this paper. It would be great if the authors can give a discussion about the real application scenarios.

---

> ### Author Response · Authors · 2022-08-02
> **Response to reviewer uZNP**
>
> We thank the reviewer for the positive assessment. We hope that our general comment
> and the replies to **reviewers PqGg and WFjz** answer your question about the real application scenarios. Please let us know if this is not the case and we are happy to answer any further questions.

---

> > ### Comment · Reviewer_uZNP · 2022-08-08
> > **Reply to authors**
> >
> > Thank you very much for the detailed rebuttal.
> >
> > I think that this draft provides valuable contributions to an exciting topic. Again, I am not an expert in this field and may not really judge the significance of this work.

---

### Official Review · Reviewer_WFjz · 2022-07-10

**Rating:** 4
**Confidence:** 4
**Soundness:** 3 good
**Presentation:** 4 excellent
**Contribution:** 3 good

**Summary:**

This manuscript extends the previous identifiability result on conformal maps where the dimension is fixed to two. Specifically, The main result is that the Mobius transformation, which is a conformal map where the dimension is larger than two, is identifiable up to trivial indeterminacies. Besides, partial/local results on the orthogonal coordinate transformation (OCT) and unidentifiability of volume-preserving transformation have been presented. No experiments have been conducted.

The theoretical contributions could be summarized as follows:

1. (Main result) Identifiability of high-dimensional conformal maps.
2. Unidentifiability of OCTs and volume-preserving transformation.
3. New notion of local identifiability and its connection with OCTs.

**Questions:**

Please refer to the above section for the corresponding questions. I will change the score if the related confusion can be addressed during the rebuttal.

**Limitations:**

The authors discussed the limitations of the work.

**Strengths And Weaknesses:**

Pros:

1. The paper is well-written.
2. Theoretical results are clearly and rigorously presented. The proof seems rigorous and its technique provides novel insight.
3. The non-identifiability of OCTs and volume-preserving transformations are helpful and consistent with intuition.
4. Limitations have been clearly and insightfully discussed.
5. The notion of local identifiability is original and interesting.

Cons:

1. The core assumption of the main result, i.e., high-dimensional conformal map, seems to be impossible in practice. In [1], OCTs (orthogonal columns of the Jacobian of the function) have been shown to be helpful to the empirical identification of nonlinear ICA. Although no identifiability result is given in [1], the corresponding intuition of independent mechanism could be found in the real world. However, compared to OCTs, high-dimensional conformal map restrict the Jacobian of the mixing function to be a scaled version of an orthogonal matrix, where the L2 norms of the columns and rows are exactly identical to each other, which seems to be impossible in practice.

2. Lack of discussion of the assumption, especially for its motivation and rationality. In L153, the motivation of OCTs has been referred to [1]. However, as described above, there exist key differences between high-dimensional conformal maps and OCTs, where the former seems to be impossible in practice. Therefore, it may be better to discuss the assumption of the main results, perhaps from a perspective of the chance/scenario of it being true. I couldn't find a related discussion in the current manuscript.

3. Lack of experiments. According to the reasons mentioned above, it is hard for me to check the significance of the main result. It may be of independent theoretical interest to study the identifiability of a specific functional form, i.e., the high-dimensional conformal map (or Mobius transformation). In that case, simulations focusing on the validity/contribution of the proposed assumption should be very helpful to support the claims. In the current manuscripts, no experimental results are presented.

4. Although OCTs (or IMA [1]) and volume-preserving transformation [2] have been proved unidentifiable in the manuscripts, there exists previous work that provides the full identifiability result with assumptions only on the mixing function. For example, [3] prove that, under assumptions of independent influences and factorial changes of volume, unconditional ICA could be identified up to trivial indeterminacies as the same as those shown in this manuscript. The assumption of independent influences seems to be closely related to OCTs and the assumption of factorial changes in volume contains volume-preserving transformation as a special case.  Therefore, it seems that the combination of the principles of the two assumptions discussed in this paper is sufficient for the identifiability of unconditional ICA. Thus, I believe the identifiability result of unconditional ICA in [3] is related to the task of this manuscript, which is currently missed in the discussion of related works.


-------

[1] Luigi Gresele, Julius von Kügelgen, Vincent Stimper, Bernhard Schölkopf, and Michel Besserve. Independent mechanism analysis, a new concept?

[2] Xiaojiang Yang, Yi Wang, Jiacheng Sun, Xing Zhang, Shifeng Zhang, Zhenguo Li, and
Junchi Yan. Nonlinear ICA using volume-preserving transformations.

[3] Yujia Zheng, Ignavier Ng, and Kun Zhang. On the identifiability of nonlinear ICA with unconditional priors.

---

> ### Author Response · Authors · 2022-08-02
> **Response to reviewer WFjz**
>
> Thank you for your detailed review and questions that helped us to underline the relevance of our contribution.
>
>  1. **Justification for conformal map assumption.**
>
> It is natural to consider the conformal maps for non-linear ICA as:
>
>    - they constitute the simplest class of non-linear function which preserve key geometric properties, namely angles, between the latent space to observation space (in contrast isometries are all linear functions),
>
>    - it is, to the best of our knowledge the only class comprising non-linear functions for which an identifiability result existed for unconditional nonlinear ICA proved by  Hyvärinen and  Pajunen in [2] (see point 4 below for discussion on related work),
>
>    - it is a strict subclass of the IMA class, which has been justified to entail causality assumptions, see Gresele et al. [1]. It is thus interesting to contrast both classical and local identifiability results of each class. Overall, conformal maps appear central to the field of unconditional non-linear ICA, as the most general class for which a classical identifiability result exists to date.
>
> 2. **Lack of discussion on the function class assumptions**
>
> We now discuss this at the end of Section 2 (see general reply and reply to **Rev. PqGg**, point 2).
>
> 3. **"Lack of experiments. [...] it is hard for me to check the significance of the main result"**
>
> The significance of this theoretical work can be directly checked from the state of the art: the last major identifiability result (see next paragraph for recent related work) for unconditional non-linear ICA dates back from more than 20 years ago and was proven for one dimensionality ($d=2$) and restrictive assumptions. We extend this result to all dimensions and reduced the number of assumptions. Encouraging experimental result for our setting (conformal and OCT) have been provided in [1] which was missing the very results that we provide: identifiability. Our contribution thus brings a key missing element to the field.
>
>
> 4. **Related work**
>
> We apologize that we did not discuss the work of Zheng et al. [3] in our submission because it appeared shortly before the submission deadline. A comparison to this work is obviously very important as the results are closely related, and  their main result may, at first look, even seem stronger than our results.  Therefore, we now explain carefully the difference between the assumptions of the main result Proposition 2.1 in [3] and our results.
> In short, Zheng et al. show identifiability only in a (very) weak sense and in particular they do not show that OCT + volume preserving is identifiable, at least in the usual sense.
>
> The crucial point is that above their Proposition 2.1 it is stated (paragraph above eq. (3) on page 2) *``Let $g : x \to \widehat{s}$ be the estimated unmixing function with [...] $J_g(x)= {O}(x) \lambda(x)$ [...] $\widehat{s}$ follows a multivariate factorial Gaussian''* which means that **they assume the unmixing function $g$ is conformal and  maps to a Gaussian density.**
> Combined with the factorized determinant assumption iii, also assumed for $g$ in the proof (below Eq. (7)), this implies that $g$ can only be a linear conformal map for $d\geq 3$, as the determinant of nonlinear conformal maps does not factorize (see Step 3 in the proof of our Lemma 2). Thus, their estimation model can only estimate linear transformations of Gaussian distributions, i.e., Gaussian distributions. For all other observational distributions the statement is empty as no unmixing $g$ satisfying the assumptions exists.
> Note additionally that even when $g$ could be any conformal map (requiring changes in their proof) the possible class of observational distributions is still parametric.
>
> This also highlights that our precise definition of identifiability is a useful contribution to the field. Note that in contrast, our results apply to (almost) any base probability distribution in the (nonlinear!) conformal case and are therefore substantially more general.
>
> We believe that this is an important point to judge the contribution of our paper so we would be happy to discuss this further should you have any questions.
>
> **References**
>
> [1] Luigi Gresele, Julius von Kügelgen, Vincent Stimper, Bernhard Schölkopf, and Michel Besserve. Independent mechanism analysis, a new concept?
>
> [2] Aapo Hyvärinen and Petteri Pajunen, Nonlinear independent component analysis: Existence and uniqueness results, Neural Networks.
>
> [3] Yujia Zheng, Ignavier Ng, and Kun Zhang. On the identifiability of nonlinear ICA with unconditional priors.

---

> > ### Comment · Reviewer_WFjz · 2022-08-07
> > **Response to the rebuttal**
> >
> > Thanks so much for the response. The clarifications authors made in the revised version, especially the discussion of the constraint of conformal maps and OCTs are helpful and make the manuscript clearer. At the same time, I politely tend to keep some of my original concerns. The largest one is the applicability of conformal maps, especially compared with OCTs. I fully agree with the author that conformal maps are widely used in several computer vision tasks related to conformal geometry. But I still hold the opinion that it may not be a rather practical assumption in a general sense, especially compared with OCTs. The assumption of conformal maps requires **the Euclidean norms of the columns of the Jacobian of the mixing/generating function to be *exactly* the same**. Even in the cases that the natural generating processes consist of exactly equivalent columns of the Jacobian, which may already be rather rare, any **noise or rescaling** could make the assumption of conformal maps invalid. Since **rescaling is one of the indeterminacies in the proposed identifiability results**, I tend to argue that the assumption of conformal maps may not be a very sensible one in the context of full identifiability [1]. However, OCTs do not have that **equivalent norms** constraint, and thus arguably be an assumption of more practical sense. Unfortunately, OCTs are only proved to be locally identifiable in the current manuscript.
> >
> > Regarding the author's response to the experimental results, honestly, I'm a little confused about the results cited (i.e., (Gresele et al.)). In (Gresele et al.), they used conformal maps as the generating process of the ground-truth data. Their results show that, without IMA-regularization in the estimating process, conformal maps are not identifiable (Figure 5.). However, in Theorem 2 of the current manuscript, I didn't find any restrictions on the estimating process, which should mean that conformal maps are identifiable without IMA-regularization. Therefore, honestly, I'm a little confused about whether (Gresele et al.) should be direct empirical support of the proposed theory or a counter-example. However, I might misunderstand something. **Please correct me if I was wrong since this is important to judge the contribution of the work.**
> >
> > Besides, thanks for helping to introduce the details of the related work. After reading the related section, I agree that that work puts constraints on the observational data. However, I'm not sure whether constraints on the observational data should be an unsolvable obstacle to the identification of the true sources. It seems that we could always add any point-wise gaussianization on the observations and then estimate the sources?
> >
> > [1] I do acknowledge that the first identifiability result (up to rotations) on nonlinear ICA with only functional constraints is based on two-dimensional conformal maps (Hyvärinen and Pajunen, 1999).

---

> > > ### Author Response · Authors · 2022-08-07
> > > **Response to reviewer WFjz**
> > >
> > > Thank you very much for engaging in the discussion. Please do not hesitate to ask further questions.
> > >
> > > ### Relevance of conformal maps.
> > >
> > > There are two aspects of your question that we like to address:
> > >
> > > - Practical relevance: We agree that the practical relevance of OCTs is substantially higher compared to conformal maps. In our first answer,  we explained why we nevertheless think that it is of theoretical interest to have an identifiability result for conformal maps.
> > >
> > > - Constraints on conformal maps: We, however, disagree with your assessment that the constraints on conformal maps are fundamentally less generic than on other function classes. Indeed, for linear functions all higher order derivatives have to vanish exactly and for OCTs the columns of the Jacobian have to be exactly orthogonal.
> > >
> > > We like to add  two comments on parts of your answer.
> > >
> > > >>  Since rescaling is one of the indeterminacies in the  proposed identifiability results, [...]  conformal maps may not be a very sensible one in the context of full identifiability
> > >
> > > The rescaling indeterminacy for conformal maps refers to equal rescaling of all coordinates, this preserves the equal norm condition and conformal maps. Does this address your general concern regarding the result?
> > >
> > > >> Unfortunately, OCTs are only proved to be locally identifiable in the current manuscript.
> > >
> > > Given the recent interest in and practical relevance of volume preserving maps and OCTs in the context of representation learning, identifiability results are of great interest to the community.
> > > We think that local identifiability is  a substantial step forward in this direction, and judged by the methodological novelty, this is a major contribution of this paper.
> > > Note that this is the first (partial) identifiability result for nonlinear mixing functions in arbitrary dimension: All previous
> > > results considered $d=2$ (and conformal) or linear mixing functions, potentially followed by a non-parametric coordinate-wise nonlinearity. It is also of practical relevance as it rules out for the first time important known classes of spurious solutions (see also the first part of our answer to **Reviewer PqGg**).
> > > While we discuss our major contributions, we also point out again that our negative result regarding volume preserving maps (even local identifiability is not satisfied), is highly relevant given recent work trying to leverage this property for identifiability.
> > >
> > > ### Confusion regarding the relation to the experiments of Gresele et al.
> > >
> > > We are happy to clarify that this is just a general misunderstanding about the form of guaranties provided by identifiability results and how such results relate to estimation/learning.
> > > The restriction of the estimating process is contained in Definition 1 where we define *identifiability*. In line with previous work, we always consider the question: Given a function class $\mathcal{F}$ and data $f(s)$ generated by $f\in \mathcal{F}$. Suppose $f'(s)$ matches the observed data **for some $\mathbf{f'\in \mathcal{{F}}}$**. Does this imply $f=f'$ up to certain symmetries, i.e., can we recover $f$? As you correctly notice, we can never identify the mixing $f$ without making assumptions on the functional form of $f$  (e.g., the Darmois construction or the measure preserving transformations discussed in the paper give generic counter-examples). This implies that unconstrained maximum likelihood, that can fit the observational distribution perfectly in the limit of infinite data, does not guaranty to retrieve the ground truth function and latent factors.
> > > Our theoretical results indeed match the experimental results by Gresele et al.: There is no uniqueness for nonlinear functions (maximum likelihood does not retrieve ground truth), but when restricting representation to OCTs (through regularization) they recover the ground truth (we prove that this holds when restricting further to conformal maps).
> > >
> > > ### '' It seems that we could always add any point-wise gaussianization on the observations and then estimate the sources?''
> > >
> > > We are uncertain whether we understand the question correctly. What exactly does point-wise refer to? We agree that the observations $x$ can be transformed to a Gaussian distribution $\hat{x}$ using a mapping $m$, e.g., leveraging a variant of the Darmois construction. Then, applying some identifiability result of class $\mathcal{F}$ to Gaussian $\hat{x}$, we would obtain $\hat{x}=f (s)$ with $f\in\mathcal{F}$ and $s$ Gaussian. However, this result translates to $x= (m^{-1}\circ f) (s)$ for the original problem, and the class of solution $m^{-1}\circ f$ is unknown a priori, so we cannot check whether identifiability is achieved.
> > > In short, to obtain an identifiability result in the sense explained above one cannot simply consider arbitrary mappings to Gaussian distributions. Instead, one needs to preserve the considered function class, which essentially brings us back to the task of solving the initial problem.

---

> > > > ### Author Response · Authors · 2022-08-09
> > > > **Applicability of conformal maps**
> > > >
> > > > We now clarified that there is a simple extension of our result on conformal maps to coordinate-wise reparameterized conformal maps. This class is stable under rescaling of the sources and does not enforce equal norms of the Jacobian columns.
> > > > More details on this can be found in our reply to **Reviewer PqGg** and in the updated manuscript that we uploaded.

---

### Official Review · Reviewer_PqGg · 2022-07-11

**Rating:** 4
**Confidence:** 3
**Soundness:** 3 good
**Presentation:** 3 good
**Contribution:** 3 good

**Summary:**

This paper studies the identifiability of nonlinear ICA by focusing on different types of function classes (i.e., conformal, orthogonal, and volume preserving maps). In particular, the paper extends previous results of dimension 2 to show that ICA with conformal maps is identifiable for dimension larger than 2. The authors then define the notion of local identifiability and show that ICA with orthogonal maps is locally identifiable. Finally, the authors show that ICA with volume preserving maps is not identifiable. Throughout the theoretical analysis, the authors draw connection to the rigidity theory.

**Questions:**

As noted in the "weaknesses" section above, it is helpful to provide more explanations and empirical studies regarding local identifiability and the different function classes.

**Limitations:**

Yes

**Strengths And Weaknesses:**

Strengths:
- The notion of local identifiability is novel to the best of my knowledge
- Theoretical analysis for different function classes and the connection with rigidity theory are provided and appear to be sound

Weaknesses: Overall, while this paper provides theoretical analysis for nonlinear ICA, in my opinion it will be helpful if there are some empirical studies to illustrate the theoretical claims. Details comments are as follows.
- Since the concept of local identifiability is new, it would be more appealing to provide more explanations regarding the intuition of local identifiability (e.g., how it specifically contributes to one identifying the desired independent components in practice), and empirical studies to demonstrate its practical applications. This may be necessary for others to better understand the significance of local identifiability. On the other hand, for the empirical studies, how does one evaluate the performance of an algorithm w.r.t. local identifiability?
- Related to my point above, more explanations (e.g., in what situation one will expect the corresponding function assumption to likely hold) and empirical studies for the function classes will also make the contributions more convincing.
- While the theoretical analysis of identifiability for conformal map might be potentially interesting, it might be rather restrictive and less likely to hold, and thus might limit its practical applications.

-----
After reading the responses and discussion, including the further comments from the authors and also the other reviewers, my concern about the applicability of the assumption persists.

---

> ### Author Response · Authors · 2022-08-02
> **Response to reviewer PqGg**
>
> Thank you for your careful review and your suggestions that will help us clarify the significance of our results.
>
> 1. **``Provide more explanations regarding the intuition of local identifiability''**
>
> We view this result as an indication that OCTs may be identifiable, and a starting point to prove this. In particular, it excludes a large class of known spurious solutions for unconstrained nonlinear ICA as we explain in the paper.
> Note that many initial theoretical results in, e.g., optimization or neural network theory are
> local in the sense that they assume closeness to the optimum of the initialization.
> We also now extended the result (see general reply) so that it can be applied to settings of practical relevance. Namely, the result proves identifiability in a setting of concept drift: Suppose we know the factors of variations
> of some initial mixing $f(t=0)$ that is an OCT. Then we can find the
> unmixing of a smoothly deformed OCT $f(t)$  given access to the observational distribution along the path.
> On the other hand, any change of a (known) mixing function makes  the resulting model unidentifiable for general non-linear functions. We refer to Appendix H for an illustration.
>
> We do not think (local or global) identifiability is a tool to evaluate the performance of an algorithm. It is used to explain the success or failures of algorithms in certain setting (because they are, e.g., in our setting,  biased towards OCTs).
>
> 2. **``More explanations (e.g., in what situation one will expect the corresponding function assumption to likely hold) and empirical studies for the function classes''**
>
> Regarding the OCT class (see point below for conformal maps), multiple theoretical and empirical works in representation learning emphasize that orthogonality of the Jacobian is an implicit bias in variational autocencoders (VAEs) and $\beta$-VAEs [1,2,3] that appears useful for disentanglement. Indeed, these algorithms are widely used in representation learning and often recover semantically meaningful representations  [4-7]. This suggests that the OCT assumptions is a useful inductive bias for classical applications of disentanglement representation learning, e.g., image generation. Therefore, our result are highly relevant to the practice of disentangled representation learning. We added this point at the end of section 2.
>
>
> 3. **Restrictiveness of conformal maps**
>
> We agree that conformal maps is a restrictive assumption, however it plays de facto a central role in non-linear ICA, see point 1 in the reply to reviewer WFjz for details.
>
> **References**
>
> [1] Michal Rolínek, Dominik Zietlow, and Georg Martius. Variational autoencoders pursue
> PCA directions (by accident).  CVPR 2019.
>
> [2]  Abhishek Kumar and Ben Poole. On implicit regularization in $\beta$-vaes. ICML 2020.
>
> [3]     Dominik Zietlow, Michal Rolínek, and Georg Martius. Demystifying inductive biases for
> (beta-)VAE based architectures. ICML 2021.
>
> [4] Abhishek Kumar, Prasanna Sattigeri, and Avinash Balakrishnan. Variational inference of disentangled latent concepts from unlabeled observations. ICLR 2018.
>
> [5] Tian Qi Chen, Xuechen Li, Roger B. Grosse, and David Duvenaud. Isolating Sources of Disentanglement in Variational Autoencoders. NeurIPS 2018
>
> [6] Hyunjik Kim and Andriy Mnih. Disentangling by factorising. ICML 2018.
>
> [7] Christopher P Burgess, Irina Higgins, Arka Pal, Loic Matthey, Nick Watters, Guillaume Des383
> jardins, and Alexander Lerchner. Understanding disentangling in $\beta$-VAE. arXiv preprint 2018.

---

> ### Author Response · Authors · 2022-08-09
> **Request for feedback**
>
> Dear Reviewer PqGg, we thank you for considering our rebuttal. We would be very grateful for any additional feedback on the updates implemented in the revision. We agree that local identifiability is a slightly involved notion. So, we would in particular appreciate any comment on whether the updated presentation and the experimental illustration clarified the setting of local identifiability and which aspects could be further improved.

---

> > ### Comment · Reviewer_PqGg · 2022-08-09
> > **Thanks for the response**
> >
> > Thanks for the detailed response and the revision with experimental illustrations. After reading other reviews, my concern about the applicability of conformal maps (also mentioned by Reviewer WFjz) still remains, since it might not be practical (while I understand that the first identifiability result leverages conformal maps for dimension 2). For instance, as Reviewer WFjz mentioned, any rescaling of some (not all) variables may lead to violation of conformal mapping assumption, and rescaling is one of the indeterminacies for nonlinear ICA.

---

> > > ### Author Response · Authors · 2022-08-09
> > > **Applicability of conformal maps**
> > >
> > > Thank you, we appreciate your feedback, as it helped us clarify the broader generality of our result on conformal maps. While we agree that the conformal assumption is somewhat artificially restrictive in the way it constrains equality of the norms, this does not prevent exploiting this result in contexts where this assumption is relaxed. Indeed, we can simply  define the parameterized conformal maps class
> > > $\mathcal{F_{{\rm r-conf}}}$ consisting of conformal maps precomposed by coordinate-wise reparametrizations. Notably, this relaxes the equal norm constraint on the columns of the Jacobian and is still contained in OCTs.
> > > This class is then identifiable up to entrywise non-linear rescaling of each source, as a consequence of Theorem 2. The idea is simple: we can rescale entry-wise the sources to convert the global identifiability problem in $\mathcal{F_{\rm r-conf}}$ into a global identifiability problem in $\mathcal{F_{\rm conf}}$, which is solved by Theorem 2. This entails the symmetry indeterminacy set $\mathcal{S_{\rm conf}}$ is turned into the broader class $\mathcal{S_{\rm reparam}}$, identical to the OCT case, and which is a reasonable set of ambiguities for practical applications. We updated the manuscript accordingly with **Corollary 1**, and truly hope this clarifies the question raised about the restrictiveness of norm equality constraint and rescalings of the conformal class, raised as a major concern by you and **Reviewer WFjz**.

---

### Official Review · Reviewer_4HKX · 2022-07-11

**Rating:** 6
**Confidence:** 3
**Soundness:** 4 excellent
**Presentation:** 3 good
**Contribution:** 3 good

**Summary:**

Two common ways of avoiding the non-identifiability in nonlinear ICA are 1) conditioning on auxiliary variables, and 2) staying in the unconditional setting but instead restricting the function classes.  This paper focuses on the latter (i.e. unconditional ICA), and studies the identifiability for conformal maps, orthogonal coordinate transforms (OCTs), and measure-preserving maps.

Specifically, the results are (summarized in Table 1):
- For conformal maps, the paper extends Hyvarinen & Pajunen 1999's results from d=2 to d>2.
- For OCTs, the paper proves a weaker notion of "local identifiability", and conjectures that ICA would be identifiable for "typical" pairs of elements, though the formal statement is left for future work. In addition, local identifiability does not hold for nonlinear maps in general, highlighting the importance of constraining to OCTs.
- Measure-preserving maps are not locally identifiable and hence also not identifiable.

Proof technique wise, the paper draws connection to rigidity theory and tools from PDEs.


**Questions:**

- Thm 3: Noting that $\mathcal{S}_{\text{conf}}$ contains a shift, could you explain how is this stronger than the identifiability result in Hyvarinen & Pajunen 1999?
- Clarification: Thm 8: does $|\cdot|$ denote the l2 norm?

=== Update after discussion ===
The authors have answered this in the reply.

**Limitations:**

The paper discusses the limitations well.

There's no direct societal impact.

**Strengths And Weaknesses:**

Strengths
- The paper provides an alternative lens (i.e. restricting the function class) to circumvent the non-identifiability problem in ICA.
- The paper draws connection to techniques in PDEs. Concretely:
  - The local identifiability result in Theorem 4 follows from first casting the independence conditions of ICA as second order PDEs and then applying the uniqueness of the Cauchy problem of second-order hyperbolic PDEs.
  - The constraints in Theorem 5 are because the Dirichlet problem may not be well posed in general and doesn't always have a unique solution, unless under some specific constraints.
- The paper is clear about technical limitations, provides partial results whenever possible, and provides interesting open directions.

Weaknesses: I have no major concerns about the paper. I should note that I'm not the best judge for the impact of the results and the technique novelty.

---

> ### Author Response · Authors · 2022-08-02
> **Response to reviewer 4HKX**
>
>  We are thankful for the positive review and the questions. Our replies to the two questions raised are:
>
> 1. We understand the questions as follows: *Given that shifts are contained in our symmetry group, couldn't this shift be used to enforce, $f(0)=0$ which is the assumption in Hyvärinen \& Pajunen 1999?* Please let us know if we did not understand the question correctly.
>
>     Hyvärinen \& Pajunen show that the conformal mixing is unique (up to a rotation) if
>     we fix the support of the distribution, and we fix the function value at 0 (i.e., $f(0)=0$).
>     We phrase our result differently, but essentially, it boils down to the statement that the mixing is unique (up to reflections and potentially coordinate exchange)  as soon as we match the support of the distribution.
>     So, we removed the second assumption.
>     To connect this to your question, we note that the
>     condition on the support already fixes the shift from the symmetry group, and
>     we cannot choose the shift such that $f(0)=0$ is ensured.
>
>     Note that in addition, we also filled the remaining gap
>     in their result (identification of the rotation, arbitrary support sizes).
>
> 2. Yes, we will also add a footnote to clarify this.
>
> We are happy to answer any further questions and concerns.

---

> > ### Comment · Reviewer_4HKX · 2022-08-06
> > **Thank you for the clarifications**
> >
> > Thank you to the authors for clarifying my concern. I stand by my original review and recommend an accept.

---

### Author Response · Authors · 2022-08-02
**General response to all reviewers**

We thank the reviewers for their constructive reviews which agree that this paper
is a sound, substantial, and well written theoretical contribution.
The reviews state that *"This paper gives some deep insights for the identifiability of the limited function classes between linear non-Gaussian ICA and unrestricted nonlinear ICA"* (**Reviewer uZNP**), and it *"provides an alternative lens (i.e. restricting the function class) to circumvent the non-identifiability problem in ICA"* (**Reviewer 4HKX**). In spite of some recent progress, the theoretical understanding of ICA is still incomplete, and we believe the current paper provides a serious contribution to this, even if --- as the reviews point out --- some questions remain open.

We revised our submission to address the reviewers' concerns and think that our changes clarify and increase the relevance of our contribution. The main additions to the paper are the following (colored blue in the revised document):

- **(Undercomplete ICA)** We observed that with very minor changes our results apply to mixing functions mapping to manifolds. This covers in particular the case of submanifolds of $R^{n}$ and thus undercomplete ICA (i.e, $f:R^d\to R^{n}$ with $d<n$). This setting has been rarely investigated theoretically and make our results more practically relevant for two reasons:

    1.  In most applications, few factors of variations generate data in a high-dimensional space, a setting our results now cover.
    2. While the class of conformal maps mapping $R^d\to R^d$ is extremely rigid, this is no longer true when considering conformal maps or orthogonal coordinate transformations mapping $R^d\to R^{n}$, $d<n$. Indeed, already the subclass of local isometries ($Df^\top Df = \mathrm{Id}$) is quite flexible and has been used in ML frequently (e.g. see [1]).



- **(Relevance of function classes)**
In addition to the first point, we discuss the relevance of the function class in a specific paragraph at the end of Section 2. We added references showing that conformal maps can be applied in computer vision and how orthogonality of the Jacobian columns
is a favorable inductive bias for disentanglement. Our answer to **Reviewer PqGg** explains the relevance of OCTs in more detail and
our reply to **Reviewer WFjz** provides a justification to consider ICA with conformal maps. The last two points address concerns raised by **Reviewers WFjz, PqGg**, and **4HKX**.


- **(Definition of local stability)**
We generalized Theorem 4 to cover the case where the ground truth mixing $f_t$ is time varying (we also adapted the definition of local identifiability). This makes our result applicable  to  settings with concept drift [1]. We also added a toy experiment in Appendix H to illustrate local stability and the implications of Theorem 4. This addresses
criticism from **Reviewer PqGg** and we expand our answer in our direct reply to **Reviewer PqGg**.



We admit that this is a pure theory paper, but we hope that the reviewers will still find our work worthy of acceptance. We note that NeurIPS has traditionally indeed published theory papers if the results are relevant and the methods used are nontrivial --- we believe that both is the case, since we address a long-standing problem highly relevant to Machine Learning, namely the *identifiability of nonlinear latent variable models*, using mathematical techniques not normally seen at NeurIPS.


**References**

[1] Gerhard Widmer and Miroslav Kubat, Learning in the presence of concept drift and hidden contexts, Machine Learning, 1996

[2] Nonlinear Dimensionality Reduction by Locally Linear Embedding, Sam T. Roweis and Lawrence K. Saul. Science 2000.

---

### Author Response · Authors · 2022-08-09
**Overview of the discussion period**

Dear Reviewers, Area Chair, and Senior Area Chair,

We thank the reviewers for engaging actively in the discussion.
Here, we gather key elements scattered across the multiple comments on this page, to provide you with an overview of the points raised by reviewers and how we addressed them.
There were no doubts cast about the soundness and relevance of the theoretical contributions. Instead, the discussion focused on the illustration and practical relevance of the results with the following key points:

- The review of **Reviewer PqGg** pointed out that an illustration of the principle underlying local identifiability would be appreciated. We addressed this request by introducing the notion of *concept drift* and adding an experiment in the appendix illustrating this setting. The result notably showcases the inability to locally identify the ground truth model within the class of general non-linear functions, while the OCT class constraint allows it.

- Also in the initial review, **Reviewers PqGg, uZNP and WFjz** requested empirical studies that justify the relevance of the function classes. We referred to those studies and updated the manuscript accordingly. **Reviewers WFjz** had further questions on how the study of Gresle et al. relates to our result, which we clarified by explaining how identifiability within a function class relates to constraining the function estimation process.


- The restrictiveness of the conformal class was pointed out multiple times by **Reviewer PqGg** and **Reviewer WFjz**, both in their initial review and further discussion. Specifically, there was the concern that
> "high-dimensional conformal map restrict the Jacobian of the mixing function to be a scaled version of an orthogonal matrix, where the L2 norms of the columns and rows are exactly identical to each other, which seems to be impossible in practice." (**Reviewer WFjz**)

    and
    > "any rescaling of some (not all) variables may lead to violation of conformal mapping assumption" (**Reviewer PqGg**)

    While we generally agree that conformal maps are less relevant for practical settings we  address these specific concerns by a new corollary of Theorem 2 that makes clear that the equality of Jacobian column norms assumption can be relaxed to yield **the broader class of reparameterized conformal maps (contained in OCTs), which does not suffer from this restriction and also satisfies global identifiability.**

Overall, we think we have left no issues open and thank the reviewers for helping us improve the accessibility of this work to a broader audience.

---

### Meta-Review · Area_Chair_2i85 · 2022-08-27

**Recommendation:** Accept
**Confidence:** Certain

**Metareview:**

The paper studies identifiability of ICA for two families of non-linear functions: conformal maps and orthogonal coordinate transforms. For conformal maps, they prove identifiability for d > 2, improving an old '99 result for d=2 due to Hyvarinen and Pajunen. For orthogonal coord. transforms, they prove a weaker notion of "local" identifiability.

There was quite a lot of discussion on the various strengths and weakness of the paper.

(1) *Experiments*: Though the paper had very little experiments, the reviewers agreed that since the paper is primarily a theory paper, an extensive experimental section is not necessary.

(2) *Theory*: The reviewers found both the proofs of Theorems 2 and 3 quite interesting, involving new ideas. They're heavy on tools from complex analysis, which is not surprising giving that conformal maps are natural through the lens of complex analysis; but they found the connections to PDEs interesting and potentially useful in the future. There were some potential worries about correctness, but no definite error was identified.

(3) *How strong of an assumption is conformality in practice*: The reviewers agree this is probably quite restrictive as an assumption, but idenfiability of non-linear ICA is always going to require some strong conditions, and we're still very far from understanding when it's possible (when no auxiliary variables are involved). The paper shrinks the gap b/w theory and practice even if the theory has very strong assumptions.

**Award:**

No

---

### Decision · Program_Chairs · 2022-09-14

Accept